# Piecewise-Stationary Bandits with Knapsacks

**Xilin Zhang**
Department of ISEM
National University of Singapore
Singapore, 117578
zhangxilin@u.nus.edu

**Cheung Wang Chi**
Department of ISEM
National University of Singapore
Singapore, 117578
isecwc@nus.edu.sg

## Abstract

We propose a novel inventory reserving algorithm which draws new insights into Bandits with Knapsacks (Bwk) problems in piecewise-stationary environments. Suppose parameters $\eta_{\min}, \eta_{\max} \in (0, 1]$ respectively lower and upper bound the ratio between the reward earned and the resources consumed in a round. Our algorithm achieves a provably near-optimal competitive ratio of $O(\log(\eta_{\max}/\eta_{\min}))$, with a matching lower bound provided. Our performance guarantee is based on a *dynamic benchmark* that upper bounds the optimum, different from existing works on adversarial Bwk Immorlica et al. (2019); Kesselheim and Singla (2020) who compare with the stationary benchmark. Different from existing non-stationary Bwk work Liu et al. (2022), we do not require a bounded global variation.

## 1  Introduction

In a bandits with knapsack (Bwk) problem, each action $a$ in the action set $\mathcal{K}$ is associated with a latent and random amount of reward earned, $R_t(a)$, and resource consumed, $C_t(a)$, in each round $t = 1, \ldots, T$. A decision maker (DM) selects an action $a_t \in \mathcal{K}$ in round $t$, and observes bandit feedback $(R_t(a_t), C_t(a_t))$. The DM targets at maximizing the total reward $\sum_{t=1}^{T} R_t(a_t)$, while satisfying the hard capacity constraint $\sum_{t=1}^{T} C_t(a_t) \leqslant B$. Bwk has many real-life applications such as dynamic pricing Babaioff et al. (2015), resource allocation Zhalechian et al. (2022), online auction Balseiro and Gur (2019) and assortment planning Agrawal et al. (2019). Stochastic Bwk is first introduced by Badanidiyuru et al. (2018), followed by generalizations to concave reward with convex constraints Agrawal and Devanur (2014), combinatorial bandits Sankararaman and Slivkins (2018) and contextual bandits Badanidiyuru et al. (2014); Agrawal and Devanur (2016). In stochastic Bwk problems, the expected feedback $\mathbb{E}[(R_t(a), C_t(a))] = (r(a), c(a))$ is stationary for all $a \in \mathcal{K}$, $t \in \{1, \ldots, T\}$, and a sublinear-in-$T$ regret is achievable. Nevertheless, the stationary model could be too ideal in many applications.

Adversarial Bwk is firstly considered in Immorlica et al. (2019) where $(r_t, c_t) = \{(r_t(a), c_t(a))\}_{a \in \mathcal{K}}$ can change arbitrarily over the horizon. They achieve a competitive ratio (CR) of $O(d \log(T))$ with respect to a *static benchmark* when there are $d$ budget constraints. A static benchmark picks a fixed optimal action (or a fixed optimal distribution over arms), and applies the same action (or distribution) in all $T$ rounds. Kesselheim and Singla (2020) further improve the CR to $O(\log(d) \log(T))$. Other papers consider different regimes such as unlimited rounds (Rangi et al. (2018)), large budget $B = \Omega(T)$ (Castiglioni et al. (2022a)), strict feasibility (Castiglioni et al. (2022b)) and approximate stationarity (Fikioris and Tardos (2023)). All these works compare with static benchmarks (see Appendix A.1). Moreover, adversarial Bwk could be too conservative in certain real-life scenarios. For instance, sales patterns could be stationary for a duration of time, but only change during periods of hot seasons/promotions/new trends, which fits into our piecewise-stationary Bwk regime.

38th Conference on Neural Information Processing Systems (NeurIPS 2024).

An abundance of existing works explore adversarial online knapsack problems with full feedback, where $(r_t, c_t)$ can change arbitrarily but their realized value of $\{(R_t(a), C_t(a))\}_{a \in \mathcal{K}}$ is observed before choosing $a_t$ (Karp et al. (1990); Mehta et al. (2007); Zhou et al. (2008)). Many of these works compare their accrued rewards with dynamic benchmarks (stronger than the static benchmarks used in adversarial Bwk), where the DM picks different optimal actions $a_t^*$ in different rounds. However, the dynamic benchmarks considered in the above papers are *best single arm* benchmark, which only allows pulling a single arm in each round; while our benchmark is a *best distribution over arms* benchmark (see Appendix A.2 for a more detailed elaboration). Further, the ability of observing $\{(R_t(a), C_t(a))\}_{a \in \mathcal{K}}$ before selecting $a_t$ is crucial in the algorithm designs in Karp et al. (1990); Mehta et al. (2007); Zhou et al. (2008). Their algorithm design cannot be readily generalized to the bandit setting, where the DM only observes $(R_t(a_t), C_t(a_t))$ after selecting $a_t$.

Another line of recent research investigates non-stationary online knapsack problems with either full feedback Jiang et al. (2020); Balseiro et al. (2022) or bandit feedback Liu et al. (2022). These works quantify the scale of non-stationarity in terms of both the local variation $\mathsf{loc} = \sum_{t=1}^{T-1} \mathrm{dist}((r_{t+1}, c_{t+1}), (r_t, c_t))$ and the global variation $\mathsf{glo} = \sum_{t=1}^{T} \mathrm{dist}(\sum_{t=1}^{T}(r_t, c_t)/T, (r_t, c_t))$, where dist is a certain metric. Assuming $\max\{\mathsf{loc}, \mathsf{glo}\}$ grows sublinearly in $T$, they achieve sublinear-in-$T$ regret bounds compared to the dynamic benchmark. However, $\mathsf{glo}$ growing sublinearly in $T$ is rather strong assumption. We could have $\mathsf{glo}$ linear in $T$, even with one change point (see Remark A.3 for more detail). In this work, we consider piece-wise stationary models that allow a higher degree of non-stationarity, which is yet to be studied in all aforementioned works.

**Our contributions.** Firstly, on **modeling** (see Section 2), the DM does not know the number of change points and when changes happen. We formulate our model as a single-resource problem, and extend to $d$-resource problems in Appendix B.5 with an extra multiplicative factor of $d$ on the competitve ratio. Secondly, on **algorithm design** (see Sections 3.1 and 4.1), we propose novel algorithms which are natural, intuitive and easy to implement. Our idea of reserving inventory based on the reward-consumption ratio provides new insights into the problem. Thirdly, on **performance guarantee** (see Sections 3.2 and 4.2), we achieve a provably near-optimal competitive ratio with respect to a *best distribution over arms* benchmark without requiring a bounded $\mathsf{glo}$, which distinguishes our work from the existing literature. Specifically, suppose there exists parameters $\eta_{\min}, \eta_{\max} \in (0, 1]$ such that $\eta_{\min} \leqslant r_t(a), c_t(a) \leqslant \eta_{\max}$ for all $t, a$. Our algorithms achieve a competitive ratio of $O(\log(\eta_{\max}/\eta_{\min}))$, which requires a novel analysis. We prove the tightness of our competitive ratio by providing a matching lower bound (see Section 4.4). We also run some illustrative **numerical experiments** (see Section 5) to compare our algorithm performance with Immorlica et al. (2019) and Zhou et al. (2008) under the piecewise-stationary settings.

## 2 Model

### 2.1 Problem formulation

**Problem dynamics.** The online model involves $T$ rounds, indexed as $t \in \mathcal{T} = \{1, 2, \ldots, T\}$. We index an arm as $a \in \mathcal{K}$. Additionally, we define the null arm $a_{\mathrm{null}}$, where no allocation is made when $a_{\mathrm{null}}$ is chosen. In round $t \in \mathcal{T}$, the DM chooses an arm $a_t \in \mathcal{K} \bigcup \{a_{\mathrm{null}}\}$, and observes a noisy outcome vector $(R_t(a_t), C_t(a_t)) \in [0, 1]^2$ as the bandit feedback, where $R_t(a_t)$ and $C_t(a_t)$ are the reward and the resource consumption in round $t$ respectively. We set $R_t(a_{\mathrm{null}}) = C_t(a_{\mathrm{null}}) = 0$ with certainty for all $t \in \mathcal{T}$. The DM is endowed with $B \leqslant T$ units of the resource. The DM's goal is to maximize the total reward, with the constraint that the total resource consumption is at most $B$ with certainty. Denote $r_t = \{r_t(a)\}_{a \in \mathcal{K}} = \{\mathbb{E}[R_t(a)]\}_{a \in \mathcal{K}}$ and $c_t = \{c_t(a)\}_{a \in \mathcal{K}} = \{\mathbb{E}[C_t(a)]\}_{a \in \mathcal{K}}$. We consider a piece-wise stationary setting, where the planning horizon $\mathcal{T}$ is partitioned into $L$ stationary pieces $\{t_0 = 1, \ldots, t_1\}, \{t_1 + 1, \ldots, t_2\}, \ldots, \{t_{L-1} + 1, \ldots, t_L = T\}$. On each stationary piece $l$, we have $(r_t, c_t) = (r^{(l)}, c^{(l)})$ for all $t \in \{t_{l-1} + 1, \ldots, t_l\}$.

The DM does *not* know the number of rounds $T$, the number of stationary pieces $L$, the rounds $t_1, \ldots, t_{L-1}$ where changes happen, the values of $\{(r^{(l)}, c^{(l)})\}_{l \in \{1, \ldots, L\}}$ and their realized outcomes.

**Goal and benchmark.** Our goal is to develop an online algorithm that maximize the expected total reward $\mathbb{E}[\sum_{t=1}^{T} R_t(a_t)]$ while satisfying the inventory constraint $\sum_{t=1}^{T} C_t(a_t) \leqslant B$, which can be formulated as the following dynamic program DP. In DP, $X_t = \{X_t(a)\}_{a \in \mathcal{K}}$ is a binary decision vari-

able indicating whether to pick arm $a$ in round $t$ (i.e., $X_t(a) = 1$) or not (i.e., $X_t(a) = 0$). An online algorithm is non-anticipatory in the sense that $X_t$ depends only on $B$ and $\{(R_s(a_s), C_s(a_s), X_s)\}_{s=1}^{t-1}$.

$$\text{DP} := \max_{X_t} \mathbb{E}\left[\sum_{t=1}^{T}\sum_{a\in\mathcal{K}} R_t(a)X_t(a)\right] \qquad \text{FA} := \max \sum_{l=1}^{L}(t_l - t_{l-1})\sum_{a\in\mathcal{K}} r^{(l)}(a)x_l(a)$$

$$\text{s.t. } \sum_{t=1}^{T}\sum_{a\in\mathcal{K}} C_t(a)X_t(a) \leqslant B \qquad \text{s.t. } \sum_{l=1}^{L}(t_l - t_{l-1})\sum_{a\in\mathcal{K}} c^{(l)}(a)x_l(a) \leqslant B$$

$$\sum_{a\in\mathcal{K}} X_t(a) \leqslant 1 \quad \forall t \in \mathcal{T} \qquad \sum_{a\in\mathcal{K}} x_l(a) \leqslant 1 \quad \forall l = 1,\ldots,L$$

$$X_t(a) \in \{0,1\} \quad \forall a \in \mathcal{K},\ t \in \mathcal{T}. \qquad x_l(a) \geqslant 0 \quad \forall a \in \mathcal{K},\ l = 1,\ldots,L.$$

In non-stationary bandits without resource constraints, the performance bound of an online algorithm is in the form of $\sum_{t=1}^{T} R_t(a_t) \geqslant \sum_{t=1}^{T} r_t(a_t^*) - \text{Reg}$, where $\sum_{t=1}^{T} r_t(a_t^*)$ is the optimal expected reward obtained by choosing the best arm in each round, and Reg is a sublinear-in-$T$ regret characterizing the reward loss. In our Bwk setting, due to the inventory constraint, achieving a sublinear-in-$T$ regret is impossible without assuming a bounded global variation (see Appendix A.3). We denote opt(P) as the optimum of an optimization problem P. We aim for a performance guarantee of the form $\sum_{t=1}^{T} R_t(a_t) \geqslant \frac{1}{\text{CR}} \cdot \text{opt(DP)} - \text{Reg}$, where opt(DP) is the optimum of DP, CR is a competitive ratio, and Reg is a sublinear-in-$T$ regret. Unfortunately, DP is hard to solve. Therefore, we define a fluid approximation FA of DP, where $R_t$ and $C_t$ are replaced by their respective expectations, and the decision variables are fractional. In the following Lemma 2.1 (proved in Appendix C.1), we justify that opt(FA) can serve as a benchmark for our algorithms' performance since

$$\sum_{t=1}^{T} R_t(a_t) \geqslant \frac{1}{\text{CR}} \cdot \text{opt(FA)} - \text{Reg} \geqslant \frac{1}{\text{CR}} \cdot \text{opt(DP)} - \text{Reg},$$

and we aim to derive performance guarantees of the form $\sum_{t=1}^{T} R_t(a_t) \geqslant \frac{1}{\text{CR}} \cdot \text{opt(FA)} - \text{Reg}$.

**Lemma 2.1.** *opt(FA)$\geqslant$opt(DP).*

## 2.2 Assumptions, limitations and discussions

Compared with the adversarial Bwk literature, our piecewise-stationary setting has two limitations. The first is on $L$, the number of change points. When $L$ is not known, our result is meaningful only when $L = o(\sqrt{T \cdot \eta_{\min}})$. When $L$ is known, our result is meaningful when $L = o(T \cdot \eta_{\min})$ (see Theorem 4.2). In contrast, existing works on adversarial Bwk generally allow $L = T$. The second is on the value range of non-null actions:

**Assumption 2.2.** For all $a \in \mathcal{K}, l \in \{1,\ldots,L\}$, there exists known constants $\eta_{\min}, \eta_{\max} \in (0,1]$ such that $\eta_{\min} \leqslant r^{(l)}(a), c^{(l)}(a) \leqslant \eta_{\max}$.

While $\eta_{\min}$ can be as small as $0$ generally, we argue that this assumption is mild. Assumption 2.2 holds in many real-life scenarios. For instance, in portfolio management, an investor allocates a limited budget among different investment options (arms) to maximize the overall return. The investor has assessments on lower and upper ranges of the expected returns for each investment option. The lower range is usually strictly positive, since the investor would not consider investment options with $0$ or negative expected return. In applications such as dynamic pricing, assortment planning, network resource allocation and energy management, expected profits and consumer demands are usually within a known positive value range. We further justify that Assumption 2.2 *theoretically* in the following Lemma 2.3 (proved in Appendix C.5).

**Lemma 2.3.** *For any online algorithm, there exist an instance for which $0 \leqslant r_t(a), c_t(a) \leqslant 1$ for all $a \in \mathcal{K}, t \in \mathcal{T}$, and that CR$> \Omega(\log(\eta_{\max}/\eta_{\min}))$.*

Additionally, in Appendix C.5 we demonstrate that knowing the values of $\eta_{\min}, \eta_{\max}$ is *necessary* for achieving CR$= O(\log(\eta_{\max}/\eta_{\min}))$.

## 2.3 High-level idea of our algorithm

**Decomposing opt(FA) in terms of reward-consumption ratios.** Throughout our paper, we fix an optimal solution $\{x_l^*\}_{l=1}^L$ to FA. We define the set $\mathcal{L} = \{l \in \{1, \ldots, L\} : \sum_{a \in \mathcal{K}} x_l^*(a) > 0\}$, which indexes stationary pieces where non-null allocations are made under the optimal solution $\{x_l^*\}_{l=1}^L$ to FA . For each $l \in \mathcal{L}$, we define

$$\text{Ratio}^{(l)*} = \frac{\sum_{a \in \mathcal{K}} r^{(l)}(a) x_l^*(a)}{\sum_{a \in \mathcal{K}} c^{(l)}(a) x_l^*(a)} \in \left[\frac{\eta_{\min}}{\eta_{\max}}, \frac{\eta_{\max}}{\eta_{\min}}\right], \quad B_l^* = (t_l - t_{l-1}) \sum_{a \in \mathcal{K}} c^{(l)}(a) x_l^*(a). \quad (1)$$

$\text{Ratio}^{(l)*}$, whose value range follows from Assumption 2.2, is the optimal expected reward earned per unit of consumed resource under $\{x_l^*\}_{l=1}^L$. We call $\text{Ratio}^{(l)*}$ the optimal expected *reward-consumption ratio* of stationary piece $l$. $B_l^*$ represents the optimal expected amount of resources assigned for stationary piece $l$. To aid our algorithm design, we define the following linear program:

$$\text{LP}(\tilde{r}, \tilde{c}, \tilde{B}) := \max \sum_{a \in \mathcal{K}} \tilde{r}(a) x(a)$$

$$\text{s.t.} \sum_{a \in \mathcal{K}} \tilde{c}(a) x(a) \leqslant \tilde{B}$$

$$\sum_{a \in \mathcal{K}} x(a) \leqslant 1$$

$$x(a) \geqslant 0 \qquad \forall a \in \mathcal{K}.$$

Then, we can express opt(FA) in terms of $\text{LP}(\tilde{r}, \tilde{c}, \tilde{B})$ and $\text{Ratio}^{(l)*}, B_l^*$ as follows:

$$\text{opt(FA)} = \sum_{l \in \mathcal{L}} (t_l - t_{l-1}) \cdot \text{opt}\left(\text{LP}\left(r^{(l)}, c^{(l)}, B_l^*/(t_l - t_{l-1})\right)\right) = \sum_{l \in \mathcal{L}} \text{Ratio}^{(l)*} \cdot B_l^*. \quad (2)$$

The first equation in (2) can be verified by noting that $x_l^*$ is feasible to $\text{LP}(r^{(l)}, c^{(l)}, B_l^*/(t_l - t_{l-1}))$ for each $l \in \{1, \ldots, L\}$, and the concatenation of the optimal solutions of $\{\text{LP}(r^{(l)}, c^{(l)}, B_l^*/(t_l - t_{l-1})\}_{l \in \mathcal{L}}$ forms a feasible solution to FA. The second equation in (2) holds, by the definitions of $\text{Ratio}^{(l)*}, B_l^*$ and the fact that $\text{opt}(\text{LP}(r^{(l)}, c^{(l)}, B_l^*/(t_l - t_{l-1}))) = \sum_{a \in \mathcal{K}} r^{(l)}(a) x_l^*(a)$.

**Algorithm design.** Fix an arbitrary constant $\alpha > 1$ (we set $\alpha = e$ by default, but our results hold for any constant $\alpha > 1$). We define $M = \lceil \log_\alpha(\eta_{\max}/\eta_{\min}) \rceil$ and partition $[\eta_{\min}/\eta_{\max}, \eta_{\max}/\eta_{\min}]$ into $2M$ intervals $[\alpha^{-M}, \alpha^{-M+1}] \bigcup \{(\alpha^m, \alpha^{m+1}]\}_{m=-M+1}^{M-1}$. For each stationary piece $l \in \mathcal{L}$, we denote $m_l^* \in \{-M, \ldots, M-1\}$ as the interval such that $\text{Ratio}^{(l)*} \in (\alpha^{m_l^*}, \alpha^{m_l^*+1}]$. In the forthcoming discussion, with some abuse of notation, we sometimes write interval $[\alpha^{-M}, \alpha^{-M+1}]$ as $(\alpha^{-M}, \alpha^{-M+1}]$, and we refer to interval $(\alpha^m, \alpha^{m+1}]$ as reward-consumption ratio interval $m$, or "interval $m$" in short. Then we can decompose opt(FA) regarding reward-consumption ratio intervals:

$$\text{opt(FA)} = (2) = \sum_{m=-M}^{M-1} \sum_{l \in \mathcal{L}} \text{Ratio}^{(l)*} \cdot \mathbf{1}(\text{Ratio}^{(l)*} \in (\alpha^m, \alpha^{m+1}]) \cdot B_l^*$$

$$= \sum_{m=-M}^{M-1} \sum_{l \in \mathcal{L}} \text{Ratio}^{(l)*} \cdot \mathbf{1}(m_l^* = m) \cdot B_l^*. \quad (3)$$

The *key intuition* of our algorithm is to achieve a reward guarantee for each interval $m$ regarding the reward-consumption ratio $\mathbf{1}(m_l^* = m) \cdot \text{Ratio}^{(l)*}$ and the resource consumption $\sum_{l \in \mathcal{L}} \mathbf{1}(m_l^* = m) \cdot B_l^*$, which is done by performing two tasks: (a) for each $l \in \mathcal{L}$, we guess the value of $m$ such that $\text{Ratio}^{(l)*} \in (\alpha^m, \alpha^{m+1}]$. We guarantee that for at least $1/(M+1)$ fraction of requests on each $l$, our guessed ratio interval are close to the correct interval $m_l^*$; (b) for each interval $m$, we "reserve" $B/2M$ resource units. That is, we reserve an inventory of $B/2M$ resource units to satisfy requests with a guessed reward-consumption ratio interval $m$. When the inventory reserved for interval $m$ is depleted, the DM rejects (by choosing $a_{\text{null}}$) all future requests with a guessed interval $m$.

By accomplishing task (a), we ensure that for each $l \in \mathcal{L}$, at least $\mathbf{1}(m_l^* = m) \cdot B_l^*/(M+1)$ requested resources are served by resources reserved for interval $m$, generating reward at a ratio of at least

$\alpha^m$. Then, by accomplishing task (b), if the reserved inventory for interval $m$ are not depleted by round $T$, our algorithm earns a reward of at least $\alpha^m \cdot \mathbf{1}(m_l^* = m) \cdot B_l^*/(M+1)$ during stationary piece $l$. Else, if the reserved $B/2M$ resource units for interval $m$ are depleted by round $T$, then the DM earns a reward of at least $\alpha^m \cdot B/(2M) \geqslant \alpha^m \cdot \sum_{l \in \mathcal{L}} \mathbf{1}(m_l^* = m) \cdot B_l^*/(2M)$ from resources reserved for interval $m$, since $\sum_{l \in \mathcal{L}} \mathbf{1}(m_l^* = m) \cdot B_l^* \leqslant B$. By judiciously analyzing the relationship between stationary pieces and reward-consumption ratio intervals, for each interval $m$, we ensure that a reward of

$$\frac{1}{O(M)} \cdot \sum_{l \in \mathcal{L}} \alpha^m \cdot \mathbf{1}(m_l^* = m) \cdot B_l^*$$

is accrued, which is $1/O(M)$ of the benchmark resources consumed on all stationary pieces $l \in \mathcal{L}$ whose Ratio$^{(l)*} \in (\alpha^m, \alpha^{m+1}]$. By summing over $m \in \{-M, \ldots, M-1\}$, we achieve $1/O(M)$ fraction of reward (3).

## 3 Warm-up: Full-feedback deterministic outcome setting

In this section, we introduce the main idea of our algorithm on the bandit model by relaxing the model uncertainty assumptions and specializing to the full-feedback deterministic setting: $(R_t, C_t) = (r^{(l)}, c^{(l)})$ with certainty for each $t \in \{t_{l-1}+1, \ldots, t_l\}$. Thus, $(r^{(l)}, c^{(l)})$ is observed at the start of the stationary piece $l$. The DM does not know $L$ and $\{t_1, \ldots, t_{L-1}\}$ before the online process begins. The DM's decision can be fractional, which means on each stationary piece $l$, a decision can take the form of $x_l \in \{x \in [0,1]^{|\mathcal{K}|} : \sum_{a \in \mathcal{K}} x(a) \leqslant 1, x(a) \geqslant 0 \ \forall a \in \mathcal{K}\}$, resulting in a reward of $\sum_{a \in \mathcal{K}} r^{(l)}(a)x_l(a)$ and resource consumption of $\sum_{a \in \mathcal{K}} c^{(l)}(a)x_l(a)$ in a round.

### 3.1 Inventory REServing (IRES) algorithm

Upon observing $(R_t, C_t) = (r^{(l)}, c^{(l)})$, guessing Ratio$^{(l)*}$ (see Section 2.3) is equivalent to guessing $x_l^*$, which is equivalent to guessing $B_l^*/(t_l - t_{l-1})$, since $\{x_l^*\}_{l=1}^L$ is an optimal solution to LP$(r^{(l)}, c^{(l)}, B_l^*/(t_l - t_{l-1}))$. In this section, when we say "Line xx", we refer to a line of Algorithm 1, which displays IRES. At the start, the IRES *reserves* $B/2M$ units of resource to each interval $m \in \{-M, \ldots, M-1\}$, and each resource unit is reserved by exactly one interval. In Line 5, for each stationary piece $l$, we firstly solve LP$(r^{(l)}, c^{(l)}, \eta_{\min} \cdot \alpha^q)$ for each $q \in \{0, \ldots, M\}$, and get an optimal solution $x_l^{(q)*} = \{x_l^{(q)*}(a)\}_{a \in \mathcal{K}}$. Each $\eta_{\min} \cdot \alpha^q$ is a guess of $B_l^*/(t_l - t_{l-1})$. For each $q \in \{0, \ldots, M\}$, we define

$$\text{Ratio}_l^{(q)} := \frac{\sum_{a \in \mathcal{K}} r^{(l)}(a)x_l^{(q)*}(a)}{\sum_{a \in \mathcal{K}} c^{(l)}(a)x_l^{(q)*}(a)}$$

as a guess of Ratio$^{(l)*}$. Claim 3 in Appendix B.3 shows that, by guessing a $q$ such that $\eta_{\min} \cdot \alpha^q$ is within a factor of $\alpha$ from $B_l^*/(t_l - t_{l-1})$, we also have Ratio$_l^{(q)}$ to be at most a factor of $\alpha$ from Ratio$^{(l)*}$. As time progresses on $l$, we go round-robin on the choices of $q_t \in \{0, \ldots, M\}$ for each round $t$ (Lines 7, 15). In round $t \in \{t_{l-1}+1, \ldots, t_l\}$, we identify $m_t \in \{-M, \ldots, M-1\}$ such that Ratio$_l^{(q_t)} \in (\alpha^{m_t}, \alpha^{m_t+1}]$ (Line 8). If there remains enough reserved resource units for interval $m_t$ (Line 10), the DM fulfils the $t$-th request by selecting fractional action $x_t = x_l^{(q_t)*}$ (Line 11), which consumes resources reserved for $m_t$ (Lines 9, 11). Otherwise, the DM selects $a_{\text{null}}$ and rejects the request (Line 13). By Line 9, we have $\mathcal{T}_t^{(m)} = \{s \in \{1, \ldots, t\} : m_s = m\}$, which consists of rounds in $\{1, \ldots, t\}$ when the DM attempts to fulfil a request with resources reserved for interval $m$.

### 3.2 Performance guarantee of IRES

We provide a performance guarantee to IRES in Theorem 3.1.

**Theorem 3.1.** *For any given $\alpha > 1$, IRES achieves a reward of at least*

$$\frac{\left(1 - \frac{2\log_\alpha(\eta_{\max}/\eta_{\min})+1}{B}\right) \cdot opt(FA)}{6\alpha^2 \cdot \log_\alpha(\eta_{\max}/\eta_{\min})},$$

*under mild requirements that $t_l - t_{l-1} \geqslant M + 1 \ \forall l \in \mathcal{L}$. In particular, IRES achieves a competitive ratio of $O(\log_\alpha(\eta_{\max}/\eta_{\min}))$ if $B \geqslant \Omega(\log_\alpha(\eta_{\max}/\eta_{\min}))$.*

---

**Algorithm 1** Inventory REServing with deterministic input (IRES)

---
1: Input: resource capacity $B$, $\eta_{\min}$, $\eta_{\max}$.
2: Initialize $l = 0$, $t = 1$, $q_1 = 0$, $\mathcal{T}_t^{(m)} = \varnothing$ for all $m, t$.
3: **while** $t \leqslant T$ **do**
4:      Set $l = l + 1$.
5:      Solve LP$(r^{(l)}, c^{(l)}, \eta_{\min} \cdot \alpha^q)$ $\forall q \in \{0, 1, \ldots, M\}$ for optimal $x_l^{(q)*} = \{x_l^{(q)*}(a)\}_{a \in \mathcal{K}}$.
6:      **while** stationary piece $l$ not end **do**
7:          Let $(r_t, c_t) = (r^{(l)}, c^{(l)})$, $x_t = x_l^{(q_t)*}$.
8:          Find $m_t \in \{-M, \ldots, M - 1\}$ such that $\text{Ratio}_l^{(q_t)} \in (\alpha^{m_t}, \alpha^{m_t+1}]$.
9:          Set $\mathcal{T}_t^{(m_t)} = \mathcal{T}_{t-1}^{(m_t)} \bigcup \{t\}$.
10:         **if** $\sum_{s \in \mathcal{T}_{t-1}^{(m_t)}} \sum_{a \in \mathcal{K}} c_s(a) x_s(a) \leqslant \frac{B}{2M} - 1$ **then**
11:            Pick fractional arms $x_t$.
12:         **else**
13:            Pick arm $a_t = a_{\text{null}}$.
14:         **end if**
15:         **if** $q_t \leqslant M - 1$ **then** set $q_{t+1} = q_t + 1$ **else** set $q_{t+1} = 0$.
16:         Set $t = t + 1$.
17:      **end while**
18: **end while**

---

*Remark* 3.2 (Comparing with online knapsack problems). Our deterministic setting resembles online knapsack problems with adversarial $(r_t, c_t)$ revealed in each round Zhou et al. (2008), but our $(r_t, c_t)$ remains the same for an unknown number of rounds. Assuming $B \geqslant \Omega(\eta_{\max})$, Zhou et al. (2008) achieve a competitive ratio of $2 \log(\eta_{\max}/\eta_{\min}) + 1$ and they provide a nearly-matching lower bound. We recover their competitive ratio with an extra $3\alpha^2$ multiplicative factor in a piece-wise stationary setting and a stricter requirement on $B$.

### 3.3 Analysis

Denote $\mathcal{T}_T^{(m)} = \{\tau^{(m)}(1), \tau^{(m)}(2), \ldots\}$ where $\tau^{(m)}(1) < \tau^{(m)}(2) < \ldots$, and $\tilde{\mathcal{T}}^{(m)}$ is the prefix of $\mathcal{T}_T^{(m)}$ satisfying

$$\tilde{\mathcal{T}}^{(m)} = \left\{ \tau^{(m)}(n) \in \mathcal{T}_T^{(m)} : \sum_{s=1}^n \sum_{a \in \mathcal{K}} c_{\tau^{(m)}(s)}(a) x_{\tau^{(m)}(s)}(a) \leqslant \frac{B}{2M} - 1 \right\}.$$

That is, $\tilde{\mathcal{T}}^{(m)}$ consist up to the last round assigned to interval $m$ such that the reserved inventory is not fully consumed. It is evident that if $\sum_{s \in \mathcal{T}_T^{(m)}} c_s(a_s) \leqslant B/(2M) - 1$, then $\tilde{\mathcal{T}}^{(m)} = \mathcal{T}_T^{(m)}$.

Define $\mathcal{J}_l = \{t_{l-1} + 1, \ldots, t_l\}$, the time interval of the $l$-th piece. The reward achieved by IRES is REW $= \sum_{t \in \mathcal{T}} \sum_{a \in \mathcal{K}} r_t(a) x_t(a)$, which can be decomposed as REW $= \sum_{m=-M}^{M-1} \text{REW}^{(m)}$ where

$$\text{REW}^{(m)} = \sum_{l \in \mathcal{L}} \mathbf{1}(m_l^* = m) \sum_{t \in (\bigcup_{n=-M}^{M-1} \tilde{\mathcal{T}}^{(n)}) \bigcap \mathcal{J}_l} \sum_{a \in \mathcal{K}} r_t(a) x_t(a).$$

The set $(\bigcup_{n=-M}^{M-1} \tilde{\mathcal{T}}^{(n)}) \bigcap \mathcal{J}_l$ consists of rounds in stationary piece $l$, which requests are not rejected due to shortage in reserved resource units. The summation $\sum_{l \in \mathcal{L}} \mathbf{1}(m_l^* = m)$ yields the reward accrued on pieces where $m_l^* = m$. By the summation $\sum_{m=-M}^{M-1} \text{REW}^{(m)}$, we obtain the total reward accrued with resources reserved for $2M$ intervals.

Similarly, we decompose the benchmark opt(FA) $= \sum_{m=-M}^{M-1} \text{opt(FA)}^{(m)}$ where

$$\text{opt(FA)}^{(m)} = \sum_{l \in \mathcal{L}} \mathbf{1}(m_l^* = m) \cdot \text{Ratio}^{(l)*} \cdot B_l^*,$$

To prove Theorem 3.1, it suffices to show $\text{REW}^{(m)} \geqslant \frac{1 - (2M+1)/B}{6\alpha^2 M} \cdot \text{opt(FA)}^{(m)}$ for each interval $m$, as in the following Claim 1 and Claim 2. Then Theorem 3.1 can be established by summing over $m \in \{-M, \ldots, M - 1\}$.

**Claim 1.** For any interval $m \in \{-M, \ldots, M-1\}$, if for all $n \in \{\max\{m-1, -M\}, m\}$ we have $\tilde{\mathcal{T}}^{(n)} = \mathcal{T}_T^{(n)}$, then $\text{REW}^{(m)} \geqslant \frac{1}{2\alpha M} \cdot \text{opt(FA)}^{(m)}$.

**Claim 2.** For any interval $m \in \{-M, \ldots, M-1\}$, if for at least one element $n \in \{\max\{m-1, -M\}, m\}$ we have $\tilde{\mathcal{T}}^{(n)} \subsetneq \mathcal{T}_T^{(n)}$, then $\text{REW}^{(m)} \geqslant \frac{1-(2M+1)/B}{6\alpha^2 M} \cdot \text{opt(FA)}^{(m)}$.

**Sketch proofs of Claims 1, 2.** Claims 1, 2 are proved in Appendices C.2, C.3 respectively. We first show in Claim 3 in Appendix B.3 that on a stationary piece $l \in \mathcal{L}$, there exists a "correct" $q_l^* \in \{0, \ldots, M\}$, such that when selecting decision $x_t = x_l^{(q_l^*)*}$ (the optimal solution to the LP$(r^{(l)}, c^{(l)}, \eta_{\min} \cdot \alpha^{q_l^*})$), our guess $\text{Ratio}_l^{(q_l^*)}$ on the ground-truth $\text{Ratio}^{(l)*} \in (\alpha^{m_l^*}, \alpha^{m_l^*+1}]$ satisfies

$$\text{Ratio}_l^{(q_l^*)} \in (\alpha^{m_l^*-1}, \alpha^{m_l^*+1}] = (\alpha^{m_l^*-1}, \alpha^{m_l^*}] \cup (\alpha^{m_l^*}, \alpha^{m_l^*+1}]. \tag{4}$$

When taking fractional action $x_l^{(q_l^*)*}$, we consume resources reserved for reward-consumption ratio intervals $m_l^* - 1$ or $m_l^*$. Therefore by our round-robin design, on each stationary piece $l$ such that $m_l^* = m$, at least $(t_l - t_{l-1})/(M+1)$ requests are assigned to intervals $m-1$ or $m$, under decision $x_l^{(q_l^*)*}$. It remains to analyze how many requests are fulfilled by resources reserved for the correct interval $m_l^* = m$ at the correct reward-consumption ratio $\text{Ratio}_l^{(q_l^*)}$, as discussed in Section 2.3.

For an interval $m$ where $\tilde{\mathcal{T}}^{(m)} = \mathcal{T}_T^{(m)}$, $\tilde{\mathcal{T}}^{(m-1)} = \mathcal{T}_T^{(m-1)}$ (Claim 1 case), there are still remaining resources reserved for intervals $m-1, m$ at the end of the horizon. Hence, for each stationary piece $l$ such that $m_l^* = m$, at least $(t_l - t_{l-1})/(M+1)$ requests (consuming $B_l^*/(M+1)$ resource units) are indeed fulfilled by resources reserved for interval $m-1$ or $m$, accruing reward at the reward-consumption ratio of at least $\text{Ratio}^{(l)*}/\alpha$ according to (4). Summing over all $l$ such that $m_l^* = m$, we have $\text{REW}^{(m)} \geqslant \sum_{l \in \mathcal{L}} (\text{Ratio}^{(l)*}/\alpha) \cdot \mathbf{1}(m_l^* = m) \cdot B_l^*/(M+1)$ and Claim 1 is validated. For an interval $m$ where there exists some $n \in \{\max\{m-1, -M\}, m\}$ such that $\tilde{\mathcal{T}}^{(n)} \subsetneq \mathcal{T}_T^{(n)}$ (Claim 2 case), the $B/2M$ resource units reserved for interval $n$ are depleted before the end of the horizon. In this case, some requests on stationary piece $l$ where $m_l^* = m$ may be rejected, but the $B/(2M)$ resource units reserved for interval $n$ have been consumed, generating reward at a reward-consumption ratio of at least $\alpha^n \geqslant \alpha^{m-1}$. Since the total resources that should be consumed w.r.t. interval $m$ under the optimal FA solution is $\sum_{l \in \mathcal{L}} \mathbf{1}(m_l^* = m) \cdot B_l^* \leqslant B$, we have $\text{REW}^{(m)} \geqslant \alpha^{m-1} \cdot B/(2M) \geqslant \sum_{l \in \mathcal{L}} (\text{Ratio}^{(l)*}/\alpha^2) \cdot \mathbf{1}(m_l^* = m) \cdot B_l^*$, and Claim 2 is validated.

## 4 Bandit-feedback stochastic outcome setting

In this section, we consider the original piece-wise stationary Bwk model, where the DM receives bandit feedback on outcomes $(R_t, C_t)$, and decisions are randomized.

### 4.1 Inventory REServing with change monitoring (IRES-CM) Algorithm

In this section, when we say "Line xx", we refer to a line of Algorithm 2, which displays IRES-CM. In the bandit-feedback setting, guessing $\text{Ratio}^{(l)*}$ requires estimating $(r^{(l)}, c^{(l)})$. To do so, we adaptively partition $\mathcal{T}$ into exploration rounds and exploitation rounds. In each round $t$, we conduct exploration with probability $\gamma_t = M\sqrt{|\mathcal{K}| \log(1/\delta)(\log(|\mathcal{K}|) + 1)}/\sqrt{Nt}$ (reflected in a Bernoulli random variable $U(t)$ in Line 7), where $\delta \in (0, 1)$ is a confidence parameter and $N$ is defined in (5). In an exploration round $t$ (Lines 9-13), we uniformly sample an arm $a \in \mathcal{K}$ and pull it for $N$ consecutive rounds. We update an estimate $(\hat{r}_t(a), \hat{c}_t(a))$ on $(r_t(a), c_t(a)) = (r^{(l)}(a), c^{(l)}(a))$ using the $\{(R_s(a), C_s(a))\}_{s \in \mathcal{T}_t^S(a)}$ information, where $\mathcal{T}_t^S(a)$ denotes the set of the most recent $N$ exploration rounds before round $t$ when arm $a$ is pulled. That is, we set $\mathcal{T}_t^S(a) = \{\tau \in \{t - s_t, \ldots, t-1\} : a_\tau = a\}$ where $s_t = \arg\max_s \{\sum_{\tau=t-s}^{t-1} \mathbf{1}(a_\tau = a) = N\}$. We define

$$N = \frac{27 \log(2/\delta)}{(1 - 1/\sqrt{\alpha})^2 \cdot \eta_{\min}}, \quad \hat{r}_t(a) = \frac{\sum_{s \in \mathcal{T}_t^S(a)} R_s(a)}{N}, \quad \hat{c}_t(a) = \frac{\sum_{s \in \mathcal{T}_t^S(a)} C_s(a)}{N}. \tag{5}$$

The estimates $\hat{r}_t, \hat{c}_t$ have two sources of error: error due to random noise, which decreases with $N$; and error due to non-stationarity, which increases with $N$. We set $N$ according to (5) to balance these two errors. We let $\mathcal{T}_t^R$ denote the set of exploration rounds.

In an exploitation round $t$ (Lines 17-25), we take turns to pull arms according to decision $\hat{x}_t^{(q_t)*}$, which is very similar to Algorithm 1 with $(\hat{r}_t, \hat{c}_t)$ in place of $(r_t, c_t)$. We define

$$\widehat{\text{Ratio}}_t^{(q)} = \frac{\sum_{a\in\mathcal{K}} \hat{r}_t(a)\hat{x}_t^{(q)*}(a)}{\sum_{a\in\mathcal{K}} \hat{c}_t(a)\hat{x}_t^{(q)*}(a)}, \quad (\dagger)_t = \max\left\{\min\left\{\widehat{\text{Ratio}}_t^{(q_t)}, \alpha^M\right\}, \alpha^{-M}\right\},$$

which can both be interpreted as a guess of $\text{Ratio}^{(l)*}$ at any round $t$ during stationary piece $l$. We reserve $B/(2M)$ units of resources for each interval $m \in \{-M, \ldots, M-1\}$. In round $t$, we serve request $t$ using resources reserved for interval $\hat{m}_t$ such that $(\dagger)_t \in (\alpha^{\hat{m}_t}, \alpha^{\hat{m}_t+1}]$. If interval $\hat{m}_t$ has remaining reserved inventory, then we pull arm $a_t = a$ with probability $\hat{x}_t^{(q_t)*}(a)$, or $a_t \sim \hat{x}_t^{(q_t)*}$ in short. We let $\mathcal{T}_t^{I(m)}$ denote the set of exploitation rounds using resources reserved for interval $m$. We finally highlight that the major performance difference between IRES and IRES-CM is due to estimating $(r_t, c_t)$ by $(\hat{r}_t, \hat{c}_t)$, which is detailed in Section 4.3.

---

**Algorithm 2** Inventory REServing with Change Monitoring (IRES-CM)

---

 1: Input: resource capacity $B$, rate $\gamma$, bounding parameters $\eta_{\min}, \eta_{\max}$.
 2: Set $\mathcal{T}_t^R = \varnothing$ for all $t$ and $\mathcal{T}_t^{I(m)} = \varnothing$ for all $m, t$.
 3: Pull each arm $a \in \mathcal{K}$ for $N$ times, get $\hat{r}_t(a), \hat{c}_t(a)$ as in (5).
 4: Set $t = N|\mathcal{K}| + 1$.
 5: **while** $t \leqslant T$ **do**
 6:     Solve LP$(\hat{r}_t, \hat{c}_t, \eta_{\min} \cdot \alpha^q)$ $\forall q \in \{0, 1, \ldots, M\}$ for optimal $\hat{x}_t^{(q)*} = \{\hat{x}_t^{(q)*}(a)\}_{a\in\mathcal{K}}$.
 7:     Sample $U(t) \sim \text{Bern}(\gamma_t)$.
 8:     **if** $U(t) = 1$ **then**
 9:         Pick arm $a \sim \text{Uni}(\mathcal{K})$, pull arm $a_s = a$.
10:         Set $U(s) = 1$ for $s \in \{t, \ldots, t+N-1\}$.
11:         Set $\mathcal{T}_s^R = \mathcal{T}_{t-1}^R \bigcup \{t, \ldots, s\}$ for $s \in \{t, \ldots, t+N-1\}$.
12:         Set $t = t+N$, $(\hat{r}_t, \hat{c}_t) = (\hat{r}_{t-N}, \hat{c}_{t-N})$.
13:         Update $\hat{c}_t(a), \hat{r}_t(a)$ as in (5).
14:     **else**
15:         **for** $q = 0, \ldots, M$ **do**
16:             Set $q_t = q$.
17:             Determine $\hat{m}_t \in \{-M, \ldots, M-1\}$ such that $(\dagger)_t \in (\alpha^{\hat{m}_t}, \alpha^{\hat{m}_t+1}]$.
18:             Set $\mathcal{T}_t^{I(\hat{m}_t)} = \mathcal{T}_{t-1}^{I(\hat{m}_t)} \bigcup \{t\}$.
19:             **if** $\sum_{s\in\mathcal{T}_{t-1}^{I(\hat{m}_t)}} C_s(a_s) \leqslant \frac{B}{2M} - 1$ **then**
20:                 Pick arm $a_t \sim \hat{x}_t^{(q_t)*}$.
21:             **else**
22:                 Pick arm $a_t = a_{\text{null}}$.
23:             **end if**
24:             Set $t = t+1$, $(\hat{r}_t, \hat{c}_t) = (\hat{r}_{t-1}, \hat{c}_{t-1})$.
25:         **end for**
26:     **end if**
27: **end while**

---

## 4.2 Performance guarantee of IRES-CM

We impose the following assumption on the ranges of $B$, opt(FA).

**Assumption 4.1.** $\min\{B, \text{opt(FA)}\} \geqslant \tilde{\Omega}(L\sqrt{|\mathcal{K}|NT})$, where $\tilde{\Omega}(\cdot)$ hides multiplicative factors in terms of $\log_\alpha(\eta_{\max}/\eta_{\min}), \log(1/\delta), (\log(|\mathcal{K}|) + 1)$.

The performance of IRES-CM is as follows:

**Theorem 4.2.** *For any given $\alpha > 1$, with probability at least $1 - 2|\mathcal{K}| \cdot (\log_\alpha(\eta_{\max}/\eta_{\min})L + T)\delta$, IRES-CM achieves a reward of at least*

$$\frac{1 - o(1)}{10\alpha^4 \cdot \log_\alpha(\eta_{\max}/\eta_{\min})} \cdot \left(opt(FA) - \tilde{O}\left(L\sqrt{|\mathcal{K}|NT}\right)\right)$$

*under Assumption 4.1, where $o(\cdot)$ hides multiplicative factors in terms of $\sqrt{M/B}$ and $\tilde{O}(\cdot)$ hides multiplicative factors in terms of $\log_\alpha(\eta_{\max}/\eta_{\min})$, $\log(1/\delta)$, $(\log(|\mathcal{K}|)+1)$. In particular, IRES-CM achieves a competitive ratio of $O(\log_\alpha(\eta_{\max}/\eta_{\min}))$ as long as $L = o(\sqrt{T \cdot \eta_{\min}})$.*

The proof of Theorem 4.2 can be found in Appendix C.4. We provide a thorough comparison of our performance guarantee with existing works on adversarial and non-stationary Bwk in Appendix A.

*Remark* 4.3 (Improved performance with known $L$). If the DM knows $L$, Assumption 4.1 can be relaxed to

$$\min\{B, \text{opt(FA)}\} \geqslant \tilde{\Omega}(\sqrt{L|\mathcal{K}|NT}).$$

Furthermore, in our performance guarantee in Theorem 4.2, the deductive term $\tilde{O}(L\sqrt{|\mathcal{K}|NT})$ from opt(FA) can be improved to $\tilde{O}(\sqrt{L|\mathcal{K}|NT})$ by setting the exploration parameter $\gamma_t = M\sqrt{L|\mathcal{K}|\log(1/\delta)(\log(|\mathcal{K}|)+1)}/\sqrt{Nt}$ in IRES-CM. Without prior knowledge of $L$, the deductive term $\tilde{O}(L\sqrt{|\mathcal{K}|NT}) = o(T)$ if $L = o(\sqrt{T \cdot \eta_{\min}})$; with prior information of $L$, the deductive term $\tilde{O}(\sqrt{L|\mathcal{K}|NT}) = o(T)$ if $L = o(T \cdot \eta_{\min})$.

*Remark* 4.4 (Deterministic setting with bandit feedback). In our full-feedback deterministic setting, since $(r^{(l)}, c^{(l)})$ is given at the beginning of each stationary piece, our performance guarantee is independent on $|\mathcal{K}|, L$. In a bandit-feedback deterministic setting, IRES-CM can be applied by setting $N = 1$. In this case, under Assumption 4.1, IRES-CM achieves a reward of at least

$$\frac{1 - o(1)}{6\alpha^2 \cdot \log_\alpha(\eta_{\max}/\eta_{\min})} \cdot \left(\text{opt(FA)} - \tilde{O}(L\sqrt{|\mathcal{K}|T})\right).$$

## 4.3 Analysis

We denote $\sigma_t(a) = \min\{s : s \in \mathcal{T}_t^{\text{S}}(a)\}$ as the 1st element in $\mathcal{T}_t^{\text{S}}(a)$. We partition the exploitation round set $\mathcal{T}_T^{\text{I}(m)}$ into two sets $\check{\mathcal{T}}^{\text{I}(m)}$ and $\hat{\mathcal{T}}^{\text{I}(m)}$, i.e., $\mathcal{T}_T^{\text{I}(m)} = \check{\mathcal{T}}^{\text{I}(m)} \bigcup \hat{\mathcal{T}}^{\text{I}(m)}$, $\check{\mathcal{T}}^{\text{I}(m)} \bigcap \hat{\mathcal{T}}^{\text{I}(m)} = \varnothing$. A time index $t \in \mathcal{T}_T^{\text{I}(m)}$ belongs to the set $\check{\mathcal{T}}^{\text{I}(m)}$ (referred to as "successful exploitation rounds regarding interval $m$") if and only if the following condition is satisfied for all $a \in \mathcal{K}$:

$$\{(r_s(a), c_s(a))\}_{a \in \mathcal{K}} = \{(r_{\sigma_t(a)}(a), c_{\sigma_t(a)}(a))\}_{a \in \mathcal{K}}, \quad \forall s \in \{\sigma_t(a), \dots, t\}. \tag{6}$$

For $t \in \hat{\mathcal{T}}^{\text{I}(m)}$ (referred to as "failed exploitation rounds regarding interval $m$"), inequality (6) is violated for at least one $a \in \mathcal{K}$. We denote $\hat{\mathcal{T}}^{\text{I}(m)} = \{\tau^{I(m)}(1), \tau^{I(m)}(2), \dots\}$ where $\tau^{I(m)}(1) < \tau^{I(m)}(2) < \dots$. We let $\tilde{\mathcal{T}}^{\text{I}(m)}$ be a prefix of $\hat{\mathcal{T}}^{\text{I}(m)}$ satisfying

$$\tilde{\mathcal{T}}^{\text{I}(m)} = \left\{\tau^{I(m)}(n) \in \check{\mathcal{T}}^{\text{I}(m)} : \sum_{s=1}^{n} \sum_{a \in \mathcal{K}} C_{\tau^{I(m)}(s)}\left(a_{\tau^{I(m)}(s)}\right) \leqslant \frac{B}{2M} - 1\right\}$$

which consists up to the last exploitation round satisfying (6) for interval $m$, such that the reserved resource is adequate. If $\sum_{s \in \mathcal{T}_T^{\text{I}(m)}} C_s(a_s) \leqslant B/(2M) - 1$, then $\tilde{\mathcal{T}}^{\text{I}(m)} = \check{\mathcal{T}}^{\text{I}(m)}$.

**Sketch proof of Theorem 4.2.** Recall that the performance guarantee of our algorithms is in the form of $\sum_{t=1}^{T} R_t(a_t) \geqslant \frac{1}{\text{CR}} \cdot \text{opt(FA)} - \text{Reg}$. The proof consists of mainly two steps: (a) we derive the CR$= O(M)$ by bounding two different cases of interval $m$ in a similar manner to Claim 1 and Claim 2 (see Appendix C.4), with $\check{\mathcal{T}}^{\text{I}(m)}$ (successful exploitation rounds in IRES-CM) in place of $\mathcal{T}_T^{(m)}$ (all rounds in IRES); (b) we derive the Reg$= \tilde{O}(L\sqrt{|\mathcal{K}|NT})$ by bounding the number of exploration rounds $|\mathcal{T}_T^{\text{R}}|$ and failed exploitation rounds $|\bigcup_{m=-M-2}^{M} \hat{\mathcal{T}}^{\text{I}(m)}|$ (see Appendix B.7).

**Comparing performance of IRES and IRES-CM.** We highlight that the major performance difference between IRES and IRES-CM is the loss caused by estimating $(r_t, c_t)$, reflected in the following aspects: (i) reward loss caused by exploration (upper bounding $|\mathcal{T}_T^{\text{R}}|$); (ii) $\mathcal{T}_t^{\text{S}}(a)$ contains change points, causing failed estimation of $(r_t, c_t)$ (upper bounding $|\bigcup_{m=-M-2}^{M} \hat{\mathcal{T}}^{\text{I}(m)}|$); (iii) $\mathcal{T}_t^{\text{S}}(a)$ does not contain change points, but the discrepancy between $(r_t, c_t)$ and $(\hat{r}_t, \hat{c}_t)$ results in assigning $\text{Ratio}^{(l)*}$ (estimated by $\widehat{\text{Ratio}}_t^{(q)}$) to the wrong interval. We remark that the losses due to (i, ii) are accounted for in Reg, while (iii) is accounted for in the CR.

### 4.4 A lower bound on competitive ratio

We complement our analysis by showing the tightness of our CR (see Appendix C.6 for proof).

**Theorem 4.5.** *Consider a fixed but arbitrary $\alpha > 1$, and set $\eta_{\min} = \alpha^{-3\nu}, \eta_{\max} = 1$ for an arbitrary $\nu \in \mathbb{Z}_{>0}$. For any online algorithm, there exist an instance for which $\eta_{\min} \leqslant r_t(a), c_t(a) \leqslant \eta_{\max}$ for all $a \in \mathcal{K}, t \in \mathcal{T}$, and that $\sum_{t=1}^{T} R_t(a_t)/opt(FA) \leqslant \Theta(1/\log_\alpha(\eta_{\max}/\eta_{\min}))$.*

## 5 Numerical Experiments

We run numerical experiments on a single-resource problem where $L = 2$, $T = 20000$ (each stationary piece has 10000 rounds), $\mathcal{K} = \{1, 2\}$, $B = 9360$ and we set $\alpha = e$ for our algorithms. The rewards and resource consumption in all rounds are uniformly distributed within a $[-0.2, +0.2]$ range from their mean values. We compare the performance of IRES-CM with Immorlica et al. (2019)'s algorithm and Zhou et al. (2008)'s algorithm. Recall that Immorlica et al. (2019) focus on an adversarial Bwk problem and achieves a CR w.r.t. a static benchmark. Zhou et al. (2008) study a full-feedback adversarial setting and achieves a CR w.r.t. a single best arm benchmark. In Figure 1, each curve represents the average cumulative reward over 10 simulations, and the shaded area around each curve marks the variance over the simulations. We provide Zhou et al. (2008)'s algorithm with extra information of $(r_t, c_t)$ before making decisions in each round, and compare the performance of algorithms with the linear program benchmark FA (dotted curves in Figure 1).

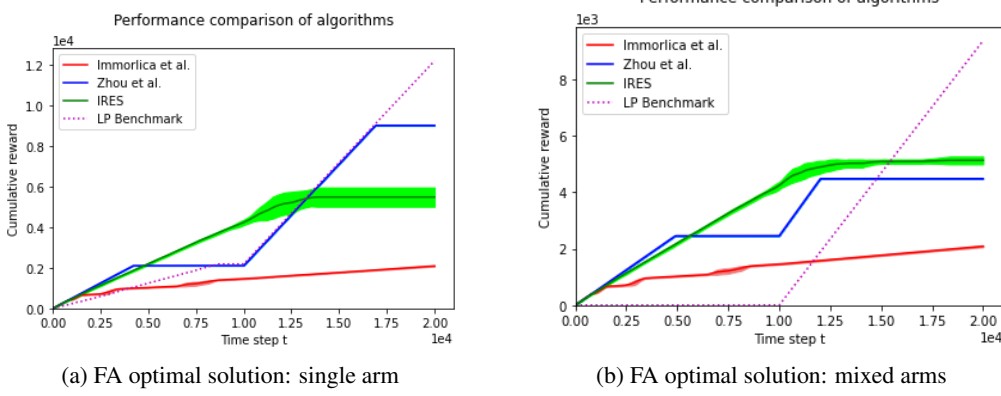

(a) FA optimal solution: single arm         (b) FA optimal solution: mixed arms

Figure 1: Performance comparison of algorithms for piecewise-stationary Bwk

In Figure 1(a), we set $r^{(1)}(1) = r^{(1)}(2) = 0.5, c^{(1)}(1) = c^{(1)}(2) = 1$ for stationary piece 1; and set $r^{(2)}(1) = 1, r^{(2)}(2) = 0.5, c^{(2)}(1) = 0.5, c^{(2)}(2) = 1$ for stationary piece 2. In Figure 1(b), we switch the values of $r^{(2)}(1)$ and $r^{(2)}(2)$. i.e., setting $r^{(2)}(1) = 0.5, r^{(2)}(2) = 1$. Observe that IRES-CM outperforms Immorlica et al. (2019)'s algorithm in both cases. This is mainly because Immorlica et al. (2019)'s algorithm is designed for a more general adversarial Bwk setting. In contrast, we utilize the extra information that $\eta_{\min} = 0.5$. Therefore, Immorlica et al. (2019)'s algorithm is significantly more conservative than IRES-CM in reserving inventories for future customers. Zhou et al. (2008)'s algorithm outperforms IRES-CM in Figure 1(a), but performs worse than IRES-CM in Figure 1(b). This is because that in Figure 1(a), the optimal solution of the benchmark FA chooses a single arm on each stationary piece, which aligns with Zhou et al. (2008)'s single best arm benchmark. Zhou et al. (2008)'s algorithm performs well with the extra information of $(r_t, c_t)$ before making decisions. In Figure 1(b), the optimal solution of the benchmark FA chooses mixed arms on the second stationary piece, where $x_2^*(1) = 0.128, x_2^*(2) = 0.872$. The numerical results are consistent with the theoretical results that Zhou et al. (2008) achieve sub-optimal rewards compared with a *best distribution over arms* benchmark, while our IRES-CM performs well. Finally, our experiments are run on a Surface Pro 7 with an i5-1035G4 processor. All results can be produced within 30 minutes.

## Acknowledgement

We would like to acknowledge the support from the Singapore Ministry of Education AcRF Tier 2 Grant (Grant number: T2EP20121-0035).

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

# A Comparing our performance guarantee with existing literature

## A.1 Comparing with adversarial BwKs

Immorlica et al. (2019) (achieving $O(d \log(T))$ competitive ratio) and Kesselheim and Singla (2020) (achieving $O(\log(d) \cdot \log(T))$ competitive ratio) study the adversarial BwKs, where the adversarial bandit feedback $(r_t(a_t), c_t(a_t))$ is revealed in each time step after pulling arm $a_t$. Their setting is more general than ours, since they require no boundedness assumption or piece-wise stationary assumption on their outcomes. Note that if our $\eta_{\min}$ can be expressed as a function of $T$, i.e., $\eta_{\min} = T^{-\beta}$ for a $\beta \in (0, 1)$, our result can extend to multiple resources and achieve a competitive ratio of $O(d \log(T))$ (see Appendix B.5). However, we highlight that their result do *not* imply our result, due to the following two reasons.

Firstly, they consider a static benchmark, dubbed FD, which is FA with the additional constraint that $x_l^* = x^*$ for all $l \in L$; while we compare our result with the dynamic benchmark FA. Specifically, Immorlica et al. (2019) prove that no algorithm can achieve a competitive ratio smaller than $T/B^2$ w.r.t. the dynamic benchmark. We further improve this lower bound to $\Theta(T/B)$ in Lemma 2.3(a) by setting $L = T/B$, and provide a matching lower bound of $\Omega(\log(T))$ comparing with opt(FA) when $\eta_{\min} > 0$. Secondly, they have more restrictive assumptions on the value ranges of $B$ and opt(FD). Specifically, Immorlica et al. (2019) assume opt(FD) $\cdot B/|\mathcal{K}| > \tilde{\Omega}(T^{7/4})$, which is strictly stronger than our Assumption 4.1.

Some research works consider more specific regimes, including Rangi et al. (2018); Castiglioni et al. (2022a); Fikioris and Tardos (2023). We highlight that all these works still compare with static benchmarks. Castiglioni et al. (2022a) focus on the regime where $B = \Omega(T)$ and achieves a competitive ratio of $T/B$. Fikioris and Tardos (2023) provides a competitive ratio depending on $(\max_t\{r_t\}/\min_t\{r_t\}, \max_t\{c_t\}/\min_t\{c_t\})$. Rangi et al. (2018) achieves a sublinear-in-$T$ regret in a different setting with no round limit (sales stop when the inventory is depleted). Therefore, their result is incomparable with ours.

Some recent papers focus on linear contextual Bwk with adversarial contextual vectors Sivakumar et al. (2022) (achieving $O(d \log(T))$ competitive ratio) or with multiple but stationary customer classes Kim et al. (2023) (achieving sublinear in $T$ regret). We highlight that contextual vectors for each arm are observable before making decision in each round, while our model only observe bandit feedback after an arm is chosen. Therefore their results do not generalize to our setting.

## A.2 Comparing with adversarial online knapsack problems with full feedback

In our Section 3, we discuss a warm-up setting where $\{R_t(a), C_t(a)\}_{a \in \mathcal{K}}$ is observable upon the arrival of the round $t$ customer, which is similar with adversarial online knapsack with full feedback (Karp et al. (1990); Mehta et al. (2007); Zhou et al. (2008)). Zhou et al. (2008) is the most closely related to our work, where they start off from an online matching problem and extends the results to the online knapsack problem. They define LB $= \min_{a,t}\{r_t(a)/c_t(a)\}$, UB $= \max_{a,t}\{r_t(a)/c_t(a)\}$ and achieves a competitive ratio of $\log(\text{UB}/\text{LB})$ w.r.t. a dynamic *best single arm* benchmark, which differs from our FA by setting $x_l(a) \in \{0, 1\}$ for each $a, l$. Our FA, on the other hand, is a *best distribution over arms* benchmark. In fact, for stationary Bwk, Badanidiyuru et al. (2018) show (see their Appendix A) that the *best single arm* benchmark is strictly weaker than *best distribution over arms*, which could noticably affect the achievable CR. It is evident that in the picewise-stationary setting, this is also the case.

Additionally, this line of works crucially require knowing $\{(R_t(a), C_t(a))\}_{a \in \mathcal{K}}$ before making decisions. By contrast, we take a different approach in algorithm design in Section 3 allows a natural generalization from full to bandit feedback shown in Section 4.

## A.3 Comparing with non-stationary bandit/full-feedback online optimization with knapsacks

In Balseiro et al. (2022); Jiang et al. (2020); Liu et al. (2022), they measure the non-stationarity of a time-varying knapsack model with a quantity called the global variation

$$\mathsf{glo} = \sum_{t=1}^{T} \mathrm{dist}\left(\sum_{t=1}^{T}(r_t, c_t)/T, (r_t, c_t)\right).$$

While dist can be any metric, to give a more concrete idea, we highlight an example form Liu et al. (2022) being

$$\text{dist}((r, c), (r', c')) = \max_{a \in \mathcal{K}} \{|r'(a) - r(a)|\} + \max_{a \in \mathcal{K}} \{|c'(a) - c(a)|\}.$$

They all have boundedness assumption on glo. In our setting (glo unbounded), their algorithms incur a linear-in-$T$ regret even when $L = 1$, and no non-trivial competitive ratio is established in their work. To see the case of $L = 1$ but glo $= \Theta(T)$, consider the case of $\mathcal{K} = \{1\}$, and we have

$$(r_1(1), c_1(1)) = \ldots = (r_{T/2}(1), c_{T/2}(1)) = (1, 0.5),$$

but

$$(r_{T/2+1}(1), c_{T/2+1}(1)) = \ldots = (r_T(1), c_T(1)) = (0.5, 1).$$

In this case, we can verify that

$$\text{dist}\left(\sum_{t=1}^{T}(r_t, c_t)/T, (r_1, c_1)\right) = \text{dist}\left(\sum_{t=1}^{T}(r_t, c_t)/T, (r_{T/2+1}, c_{T/2+1})\right) = 0.5,$$

so we have glo $= 0.5T$ despite $L = 1$.

# B    Auxiliary results

## B.1    Notation

For functions $f(x) : \mathbb{R}_{\geqslant 0} \to \mathbb{R}_{\geqslant 0}$ and $g(x) : \mathbb{R}_{\geqslant 0} \to \mathbb{R}_{\geqslant 0}$, we say $f(x)$ is $O(g(x))$ (resp. $\Omega(g(x))$) if there exist positive constants $C$ and $n$, such that for all $x \geqslant n$, $f(x) \leqslant C \cdot g(x)$ (resp. $f(x) \geqslant C \cdot g(x)$). We use $\tilde{O}(g(x))$ (resp. $\tilde{\Omega}(g(x))$) to hide logarithmic terms in $x$ other than $g(x)$.

## B.2    Concentration inequalities

**Lemma B.1** (Multiplicative Azuma-Hoeffding Inequality (Kuszmaul and Qi (2021))). $X_1, \ldots, X_n \in [0, c]$ *are real-valued random variables, and* $\{\mathcal{F}_n\}_{i=0}^{n}$ *is a filtration. Let* $\mu = \sum_{i=1}^{n} a_i$ *where* $a_i$ *are real-valued constants.*
*(i) Suppose* $\mathbb{E}[X_i | \mathcal{F}_{i-1}] \leqslant a_i$ *holds for all* $i \in \{1, \ldots, n\}$ *almost surely. Then for any* $\delta \in (0, 1)$,

$$\Pr\left[\sum_{i=1}^{n} X_i \leqslant \left(1 + \sqrt{\frac{3c}{\mu} \log\left(\frac{1}{\delta}\right)}\right)\mu\right] \geqslant 1 - \delta.$$

*(ii) Suppose* $\mathbb{E}[X_i | \mathcal{F}_{i-1}] \geqslant a_i$ *holds for all* $i \in \{1, \ldots, n\}$ *almost surely. Then for any* $\delta \in (0, 1)$,

$$\Pr\left[\sum_{i=1}^{n} X_i \geqslant \left(1 - \sqrt{\frac{2c}{\mu} \log\left(\frac{1}{\delta}\right)}\right)\mu\right] \geqslant 1 - \delta.$$

**Lemma B.2** (Lemma 2.1, Badanidiyuru et al. (2018)). *Let* $X_1, \ldots, X_N \in [0, 1]$ *be random variables. Let* $X = \sum_{i=1}^{N} X_i$ *be the sample average, and let* $\mu = \sum_{i=1}^{N} \mathbb{E}[X_i | X_1, \ldots, X_N]$. *Then, for any* $\delta \in (0, 1)$,

$$\Pr\left(|X - \mu| \leqslant \sqrt{2X \log(1/\delta)} + 4 \log(1/\delta)\right) \geqslant 1 - 3\delta.$$

## B.3    Main claim on reward-consumption ratio

Recall that $x_l^{(q)*}$ is an optimal solution to $\text{LP}(r^{(l)}, c^{(l)}, \eta_{\min} \cdot \alpha^q))$, and $x_l^*$ is an optimal solution of $\text{LP}(r^{(l)}, c^{(l)}, B_l^*/(t_l - t_{l-1}))$ which is an optimal solution of our benchmark. In the following Claim 3, we show that the round-robin technique in IRES ensures that, on each stationary piece $l$, $x_l^{(q)*}$ for at least one $q \in \{0, \ldots, M\}$ is close to $x_l^*$ in terms of both the resource consumption and the reward-consumption ratio. This leads to the important result that for all $t \in \{t_{l-1} + 1, \ldots, t_l\}$, $m_t \in \{m_l^* - 1, m_l^*\}$.

**Claim 3.** On stationary piece $l \in \mathcal{L}$, there exists $q_l^* \in \{0, 1, \ldots, M\}$ that satisfies both

$$\begin{cases} \eta_{\min} \cdot \alpha^{q_l^* - 1} < \sum_{a \in \mathcal{K}} c^{(l)}(a) x_l^*(a) \leqslant \sum_{a \in \mathcal{K}} c^{(l)}(a) x_l^{(q_l^*)*}(a) \leqslant \eta_{\min} \cdot \alpha^{q_l^*} & q_l^* > 0 \\ 0 < \sum_{a \in \mathcal{K}} c^{(l)}(a) x_l^*(a) \leqslant \sum_{a \in \mathcal{K}} c^{(l)}(a) x_l^{(q_l^*)*}(a) \leqslant \eta_{\min} \cdot \alpha^{q_l^*} & q_l^* = 0 \end{cases}, \quad (7)$$

and

$$\frac{\sum_{a \in \mathcal{K}} r^{(l)}(a) x_l^{(q_l^*)*}(a)}{\sum_{a \in \mathcal{K}} c^{(l)}(a) x_l^{(q_l^*)*}(a)} \leqslant \frac{\sum_{a \in \mathcal{K}} r^{(l)}(a) x_l^*(a)}{\sum_{a \in \mathcal{K}} c^{(l)}(a) x_l^*(a)} \leqslant \alpha \cdot \frac{\sum_{a \in \mathcal{K}} r^{(l)}(a) x_l^{(q_l^*)*}(a)}{\sum_{a \in \mathcal{K}} c^{(l)}(a) x_l^{(q_l^*)*}(a)}. \quad (8)$$

*Proof of Claim 3.* The existence of $q_l^*$ satisfying (7) is evident, since $\eta_{\min} \cdot \alpha^0 = \eta_{\min}$ and $\eta_{\min} \cdot \alpha^M = \eta_{\max}$. To show that $q_l^*$ also satisfy (8), we define

$$d^{(l)} = \frac{\sum_{a \in \mathcal{K}} c^{(l)}(a) x_l^{(q_l^*)*}(a)}{\sum_{a \in \mathcal{K}} c^{(l)}(a) x_l^*(a)}, \quad (9)$$

and claim that

$$\sum_{a \in \mathcal{K}} r^{(l)}(a) x_l^*(a) \leqslant \sum_{a \in \mathcal{K}} r^{(l)}(a) x_l^{(q_l^*)*}(a) \leqslant d^{(l)} \sum_{a \in \mathcal{K}} r^{(l)}(a) x_l^*(a). \quad (10)$$

The first inequality in (10) holds since $B_l^*/(t_l - t_{l-1}) = \sum_{a \in \mathcal{K}} c^{(l)}(a) x_l^*(a) \leqslant \eta_{\min} \cdot \alpha^{q_l^*}$. Therefore, the resource constraint in $\mathrm{LP}(r^{(l)}, c^{(l)}, B_l^*/(t_l - t_{l-1}))$ is tighter than $\mathrm{LP}(r^{(l)}, c^{(l)}, \eta_{\min} \cdot \alpha^{q_l^*})$. We prove the second inequality in (10) by contradiction. Suppose $\sum_{a \in \mathcal{K}} r^{(l)}(a) x_l^{(q_l^*)*}(a) > d^{(l)} \sum_{a \in \mathcal{K}} r^{(l)}(a) x_l^*(a)$. Then we set $x_l = x_l^{(q_l^*)*}/d^{(l)}$ and have

$$\sum_{a \in \mathcal{K}} c^{(l)}(a) x_l(a) = \frac{1}{d^{(l)}} \sum_{a \in \mathcal{K}} c^{(l)}(a) x_l^{(q_l^*)*} = \sum_{a \in \mathcal{K}} c^{(l)}(a) x_l^*(a) = B_l^*/(t_l - t_{l-1}).$$

In this case, $x_l$ is a feasible solution to $\mathrm{LP}(r^{(l)}, c^{(l)}, B_l^*/(t_l - t_{l-1}))$ and we have

$$\sum_{a \in \mathcal{K}} r^{(l)}(a) x_l(a) = \frac{1}{d^{(l)}} \sum_{a \in \mathcal{K}} r^{(l)}(a) x_l^{(q_l^*)*} > \sum_{a \in \mathcal{K}} r^{(l)}(a) x_l^*(a),$$

contradicting the fact that $x_l^*$ is an optimal solution of $\mathrm{LP}(r^{(l)}, c^{(l)}, B_l^*/(t_l - t_{l-1}))$. Therefore, combining (9) and (10), we establish

$$\frac{\sum_{a \in \mathcal{K}} r^{(l)}(a) x_l^{(q_l^*)*}(a)}{\sum_{a \in \mathcal{K}} c^{(l)}(a) x_l^{(q_l^*)*}(a)} \leqslant \frac{\sum_{a \in \mathcal{K}} r^{(l)}(a) x_l^*(a)}{\sum_{a \in \mathcal{K}} c^{(l)}(a) x_l^*(a)} \leqslant d^{(l)} \cdot \frac{\sum_{a \in \mathcal{K}} r^{(l)}(a) x_l^{(q_l^*)*}(a)}{\sum_{a \in \mathcal{K}} c^{(l)}(a) x_l^{(q_l^*)*}(a)}.$$

To establish (8), it suffices to show $d^{(l)} \leqslant \alpha$. By (7) and (9), it is evident that $d^{(l)} \leqslant \alpha$ when $q_l^* > 0$. If $q_l^* = 0$, then constraints $\sum_{a \in \mathcal{K}} x_l^*(a) \leqslant 1$ and $\sum_{a \in \mathcal{K}} x_l^{(q_l^*)*}(a) \leqslant 1$ are not tight in both $\mathrm{LP}(r^{(l)}, c^{(l)}, B_l^*/(t_l - t_{l-1}))$ and $\mathrm{LP}(r^{(l)}, c^{(l)}, \eta_{\min})$. In this case, both LPs are knapsack problems and have closed-form solutions of

$$x_l^*(a) = \begin{cases} \frac{B_l^*}{(t_l - t_{l-1}) c^{(l)}(a)} & a = \arg\max_{a \in \mathcal{K}} \left\{ \frac{r^{(l)}(a)}{c^{(l)}(a)} \right\} \\ 0 & \text{otherwise} \end{cases}, \quad x_l^{(q_l^*)*}(a) = \begin{cases} \frac{\eta_{\min}}{c^{(l)}(a)} & a = \arg\max_{a \in \mathcal{K}} \left\{ \frac{r^{(l)}(a)}{c^{(l)}(a)} \right\} \\ 0 & \text{otherwise} \end{cases}.$$

Therefore, we have

$$\frac{\sum_{a \in \mathcal{K}} r^{(l)}(a) x_l^{(q_l^*)*}(a)}{\sum_{a \in \mathcal{K}} c^{(l)}(a) x_l^{(q_l^*)*}(a)} = \frac{\sum_{a \in \mathcal{K}} r^{(l)}(a) x_l^*(a)}{\sum_{a \in \mathcal{K}} c^{(l)}(a) x_l^*(a)},$$

which shows that $d^{(l)} = 1 < \alpha$ when $q_l^* = 0$. $\qquad \square$

## B.4 Decomposing ratio of REW to opt(FA) in deterministic setting

Recall

$$\tilde{\mathcal{T}}^{(m)} = \left\{ \tau^{(m)}(n) : \sum_{s=1}^{n} \sum_{a \in \mathcal{K}} c_{\tau^{(m)}(s)}(a) x_{\tau^{(m)}(s)}(a) \leqslant \frac{B}{2M} \right\}$$

and $\mathcal{J}_l = \{t_{l-1} + 1, \ldots, t_l\}$. Then for the reward achieved by IRES, we have

$$
\begin{aligned}
\text{REW} &= \sum_{t \in \mathcal{T}} \sum_{a \in \mathcal{K}} r_t(a) x_t(a) \\
&= \sum_{l \in \mathcal{L}} \sum_{t \in \mathcal{T} \cap \mathcal{J}_l} \sum_{a \in \mathcal{K}} r_t(a) x_t(a) \\
&\geqslant \sum_{l \in \mathcal{L}} \sum_{t \in \left( \bigcup_{n=-M}^{M-1} \tilde{\mathcal{T}}^{(n)} \right) \cap \mathcal{J}_l} \sum_{a \in \mathcal{K}} r_t(a) x_t(a) \\
&\geqslant \sum_{m=-M}^{M-1} \sum_{l \in \mathcal{L}} \mathbf{1}(m_l^* = m) \sum_{t \in \left( \bigcup_{n=-M}^{M-1} \tilde{\mathcal{T}}^{(n)} \right) \cap \mathcal{J}_l} \sum_{a \in \mathcal{K}} r_t(a) x_t(a) \\
&= \sum_{m=-M}^{M-1} \text{REW}^{(m)} \\
&\geqslant \frac{1}{3} \sum_{m=-M}^{M-1} \sum_{w=\max\{m-1,-M\}}^{\min\{m+1,M-1\}} \sum_{l \in \mathcal{L}} \mathbf{1}(m_l^* = w) \sum_{t \in \left( \bigcup_{n=-M}^{M-1} \tilde{\mathcal{T}}^{(n)} \right) \cap \mathcal{J}_l} \sum_{a \in \mathcal{K}} r_t(a) x_t(a). \quad (11)
\end{aligned}
$$

Inequality (11) holds since by summing over $w \in \{\max\{m-1, -M\}, m, \min\{m+1, M-1\}\}$ for each $m \in \{-M, \ldots, M-1\}$, we repeat the sum for at most 3 times.

For the reward achieved by FA, we have

$$
\begin{aligned}
\text{opt(FA)} &= \sum_{l \in \mathcal{L}} \frac{\sum_{a \in \mathcal{K}} r^{(l)}(a) x_l^*(a)}{\sum_{a \in \mathcal{K}} c^{(l)}(a) x_l^*(a)} \cdot B_l^* \quad (12) \\
&= \sum_{m=-M}^{M-1} \sum_{l \in \mathcal{L}} \mathbf{1}(m_l^* = m) \cdot \frac{\sum_{a \in \mathcal{K}} r^{(l)}(a) x_l^*(a)}{\sum_{a \in \mathcal{K}} c^{(l)}(a) x_l^*(a)} \cdot B_l^* \\
&= \sum_{m=-M}^{M-1} \text{opt(FA)}^{(m)}.
\end{aligned}
$$

Inequality (12) follows from (2).

Therefore, the ratio of the reward achieved by IRES to opt(FA) can be decomposed as:

$$
\begin{aligned}
&\frac{\sum_{t \in \mathcal{T}} \sum_{a \in \mathcal{K}} r_t(a) x_t(a)}{\text{opt(FA)}} \\
&\geqslant \frac{\sum_{m=-M}^{M-1} \text{REW}^{(m)}}{\sum_{m=-M}^{M-1} \text{opt(FA)}^{(m)}} \\
&\geqslant \frac{\frac{1}{3} \sum_{m=-M}^{M-1} \sum_{w=m-1}^{\min\{m+1,M-1\}} \sum_{l \in \mathcal{L}} \mathbf{1}(m_l^* = w) \sum_{t \in \left( \bigcup_{n=-M}^{M-1} \tilde{\mathcal{T}}^{(n)} \right) \cap \mathcal{J}_l} \sum_{a \in \mathcal{K}} r_t(a) x_t(a)}{\sum_{m=-M}^{M-1} \sum_{l \in \mathcal{L}} \mathbf{1}(m_l^* = m) \cdot \frac{\sum_{a \in \mathcal{K}} r^{(l)}(a) x_l^*(a)}{\sum_{a \in \mathcal{K}} c^{(l)}(a) x_l^*(a)} \cdot B_l^*}. \quad (13)
\end{aligned}
$$

Then, to prove Theorem 3.1, it suffices to show

$$\frac{\sum_{m=-M}^{M-1} \text{REW}^{(m)}}{\sum_{m=-M}^{M-1} \text{opt(FA)}^{(m)}} \geqslant (13) \geqslant \frac{1 - (2M+1)/B}{6\alpha^2 M}. \quad (14)$$

To prove (14), we focus on showing in Claims 1 and 2.

$$\frac{\text{REW}^{(m)}}{\text{opt(FA)}^{(m)}} \geqslant \frac{\frac{1}{3} \cdot \sum_{w=m-1}^{\min\{m+1,M-1\}} \sum_{l \in \mathcal{L}} \mathbf{1}(m_l^* = w) \sum_{t \in \bigcup_{m=-M}^{M-1} \tilde{\mathcal{T}}^{(m)} \cap \mathcal{J}_l} \sum_{a \in \mathcal{K}} r_t(a) x_t(a)}{\sum_{l \in \mathcal{L}} \mathbf{1}(m_l^* = m) \cdot \frac{\sum_{a \in \mathcal{K}} r^{(l)}(a) x_l^*(a)}{\sum_{a \in \mathcal{K}} c^{(l)}(a) x_l^*(a)} \cdot B_l^*}$$

$$\geqslant \frac{1 - (2M+1)/B}{6\alpha^2 M}$$

for each $m \in \{-M, \ldots, M-1\}$.

## B.5 Extending results to multiple resources

Our results can be readily extend to the multiple-resource case, with $|\mathcal{I}| = d$ resources indexed by $i \in \mathcal{I}$. An upper bound for a multi-resource allocation problem (corresponding to FA in the single-resource setting) can be formulated as:

$$\text{FA}^{\text{MUL}} := \max \sum_{l=1}^{L} (t_l - t_{l-1}) \sum_{a \in \mathcal{K}} r^{(l)}(a) x_l(a)$$

$$\text{s.t.} \sum_{l=1}^{L} (t_l - t_{l-1}) \sum_{a \in \mathcal{K}} c_i^{(l)}(a) x_l(a) \leqslant B \qquad \forall i \in \mathcal{I}$$

$$\sum_{a \in \mathcal{K}} x_l(a) \leqslant 1 \qquad \forall l = 1, \ldots, L$$

$$x_l(a) \geqslant 0 \qquad \forall a \in \mathcal{K},\ l = 1, \ldots, L.$$

which is evidently upper bounded by the following LP:

$$\text{FA}^{\text{MUL-U}} := \max \sum_{l=1}^{L} (t_l - t_{l-1}) \sum_{a \in \mathcal{K}} r^{(l)}(a) x_l(a)$$

$$\text{s.t.} \sum_{l=1}^{L} (t_l - t_{l-1}) \sum_{a \in \mathcal{K}} \left( \sum_{i \in \mathcal{I}} c_i^{(l)}(a) \right) x_l(a) \leqslant |\mathcal{I}| B$$

$$\sum_{a \in \mathcal{K}} x_l(a) \leqslant 1 \qquad \forall l = 1, \ldots, L$$

$$x_l(a) \geqslant 0 \qquad \forall a \in \mathcal{K},\ l = 1, \ldots, L.$$

It is evident that the LP below achieves a reward of at least $1/d$ fraction of opt(FA$^{\text{MUL-U}}$).

$$\text{FA}^{\text{MUL-F}} := \max \sum_{l=1}^{L} (t_l - t_{l-1}) \sum_{a \in \mathcal{K}} r^{(l)}(a) x_l(a)$$

$$\text{s.t.} \sum_{l=1}^{L} (t_l - t_{l-1}) \sum_{a \in \mathcal{K}} \left( \sum_{i \in \mathcal{I}} c_i^{(l)}(a) \right) x_l(a) \leqslant B$$

$$\sum_{a \in \mathcal{K}} x_l(a) \leqslant 1 \qquad \forall l = 1, \ldots, L$$

$$x_l(a) \geqslant 0 \qquad \forall a \in \mathcal{K},\ l = 1, \ldots, L.$$

Therefore, FA$^{\text{MUL}}$ is transformed into a single-resource allocation problem FA$^{\text{MUL-F}}$, with an extra multiplicative factor $d$ in the competitive ratio.

## B.6 Core lemma on reward-consumption ratio in general setting

Recall that set $\mathcal{T}_t^{\text{S}}(a) = \{\tau \in \{t - s_t, \ldots, t - 1\} : a_\tau = a\}$ where $s_t = \arg\max_s \{\sum_{\tau=t-s}^{t-1} \mathbf{1}(a_\tau = a) = N\}$ consists of the most recent $N$ rounds where arm $a$ is sampled by exploration; and $\sigma_t(a) = \min\{s : s \in \mathcal{T}_t^{\text{S}}(a)\}$ is the 1-st element in $\mathcal{T}_t^{\text{S}}(a)$. Recall that

$$\hat{c}_t(a) = \frac{\sum_{s \in \mathcal{T}_t^{\text{S}}(a)} C_s(a)}{N}, \quad \hat{r}_t(a) = \frac{\sum_{s \in \mathcal{T}_t^{\text{S}}(a)} R_s(a)}{N}$$

are the average resource consumption and the reward earned for the most recent $N$ pulls of arm $a$. Additionally, recall that we define $\hat{x}_t^{(q)} = \{\hat{x}_t^{(q)}(a)\}_{a \in \mathcal{K}}$ as a solution to LP$(\hat{r}_t, \hat{c}_t, \eta_{\min} \cdot \alpha^q)$ ($\hat{x}_t^{(q)*}$ being the optimal solution) for $q = 0, 1, \ldots, M-1$.

We further define

$$\bar{c}_t(a) = \frac{\sum_{s \in \mathcal{T}_t^S(a)} c_s(a)}{N}, \quad \bar{r}_t(a) = \frac{\sum_{s \in \mathcal{T}_t^S(a)} r_s(a)}{N}$$

as the average *mean* resource consumption and the reward earned for the most recent $N$ pulls of arm $a$. Note that when condition (6)

$$\{(r_s(a), c_s(a))\}_{a \in \mathcal{K}} = \{(r_{\sigma_t(a)}(a), c_{\sigma_t(a)}(a))\}_{a \in \mathcal{K}}, \; \forall s \in \{\sigma_t, \ldots, t\}$$

is satisfied for all $a \in \mathcal{K}$, then rounds $s \in \{\sigma_t, \ldots, t\}$ are on the same stationary piece and we have $\bar{c}_t(a) = c_t(a)$, $\bar{r}_t(a) = r_t(a)$. We let $\bar{x}_t^{(q)} = \{\bar{x}_t^{(q)}(a)\}_{a \in \mathcal{K}}$ be a solution to LP$(r_t, c_t, \eta_{\min} \cdot \alpha^q)$ ($\bar{x}_t^{(q)*}$ being the optimal solution) for $q = 0, 1, \ldots, M-1$. By Claim 3, we know that there exists $q_l^* \in \{0, 1, \ldots, M\}$ such that for all $t \in \mathcal{J}_l$,

$$\begin{cases} \eta_{\min} \cdot \alpha^{q_l^* - 1} < \sum_{a \in \mathcal{K}} c^{(l)}(a) x_l^*(a) \leqslant \sum_{a \in \mathcal{K}} c_t(a) \bar{x}_t^{(q_l^*)*}(a) \leqslant \eta_{\min} \cdot \alpha^{q_l^*} & q_l^* > 0 \\ 0 < \sum_{a \in \mathcal{K}} c^{(l)}(a) x_l^*(a) \leqslant \sum_{a \in \mathcal{K}} c_t(a) \bar{x}_t^{(q_l^*)*}(a) \leqslant \eta_{\min} \cdot \alpha^{q_l^*} & q_l^* = 0 \end{cases}. \quad (15)$$

We show in the following Lemma B.3 that when condition (6) is satisfied for all $a \in \mathcal{K}$, our decisions have several nice properties which facilitate our proofs.

**Lemma B.3.** *Fix an arbitrary $\alpha \in (0, 1]$. For any $t \geqslant \sigma_t(a)$, if*

$$\{(r_s(a), c_s(a))\}_{a \in \mathcal{K}} = \{(r_{\sigma_t(a)}(a), c_{\sigma_t(a)}(a))\}_{a \in \mathcal{K}}, \; \forall s \in \{\sigma_t, \ldots, t\},$$

*then with probability at least $1 - 2|\mathcal{K}|\delta$, then all the following inequalities are satisfied for all $t \in \mathcal{J}_l$:*

$$\frac{1}{\sqrt{\alpha}} \cdot \frac{\sum_{a \in \mathcal{K}} \hat{r}_t(a) \hat{x}_t^{(q_l^*)*}(a)}{\sum_{a \in \mathcal{K}} \hat{c}_t(a) \hat{x}_t^{(q_l^*)*}(a)} \leqslant \frac{\sum_{a \in \mathcal{K}} r_t(a) \hat{x}_t^{(q_l^*)*}(a)}{\sum_{a \in \mathcal{K}} c_t(a) \hat{x}_t^{(q_l^*)*}(a)} \leqslant \sqrt{\alpha} \cdot \frac{\sum_{a \in \mathcal{K}} \hat{r}_t(a) \hat{x}_t^{(q_l^*)*}(a)}{\sum_{a \in \mathcal{K}} \hat{c}_t(a) \hat{x}_t^{(q_l^*)*}(a)}, \quad (16)$$

$$\frac{1}{\alpha\sqrt{\alpha}} \cdot \sum_{a \in \mathcal{K}} r_t(a) \hat{x}_t^{(q_l^*)*}(a) \leqslant \sum_{a \in \mathcal{K}} r^{(l)}(a) x_l^*(a) \leqslant \alpha \cdot \sum_{a \in \mathcal{K}} r_t(a) \hat{x}_t^{(q_l^*)*}(a), \quad (17)$$

$$\frac{1}{\alpha\sqrt{\alpha}} \cdot \frac{\sum_{a \in \mathcal{K}} \hat{r}_t(a) \hat{x}_t^{(q_l^*)*}(a)}{\sum_{a \in \mathcal{K}} \hat{c}_t(a) \hat{x}_t^{(q_l^*)*}(a)} \leqslant \frac{\sum_{a \in \mathcal{K}} r^{(l)}(a) x_l^*(a)}{\sum_{a \in \mathcal{K}} c^{(l)}(a) x_l^*(a)} \leqslant \alpha\sqrt{\alpha} \cdot \frac{\sum_{a \in \mathcal{K}} \hat{r}_t(a) \hat{x}_t^{(q_l^*)*}(a)}{\sum_{a \in \mathcal{K}} \hat{c}_t(a) \hat{x}_t^{(q_l^*)*}(a)} \quad \text{if } q_l^* = 0 \quad (18)$$

*Proof of Lemma B.3.* We define

$$\epsilon = \sqrt{\frac{3}{N \cdot \eta_{\min}} \log\left(\frac{2}{\delta}\right)} \geqslant \max\left\{\sqrt{\frac{3}{N \cdot \eta_{\max}} \log\left(\frac{2}{\delta}\right)}, \sqrt{\frac{2}{N \cdot \eta_{\min}} \log\left(\frac{2}{\delta}\right)}\right\}.$$

Then by the multiplicative Azuma-Hoeffding inequality (Lemma B.1), for any $a \in \mathcal{K}$ we have

$$\Pr[(1-\epsilon)c_t(a) \leqslant \hat{c}_t(a) \leqslant (1+\epsilon)c_t(a)] \geqslant 1 - \delta \quad (19)$$
$$\Pr[(1-\epsilon)r_t(a) \leqslant \hat{r}_t(a) \leqslant (1+\epsilon)r_t(a)] \geqslant 1 - \delta. \quad (20)$$

The above probability bounds (19) and (20) hold since

$$N \cdot \eta_{\min} \leqslant \sum_{s \in \mathcal{T}_t^S(a)} c_t(a), \quad \sum_{s \in \mathcal{T}_t^S(a)} r_t(a) \leqslant N \cdot \eta_{\max}.$$

We set $(1 - 3\epsilon)^2 = 1/\alpha$, and therefore

$$N = \frac{27 \log(2/\delta)}{(1 - 1/\sqrt{\alpha})^2 \cdot \eta_{\min}}.$$

Our following discussion is conditioned on the good event that both (19) and (20) hold for all $a \in \mathcal{K}$. This good event holds with probability $1 - 2|\mathcal{K}|\delta$.

**Validating (16).** Given that (19) and (20) hold, we have

$$\frac{1-\epsilon}{1+\epsilon} \cdot \frac{\sum_{a\in\mathcal{K}} \hat{r}_t(a)\hat{x}_t^{(q_l^*)*}(a)}{\sum_{a\in\mathcal{K}} \hat{c}_t(a)\hat{x}_t^{(q_l^*)*}(a)} \leqslant \frac{\sum_{a\in\mathcal{K}} r_t(a)\hat{x}_t^{(q_l^*)*}(a)}{\sum_{a\in\mathcal{K}} c_t(a)\hat{x}_t^{(q_l^*)*}(a)} \leqslant \frac{1+\epsilon}{1-\epsilon} \cdot \frac{\sum_{a\in\mathcal{K}} \hat{r}_t(a)\hat{x}_t^{(q_l^*)*}(a)}{\sum_{a\in\mathcal{K}} \hat{c}_t(a)\hat{x}_t^{(q_l^*)*}(a)}$$

$$\Rightarrow (1-2\epsilon) \cdot \frac{\sum_{a\in\mathcal{K}} \hat{r}_t(a)\hat{x}_t^{(q_l^*)*}(a)}{\sum_{a\in\mathcal{K}} \hat{c}_t(a)\hat{x}_t^{(q_l^*)*}(a)} \leqslant \frac{\sum_{a\in\mathcal{K}} r_t(a)\hat{x}_t^{(q_l^*)*}(a)}{\sum_{a\in\mathcal{K}} c_t(a)\hat{x}_t^{(q_l^*)*}(a)} \leqslant \frac{1}{1-2\epsilon} \cdot \frac{\sum_{a\in\mathcal{K}} \hat{r}_t(a)\hat{x}_t^{(q_l^*)*}(a)}{\sum_{a\in\mathcal{K}} \hat{c}_t(a)\hat{x}_t^{(q_l^*)*}(a)}. \tag{21}$$

Since $1 - 2\epsilon \geqslant 1 - 3\epsilon = 1/\alpha$, (21) indicates (16).

**Validating (17).** It is evident that by letting $\hat{x}_t^{(q_l^*)}(a) = \bar{x}_t^{(q_l^*)*}(a)/(1+\epsilon)$, we have

$$\begin{aligned}
\sum_{a\in\mathcal{K}} \hat{c}_t(a)\hat{x}_t^{(q_l^*)}(a) &= \sum_{a\in\mathcal{K}} \hat{c}_t(a) \cdot \frac{\bar{x}_t^{(q_l^*)*}(a)}{1+\epsilon} \\
&\leqslant \sum_{a\in\mathcal{K}} (1+\epsilon)c_t(a) \cdot \frac{\bar{x}_t^{(q_l^*)*}(a)}{1+\epsilon} \\
&= \sum_{a\in\mathcal{K}} c_t(a)\bar{x}_t^{(q_l^*)*}(a) \\
&\leqslant \eta_{\min} \cdot \alpha^q.
\end{aligned} \tag{22}$$

Inequality (22) follows from inequalities (19). Since $\hat{x}_t^{(q_l^*)}(a) = \bar{x}_t^{(q_l^*)*}(a)/(1+\epsilon)$ is a feasible solution to $\text{LP}(\hat{r}_t, \hat{c}_t, \eta_{\min} \cdot \alpha^{q_l^*})$, we have

$$\begin{aligned}
\sum_{a\in\mathcal{K}} r_t(a)\hat{x}_t^{(q_l^*)*}(a) &\geqslant \sum_{a\in\mathcal{K}} \hat{r}_t(a) \cdot \frac{\hat{x}_t^{(q_l^*)*}(a)}{1+\epsilon} \\
&\geqslant \sum_{a\in\mathcal{K}} \hat{r}_t(a) \cdot \frac{\bar{x}_t^{(q_l^*)*}(a)}{(1+\epsilon)^2} \\
&\geqslant (1-2\epsilon) \sum_{a\in\mathcal{K}} \hat{r}_t(a)\bar{x}_t^{(q_l^*)*}(a) \\
&\geqslant (1-3\epsilon) \sum_{a\in\mathcal{K}} r_t(a)\bar{x}_t^{(q_l^*)*}(a).
\end{aligned} \tag{23}$$

Similarly, by letting $\bar{x}_t^{(q_l^*)}(a) = (1-\epsilon)\hat{x}_t^{(q_l^*)*}(a)$, we have

$$\begin{aligned}
\sum_{a\in\mathcal{K}} c_t(a)\bar{x}_t^{(q_l^*)}(a) &= \sum_{a\in\mathcal{K}} c_t(a) \cdot (1-\epsilon)\hat{x}_t^{(q_l^*)*}(a) \\
&\leqslant \sum_{a\in\mathcal{K}} \frac{\hat{c}_t(a)}{1-\epsilon} \cdot (1-\epsilon)\hat{x}_t^{(q_l^*)*}(a) \\
&= \sum_{a\in\mathcal{K}} \hat{c}_t(a)\hat{x}_t^{(q_l^*)*}(a) \\
&\leqslant \eta_{\min} \cdot \alpha^q.
\end{aligned}$$

Since $\bar{x}_t^{(q_l^*)}(a) = (1-\epsilon)\hat{x}_t^{(q_l^*)*}(a)$ is a feasible solution to $\text{LP}(r_t, c_t, \eta_{\min} \cdot \alpha^{q_l^*})$, we have

$$\sum_{a\in\mathcal{K}} \hat{r}_t(a)\hat{x}_t^{(q_l^*)*}(a) \leqslant (1+\epsilon) \cdot \sum_{a\in\mathcal{K}} r_t(a)\hat{x}_t^{(q_l^*)*}(a) \leqslant \sum_{a\in\mathcal{K}} r_t(a) \cdot \frac{\bar{x}_t^{(q_l^*)*}(a)}{1-\epsilon} \leqslant \frac{1}{1-2\epsilon} \sum_{a\in\mathcal{K}} r_t(a)\bar{x}_t^{(q_l^*)*}(a). \tag{24}$$

Since $(1-3\epsilon)^2 = 1/\alpha$, putting (23) and (24) together, we have

$$\frac{1}{\sqrt{\alpha}} \cdot \sum_{a\in\mathcal{K}} r_t(a)\bar{x}_t^{(q_l^*)*}(a) \leqslant \sum_{a\in\mathcal{K}} r_t(a)\hat{x}_t^{(q_l^*)*}(a) \leqslant \sqrt{\alpha} \cdot \sum_{a\in\mathcal{K}} r_t(a)\bar{x}_t^{(q_l^*)*}(a). \tag{25}$$

By (10) in Claim 3, we have

$$\frac{1}{\alpha} \cdot \sum_{a \in \mathcal{K}} r_t(a) \bar{x}_t^{(q_l^*)*}(a) \leqslant \sum_{a \in \mathcal{K}} r^{(l)}(a) x_l^*(a) \leqslant \sum_{a \in \mathcal{K}} r_t(a) \bar{x}_t^{(q_l^*)*}(a). \tag{26}$$

Given (25) and (26), we prove (17).

**Validating (18).** For $l \in \mathcal{L}$ such that $q_l^* = 0$, we have $\sum_{a \in \mathcal{K}} c_t(a) \bar{x}_t^{(q_l^*)*}(a) = \eta_{\min}$. Given (19), for all $t \in \mathcal{J}_l$,

$$\frac{1}{\sqrt{\alpha}} \cdot \sum_{a \in \mathcal{K}} c_t(a) \bar{x}_t^{(q_l^*)*}(a) = \frac{1}{\sqrt{\alpha}} \cdot \eta_{\min} \leqslant \sum_{a \in \mathcal{K}} \hat{c}_t(a) \hat{x}_t^{(q_l^*)*}(a) \leqslant \eta_{\min} = \sum_{a \in \mathcal{K}} c_t(a) \bar{x}_t^{(q_l^*)*}(a). \tag{27}$$

Putting (23) and (24) together, we have

$$\frac{1}{\sqrt{\alpha}} \cdot \sum_{a \in \mathcal{K}} r_t(a) \bar{x}_t^{(q_l^*)*}(a) \leqslant \sum_{a \in \mathcal{K}} \hat{r}_t(a) \hat{x}_t^{(q_l^*)*}(a) \leqslant \sqrt{\alpha} \cdot \sum_{a \in \mathcal{K}} r_t(a) \bar{x}_t^{(q_l^*)*}(a). \tag{28}$$

Inequality (27) and (28) gives

$$\frac{1}{\alpha \sqrt{\alpha}} \cdot \frac{\sum_{a \in \mathcal{K}} \hat{r}_t(a) \hat{x}_t^{(q_l^*)*}(a)}{\sum_{a \in \mathcal{K}} \hat{c}_t(a) \hat{x}_t^{(q_l^*)*}(a)} \leqslant \frac{\sum_{a \in \mathcal{K}} r_t(a) \bar{x}_t^{(q_l^*)*}(a)}{\sum_{a \in \mathcal{K}} c_t(a) \bar{x}_t^{(q_l^*)*}(a)} \leqslant \sqrt{\alpha} \cdot \frac{\sum_{a \in \mathcal{K}} \hat{r}_t(a) \hat{x}_t^{(q_l^*)*}(a)}{\sum_{a \in \mathcal{K}} \hat{c}_t(a) \hat{x}_t^{(q_l^*)*}(a)}. \tag{29}$$

Recall from (8) in Claim 3, for all $t \in \mathcal{J}_l$ we have

$$\frac{\sum_{a \in \mathcal{K}} r_t(a) \bar{x}_t^{(q_l^*)*}(a)}{\sum_{a \in \mathcal{K}} c_t(a) \bar{x}_t^{(q_l^*)*}(a)} \leqslant \frac{\sum_{a \in \mathcal{K}} r^{(l)}(a) x_l^*(a)}{\sum_{a \in \mathcal{K}} c^{(l)}(a) x_l^*(a)} \leqslant \alpha \cdot \frac{\sum_{a \in \mathcal{K}} r_t(a) \bar{x}_t^{(q_l^*)*}(a)}{\sum_{a \in \mathcal{K}} c_t(a) \bar{x}_t^{(q_l^*)*}(a)}. \tag{30}$$

Putting (27) and (30) together, we establish (18). $\qquad \square$

## B.7 Bounding exploration rounds and failed exploitation rounds

To upper bound $|\mathcal{T}_T^{\mathrm{R}} \bigcup (\bigcup_{m=-M-2}^{M} \widehat{\mathcal{T}}^{\mathrm{I}(m)})|$, it suffices to upper bound $|\mathcal{T}_T^{\mathrm{R}}|$ (forthcoming Claim 4) and $|\bigcup_{m=-M-2}^{M+1} \widehat{\mathcal{T}}^{\mathrm{I}(m)}|$ (forthcoming Claim 5).

**Claim 4.** For any $\delta \in (0, 1)$, $|\mathcal{T}_T^{\mathrm{R}}| \leqslant T N \gamma_T + N \sqrt{3 T \gamma_T \log \left( \frac{2}{\delta} \right)}$ with probability at least $1 - \delta$.

*Proof of Claim 4.* Define the event $E_t^{\mathrm{R}} = \{\text{Conduct exploration in round } t\} = E_t^{\mathrm{R}(1)} \bigcup E_t^{\mathrm{R}(2)}$, where $E_t^{\mathrm{R}(1)} = \{\exists a \in \mathcal{K} \text{ s.t. } t = \sigma_t(a)\}$ and $E_t^{\mathrm{R}(2)} = \{\exists a \in \mathcal{K} \text{ s.t. } t \in \mathcal{T}_t^{\mathrm{S}}(a) \backslash \sigma_t(a)\}$. We define random variables $Y_t^{\mathrm{R}} = \mathbf{1}(E_t^{\mathrm{R}})$, $Y_t^{\mathrm{R}(1)} = \mathbf{1}(E_t^{\mathrm{R}(1)})$ and $Y_t^{\mathrm{R}(2)} = \mathbf{1}(E_t^{\mathrm{R}(2)})$. Define event $E_t^{\mathrm{I}} = \{\text{Conduct exploitation in round } t\} = E_t^{\mathrm{I}(1)} \bigcup E_t^{\mathrm{I}(2)}$, where $E_t^{\mathrm{I}(1)} = \{q_t = 0\}$ and $E_t^{\mathrm{I}(2)} = \{q_t > 0\}$. We define random variables $Y_t^{\mathrm{I}} = \mathbf{1}(E_t^{\mathrm{I}})$, $Y_t^{\mathrm{I}(1)} = \mathbf{1}(E_t^{\mathrm{I}(1)})$ and $Y_t^{\mathrm{I}(2)} = \mathbf{1}(E_t^{\mathrm{I}(2)})$.

We further define random variables $Z_t$, where $Z_t = Y_t^{\mathrm{R}(1)} \sim \mathrm{Bern}(\gamma_t)$ for all $t$ such that $Y_t^{\mathrm{R}(1)} = 1$ or $Y_t^{\mathrm{I}(1)} = 1$ and $Z_t \sim \mathrm{Bern}(\gamma_t)$ otherwise. It is evident that in each round $t$, $Z_t$ follows a Bernoulli distribution with mean $\gamma_t$ and $Z_t \geqslant Y_t^{\mathrm{R}(1)}$. Therefore, the total rounds of forced exploration over the planning horizon can be upper bounded as

$$\sum_{t=1}^{T} Y_t^{\mathrm{R}} = \sum_{t=1}^{T} Y_t^{\mathrm{R}(1)} + Y_t^{\mathrm{R}(2)} = N \cdot \sum_{t=1}^{T} Y_t^{\mathrm{R}(1)} \leqslant N \cdot \sum_{t=1}^{T} Z_t.$$

Since $Z_t$ are independent and $\mathbb{E}[\sum_{t=1}^{T} Z_t] = \sum_{t=1}^{T} \gamma_t \leqslant 2 T \gamma_T$, we can apply the Chernoff bound (which is a subcase of Lemma B.1) on $Z_t$,

$$\Pr \left[ \sum_{t=1}^{T} Z_t > 2 T \gamma_T + \sqrt{6 T \gamma_T \log \left( \frac{2}{\delta} \right)} \right] < \delta.$$

Hence we have $|\mathcal{T}_T^{\mathrm{R}}| = \sum_{t=1}^{T} Y_t^{\mathrm{R}} \leqslant 2 T N \gamma_T + N \sqrt{6 T \gamma_T \log(2/\delta)}$ with probability at least $1 - \delta$. $\quad \square$

**Claim 5.** Fix any $\delta \in (0,1)$, $|\widehat{\mathcal{T}}^{1(m)}| \leqslant L|\mathcal{K}|\log(1/\delta)(\log(|\mathcal{K}|) + 1)/\gamma_T$ with probability at most $1 - |\mathcal{K}|L\delta$.

*Proof of Claim 5.* If round $t$ is a change point, i.e., $(r_t, c_t) \neq (r_{t-1}, c_{t-1})$, we define set $\mathcal{K}(t) \subset \mathcal{K}$ such that $(r_t(a), c_t(a)) \neq (r_{t-1}(a), c_{t-1}(a))$ for all $a \in \mathcal{K}(t)$. Notice that after a change happens, some exploitation rounds could run with condition (6) violated, resulting in consuming resources from the wrong reward-consumption interval $m$. Hence, $|\widehat{\mathcal{T}}^{1(m)}|$ can be upper bounded by the total number of exploitation rounds run before all arms are updated after each change point. After all arms are updated after a change point, (6) is satisfied again. In the following Claim 6, we suppose round $t$ is a change point, and upper bound the number of exploitation rounds run before all arms $a \in \mathcal{K}(t)$ are updated.

**Claim 6.** Suppose round $t$ is a change point. With probability $1 - |\mathcal{K}(t)|\delta$, IRES-CM run at most $|\mathcal{K}|\log(1/\delta)(\log(|\mathcal{K}(t)|) + 1)/\gamma_T$ exploitation rounds (Algorithm 2, Lines 17-25) before updating the change in exploration rounds (Algorithm 2, Lines 9-13).

*Proof of Claim 6.* For some set $\mathcal{K}' \subset \mathcal{K}$, let random variable $Y(\mathcal{K}')$ denote the number of exploitation samples (i.e., line 6 of Algorithm 2 giving $U(t) = 0$) between two nearest exploration samples (i.e., line 6 of Algorithm 2 giving $U(t) = 1$) applied for arms $a \in \mathcal{K}'$. We denote $\mathcal{K}(t)^{(j)}$ as the $j$-th arm explored from set $\mathcal{K}(t)$. After each exploitation sample, IRES-CM runs for each $q \in \{-M, \ldots, M - 1\}$. Therefore, the number of exploitation rounds run before all $a \in \mathcal{K}(t)$ are updated is $2M \cdot (Y(\mathcal{K}(t)) + \sum_{j=2}^{|\mathcal{K}(t)|} Y(\mathcal{K}(t) \backslash \bigcup_{h=1}^{j-1} \mathcal{K}(t)^{(h)}))$. For any subset $\mathcal{K}' \in \mathcal{K}(t)$, we further denote random variable $Z(\mathcal{K}')$ as $Y(\mathcal{K}')$ plus the number of times that exploration is triggered for any $a \in \mathcal{K} \backslash \mathcal{K}'$ between two nearest exploration rounds applied for arms $a \in \mathcal{K}'$.

It is evident that $Z(\mathcal{K}')$ is a geometric random variable with time-varying probability $p_t = \gamma_t |\mathcal{K}'|/|\mathcal{K}|$ of success in each round. Therefore, we have $\Pr(Z(\mathcal{K}') \geqslant n) = \prod_{s=t}^{t+n-1}(1 - p_s)$. Since for any $x \in [0,1]$, it holds that $1 - x \leqslant e^{-x}$, by requiring $e^{-n \cdot p_T} \leqslant \delta$, we have $\prod_{s=t}^{t+n-1}(1 - p_s) \leqslant (1 - p_T)^n \leqslant e^{-n \cdot p_T} \leqslant \delta$. In this case,

$$e^{-n \cdot p_T} \leqslant \delta \Leftrightarrow -n \cdot p_T \leqslant -\log(1/\delta) \Leftrightarrow n \geqslant \frac{\log(1/\delta)}{p_T} = \frac{|\mathcal{K}|\log(1/\delta)}{\gamma_T |\mathcal{K}'|}.$$

Therefore, we have

$$\Pr\left(Y(\mathcal{K}') \geqslant \frac{|\mathcal{K}|\log(1/\delta)}{\gamma_T |\mathcal{K}'|}\right) \leqslant \Pr\left(Z(\mathcal{K}') \geqslant \frac{|\mathcal{K}|\log(1/\delta)}{\gamma_T |\mathcal{K}'|}\right) \leqslant \delta,$$

which suggests that with probability at least $\delta$, we run at most $|\mathcal{K}|\log(1/\delta)/(\gamma_T |\mathcal{K}'|)$ exploitation rounds before updating $(\hat{r}_t(a), \hat{c}_t(a))$ for each arm $a \in \mathcal{K}'$.

Plugging $\mathcal{K}(t) \backslash \bigcup_{h=1}^{j-1} \mathcal{K}(t)^{(h)}$ in $\mathcal{K}'$, with probability $1 - |\mathcal{K}(t)|\delta$,

$$
\begin{aligned}
Y(\mathcal{K}(t)) + \sum_{j=2}^{|\mathcal{K}(t)|} Y\left(\mathcal{K}(t) \backslash \bigcup_{h=1}^{j-1} \mathcal{K}(t)^{(h)}\right) &\leqslant \sum_{j=1}^{|\mathcal{K}(t)|} \frac{|\mathcal{K}|\log(1/\delta)}{\gamma_T \cdot j} \\
&\leqslant \frac{|\mathcal{K}|\log(1/\delta)}{\gamma_T} \sum_{j=1}^{|\mathcal{K}(t)|} \frac{1}{j} \\
&\leqslant \frac{|\mathcal{K}|\log(1/\delta)}{\gamma_T}(\log(|\mathcal{K}(t)|) + 1). \quad (31)
\end{aligned}
$$

Inequality (31) holds since

$$\sum_{j=1}^{|\mathcal{K}(t)|} \frac{1}{j} \leqslant 1 + \int_{j=1}^{|\mathcal{K}(t)|} \frac{1}{j} \leqslant 1 + \log(|\mathcal{K}(t)|) - \log(1). \quad (32)$$

$\square$

Notice that there are at most $L$ change points over the entire planning horizon, contributing to $\widehat{\mathcal{T}}^{1(m)}$ for at most $2ML|\mathcal{K}|\log(1/\delta)(\log(|\mathcal{K}|) + 1)/\gamma_T$ rounds, with probability at most $1 - |\mathcal{K}|L\delta$. $\square$

Combining Claim 4 and 5, with probability at least $1 - 2M|\mathcal{K}|L\delta$,

$$\left|\mathcal{T}_T^{\mathrm{R}} \bigcup \left(\bigcup_{m=-M}^{M-1} \widehat{\mathcal{T}}^{1(m)}\right)\right| \leqslant 2TN\gamma_T + N\sqrt{6T\gamma_T \log\left(\frac{2}{\delta}\right)} + \frac{4M^2 L|\mathcal{K}|\log(1/\delta)(\log(|\mathcal{K}|)+1)}{\gamma_T}$$

$$\leqslant 4TN\gamma_T + \frac{4M^2 L|\mathcal{K}|\log(1/\delta)(\log(|\mathcal{K}|)+1)}{\gamma_T}.$$

Recall that $N = 27\log(2/\delta)/((1-1/\alpha)^2 \cdot \eta_{\min})$ and $\gamma_t = M\sqrt{|\mathcal{K}|\log(1/\delta)(\log(|\mathcal{K}|)+1)}/\sqrt{Nt}$, we further have

$$\left|\mathcal{T}_T^{\mathrm{R}} \bigcup \left(\bigcup_{m=-M}^{M-1} \widehat{\mathcal{T}}^{1(m)}\right)\right| \leqslant 8ML\sqrt{|\mathcal{K}|NT\log(1/\delta)(\log(|\mathcal{K}|)+1)} = \tilde{O}(L\sqrt{|\mathcal{K}|NT}). \quad (33)$$

Note that if the DM knows $L$ a priori, then we can set $\gamma_t = M\sqrt{L|\mathcal{K}|\log(1/\delta)(\log(|\mathcal{K}|)+1)}/\sqrt{Nt}$, which results in $|\mathcal{T}_T^{\mathrm{R}} \bigcup (\bigcup_{m=-M}^{M-1} \widehat{\mathcal{T}}^{1(m)})| \leqslant \tilde{O}(\sqrt{L|\mathcal{K}|NT})$.

## C  Proofs

### C.1  Proof of Lemma 2.1

Let $\pi$ be a non-anticipatory feasible policy that achieves the expected optimum opt(DP) in DP, i.e. $\mathbb{E}[\sum_{t=1}^T \sum_{a\in\mathcal{K}} R_t(a)X_t^\pi(a)] = \mathbb{E}[\mathrm{opt}(\mathrm{DP})]$ where $X_t^\pi$ is the decision variable under algorithm $\pi$. We let

$$x_l(a) = \frac{1}{t_l - t_{l-1}}\mathbb{E}\left[\sum_{t=t_{l-1}+1}^{t_l} \sum_{a\in\mathcal{K}} X_t^\pi(a)\right]$$

for each $l = 1,\ldots,L$ in FA. We claim that $\{x_l\}_{l=1}^L$ is feasible to FA, with objective value equal to $\mathbb{E}[\sum_{t=1}^T \sum_{a\in\mathcal{K}} R_t(a)X_t^\pi(a)] = \mathbb{E}[\mathrm{opt}(\mathrm{DP})]$, which indicates that under $\{x_l^*\}_{l=1}^L$ we have opt(FA) $\geqslant \mathbb{E}[\mathrm{opt}(\mathrm{DP})]$. Thus, verifying the claims about the feasibility and the objective value proves the claim.

We first verify the feasibility to FA. Since the policy $\pi$ satisfies the resource constraints, the inequality $\sum_{t=1}^T \sum_{a\in\mathcal{K}} C_t(a)X_t^\pi(a) \leqslant B$ holds. Taking expectation over $X_t^\pi(a)$ and $C_t(a)$ for $t = t_{l-1} + 1,\ldots,t_l$ gives

$$\mathbb{E}\left[\sum_{t=1}^T \sum_{a\in\mathcal{K}} C_t(a)X_t^\pi(a)\right] = \sum_{l=1}^L \sum_{t=t_{l-1}+1}^{t_l} \sum_{a\in\mathcal{K}} c^{(l)}(a)\mathbb{E}[X_t^\pi(a)]$$

$$= \sum_{l=1}^L (t_l - t_{l-1}) \sum_{a\in\mathcal{K}} c^{(l)}(a)x_l(a)$$

$$\leqslant B.$$

Similarly, by taking expectation over each of the reward constraints, we have $\mathbb{E}[\sum_{t=1}^T \sum_{a\in\mathcal{K}} R_t(a)X_t^\pi(a)] = \sum_{l=1}^L (t_l - t_{l-1}) \sum_{a\in\mathcal{K}} r^{(l)}(a)x_l(a) = \mathbb{E}[\mathrm{opt}(\mathrm{DP})]$. Hence, the claim about the objective value is shown, and the Lemma is proved. □

### C.2  Proof of Claim 1

Recall that in Algorithm 1, we try each $q \in \{0,1,\ldots,M\}$ in a round-robin manner. Then we know that on each stationary piece $l$, for at least $(t_l - t_{l-1})/M - 1$ rounds, we choose $q = q_l^*$ and take fractional decision $x_l^{(q_l^*)*}$ such that (7) and (8) hold. By Claim 3 inequality (8), resources consumed under decision $x_l^{(q_l^*)*}$ are assigned with resources reserved for intervals $\{\max\{m_l^* - 1, -M\}, m_l^*\}$. Therefore, for interval $m$ where $m_l^* = m$,

$$\sum_{t\in\bigcup_{n=\max\{m-1,-M\}}^m \widetilde{\mathcal{T}}^{(n)} \bigcap \{t_{l-1}+1,\ldots,t_l\}} \mathbf{1}(q_t = q_l^*) \geqslant \frac{t_l - t_{l-1}}{M+1} - 1. \quad (34)$$

Recall that if we choose the optimal decision $x_l^*$ on stationary piece $l$, $B_l^*$ units of resources would be consumed, i.e. $(t_l - t_{l-1}) \sum_{a\in\mathcal{K}} c^{(l)}(a) x_l^*(a) = B_l^*$. Hence by Claim 3 inequality (7), we have

$$\sum_{a\in\mathcal{K}} c^{(l)}(a) x_l^{(q_l^*)*}(a) \geqslant \sum_{a\in\mathcal{K}} c^{(l)}(a) x_l^*(a) \geqslant \frac{B_l^*}{t_l - t_{l-1}}. \tag{35}$$

Putting everything together, we have

$$\sum_{t\in\bigcup_{m=-M}^{M-1} \tilde{\mathcal{T}}^{(m)} \bigcap\{t_{l-1}+1,...,t_l\}} \sum_{a\in\mathcal{K}} r_t(a) x_t(a)$$

$$\geqslant \sum_{t\in\bigcup_{n=\max\{m-1,-M\}}^{m} \tilde{\mathcal{T}}^{(n)} \bigcap\{t_{l-1}+1,...,t_l\}} \sum_{a\in\mathcal{K}} r_t(a) x_t(a)$$

$$\geqslant \sum_{t\in\bigcup_{n=\max\{m-1,-M\}}^{m} \tilde{\mathcal{T}}^{(n)} \bigcap\{t_{l-1}+1,...,t_l\}} \mathbf{1}(q_t = q_l^*) \sum_{a\in\mathcal{K}} r^{(l)}(a) x_l^{(q_l^*)*}(a)$$

$$\geqslant \sum_{t\in\bigcup_{n=\max\{m-1,-M\}}^{m} \tilde{\mathcal{T}}^{(n)} \bigcap\{t_{l-1}+1,...,t_l\}} \mathbf{1}(q_t = q_l^*) \cdot \sum_{a\in\mathcal{K}} c^{(l)}(a) x_l^{(q_l^*)*}(a) \cdot \frac{\sum_{a\in\mathcal{K}} r^{(l)}(a) x_l^{(q_l^*)*}(a)}{\sum_{a\in\mathcal{K}} c^{(l)}(a) x_l^{(q_l^*)*}(a)}$$

$$\geqslant \sum_{t\in\bigcup_{n=\max\{m-1,-M\}}^{m} \tilde{\mathcal{T}}^{(n)} \bigcap\{t_{l-1}+1,...,t_l\}} \mathbf{1}(q_t = q_l^*) \cdot \frac{B_l^*}{t_l - t_{l-1}} \cdot \frac{\sum_{a\in\mathcal{K}} r^{(l)}(a) x_l^*(a)}{\alpha \cdot \sum_{a\in\mathcal{K}} c^{(l)}(a) x_l^*(a)} \tag{36}$$

$$\geqslant \left(\frac{t_l - t_{l-1}}{M+1} - 1\right) \cdot \frac{B_l^*}{\alpha(t_l - t_{l-1})} \cdot \frac{\sum_{a\in\mathcal{K}} r^{(l)}(a) x_l^*(a)}{\sum_{a\in\mathcal{K}} c^{(l)}(a) x_l^*(a)} \tag{37}$$

$$\geqslant \frac{B_l^*}{2\alpha(M+1)} \cdot \frac{\sum_{a\in\mathcal{K}} r^{(l)}(a) x_l^*(a)}{\sum_{a\in\mathcal{K}} c^{(l)}(a) x_l^*(a)}. \tag{38}$$

Inequality (36) holds by plugging in (35) and (8). Inequality (37) holds by plugging in (34). Inequality (38) stands since we can assume $t_l - t_{l-1} \geqslant 2(M+1)$ without loss of generality. Because otherwise we can ignore the stationary pieces where $t_l - t_{l-1} \leqslant 2(M+1)$, causing a reward loss of at most $O(M)$. Following (38), we have

$$\sum_{w=\max\{m-1,-M\}}^{\min\{m+1,M-1\}} \sum_{l\in\mathcal{L}} \mathbf{1}(m_l^* = w) \sum_{t\in\bigcup_{m=-M}^{M-1} \tilde{\mathcal{T}}^{(m)} \bigcap\{t_{l-1}+1,...,t_l\}} \sum_{a\in\mathcal{K}} r_t(a) x_t(a)$$

$$\geqslant \sum_{l\in\mathcal{L}} \mathbf{1}(m_l^* = m) \sum_{t\in\bigcup_{m=-M}^{M-1} \tilde{\mathcal{T}}^{(m)} \bigcap\{t_{l-1}+1,...,t_l\}} \sum_{a\in\mathcal{K}} r_t(a) x_t(a)$$

$$\geqslant \frac{1}{2\alpha(M+1)} \cdot \sum_{l\in\mathcal{L}} \mathbf{1}(m_l^* = m) \cdot \frac{\sum_{a\in\mathcal{K}} r^{(l)}(a) x_l^*(a)}{\sum_{a\in\mathcal{K}} c^{(l)}(a) x_l^*(a)} \cdot B_l^*.$$

Therefore, for $m \in \{-M,\ldots,M-1\}$ such that $\tilde{\mathcal{T}}^{(n)} = \mathcal{T}_T^{(n)}$ for $n \in \{\max\{m-1,-M\}, m\}$, we have

$$\frac{\sum_{w=\max\{m-1,-M\}}^{\min\{m+1,M-1\}} \sum_{l\in\mathcal{L}} \mathbf{1}(m_l^* = w) \sum_{t\in\bigcup_{m=-M}^{M-1} \tilde{\mathcal{T}}^{(m)} \bigcap\{t_{l-1}+1,...,t_l\}} \sum_{a\in\mathcal{K}} r_t(a) x_t(a)}{\sum_{l\in\mathcal{L}} \mathbf{1}(m_l^* = m) \cdot \frac{\sum_{a\in\mathcal{K}} r^{(l)}(a) x_l^*(a)}{\sum_{a\in\mathcal{K}} c^{(l)}(a) x_l^*(a)} \cdot B_l^*} \geqslant \frac{1}{2\alpha(M+1)}. \tag{39}$$

$\square$

## C.3 Proof of Claim 2

We let $\tilde{m} = \min_n\{n \in \{\max\{m-1, -M\}, m\}, \tilde{\mathcal{T}}^{(n)} \subsetneq \mathcal{T}_T^{(n)}\}$. Then we have

$$\sum_{t\in\tilde{\mathcal{T}}^{(\tilde{m})}}\sum_{a\in\mathcal{K}} r_t(a)x_t(a) = \sum_{t\in\tilde{\mathcal{T}}^{(\tilde{m})}}\frac{\sum_{a\in\mathcal{K}} r_t(a)x_t(a)}{\sum_{a\in\mathcal{K}} c_t(a)x_t(a)} \cdot \sum_{a\in\mathcal{K}} c_t(a)x_t(a)$$

$$\geqslant \alpha^{\tilde{m}} \sum_{t\in\tilde{\mathcal{T}}^{(\tilde{m})}}\sum_{a\in\mathcal{K}} c_t(a)x_t(a) \tag{40}$$

$$\geqslant \alpha^{\tilde{m}} \cdot \left(\frac{B-1}{2M} - 1\right). \tag{41}$$

Inequality (40) stands since in rounds $t \in \tilde{\mathcal{T}}^{(\tilde{m})}$, we have $\sum_{a\in\mathcal{K}} r_t(a)x_t(a)/\sum_{a\in\mathcal{K}} c_t(a)x_t(a) \in [\alpha^{\tilde{m}}, \alpha^{\tilde{m}+1}]$. Inequality (41) holds since $\sum_{t\in\tilde{\mathcal{T}}^{(\tilde{m})}}\sum_{a\in\mathcal{K}} c_t(a)x_t(a) \geqslant (B - T \cdot \eta_{\min})/(2M) - 1 = (B-1)/(2M) - 1$ by the definition of $\tilde{\mathcal{T}}^{(\tilde{m})}$. Then we have

$$\sum_{w=\{\max\{m-1,-M\}}^{\min\{m+1,M-1\}}\sum_{l\in\mathcal{L}} \mathbf{1}(m_l^* = w) \sum_{t\in\bigcup_{m=-M}^{M-1}\tilde{\mathcal{T}}^{(m)}\bigcap\{t_{l-1}+1,\ldots,t_l\}}\sum_{a\in\mathcal{K}} r_t(a)x_t(a)$$

$$\geqslant \sum_{w=\{\max\{m-1,-M\}}^{\min\{m+1,M-1\}}\sum_{l\in\mathcal{L}} \mathbf{1}(m_l^* = w) \sum_{t\in\tilde{\mathcal{T}}^{(\tilde{m})}\bigcap\{t_{l-1}+1,\ldots,t_l\}}\sum_{a\in\mathcal{K}} r_t(a)x_t(a)$$

$$\geqslant \sum_{t\in\tilde{\mathcal{T}}^{(\tilde{m})}}\sum_{a\in\mathcal{K}} r_t(a)x_t(a) \tag{42}$$

$$\geqslant \alpha^{\tilde{m}} \cdot \left(\frac{B-1}{2M} - 1\right).$$

Inequality (42) holds since $m_t \in \{\{\max\{m_l^*-1, -M\}, m_l^*\}$ for all $t \in \{t_{l-1}+1, \ldots, t_l\}$. Therefore, it is possible to consume resources reserved for interval $\tilde{m} \in \{\max\{m-1, -M\}, m\}$ under $x_l^{(q_l^*)*}$ only when

$$m_l^* \in \{\max\{m-1, -M\}, m-1+1\}\bigcup\{m, \min\{m+1, M-1\}\} = \{\max\{m-1, -M\}, m, \min\{m+1, M-1\}\}.$$

We also have

$$\sum_{l\in\mathcal{L}}\mathbf{1}(m_l^* = m) \cdot \frac{\sum_{a\in\mathcal{K}} r^{(l)}(a)x_l^*(a)}{\sum_{a\in\mathcal{K}} c^{(l)}(a)x_l^*(a)} \cdot B_l^* \leqslant \alpha^{m+1} \cdot \sum_{l\in\mathcal{L}}^{L} B_l^* \leqslant \alpha^{m+1}B. \tag{43}$$

Putting together (42) and (43), we know that for $m \in \{-M, \ldots, M-1\}$ satisfying case (ii), we have

$$\frac{\sum_{w=\max\{m-1,-M\}}^{\min\{m+1,M-1\}}\sum_{l\in\mathcal{L}}\mathbf{1}(m_l^* = w)\sum_{t\in\tilde{\mathcal{T}}^{(\tilde{m})}\bigcap\{t_{l-1}+1,\ldots,t_l\}}\sum_{a\in\mathcal{K}} r_t(a)x_t(a)}{\sum_{l\in\mathcal{L}}\mathbf{1}(m_l^* = m) \cdot \frac{\sum_{a\in\mathcal{K}} r^{(l)}(a)x_l^*(a)}{\sum_{a\in\mathcal{K}} c^{(l)}(a)x_l^*(a)} \cdot B_l^*}$$

$$\geqslant \frac{\alpha^{\tilde{m}} \cdot ((B-1)/(2M) - 1)}{\alpha^{m+1}B}$$

$$\geqslant \frac{1 - \frac{2M+1}{B}}{2\alpha^2 M}$$

$$\geqslant \frac{1 - o(1)}{2\alpha^2 M}. \tag{44}$$

Inequality (44) holds since we require $B \geqslant \Omega(M)$ (see Theorem 3.1). Combining (39) and (44), we show that

$$(14) \geqslant \frac{1 - o(1)}{6\alpha^2 M}. \tag{45}$$

## C.4 Proof of Theorem 4.2

Note that for reward-consumption ratio intervals $n \in \{-M, \ldots, M-1\}$ where $\tilde{\mathcal{T}}^{1(n)} = \check{\mathcal{T}}^{1(n)}$, not all requests assigned to these intervals are necessarily satisfied. This is because resources could run out due to exploration before $\sum_{s \in \check{\mathcal{T}}^{1(n)}} C_s(a_s) > B/(2M) - 1$, i.e., when

$$\tilde{\mathcal{T}}^{1(n)} \bigcap \left\{ t \in \mathcal{T} : \sum_{s=1}^{t} C_s(a_s) \leqslant B - 1 \right\} \subsetneq \tilde{\mathcal{T}}^{1(n)} \bigcap \mathcal{T}.$$

It can be seen that requests in rounds $\left( \bigcup_{n=-M}^{M-1} \tilde{\mathcal{T}}^{1(n)} \right) \bigcap \{ t \in \mathcal{T} : \sum_{s=1}^{t} C_s(a_s) \leqslant B - 1 \}$ are satisfied. Therefore, we decompose the ratio of the IRES-CM reward to opt(FA) as follows:

$$\frac{\sum_{t \in \mathcal{T}} R_t(a_t)}{\mathrm{opt(FA)}} = \underbrace{\frac{\sum_{t \in \left( \bigcup_{n=-M}^{M-1} \tilde{\mathcal{T}}^{1(n)} \right) \bigcap \{ t \in \mathcal{T} : \sum_{s=1}^{t} C_s(a_s) \leqslant B - 1 \}} \sum_{a \in \mathcal{K}} r_t(a) \hat{x}_t^{(q_t)*}(a)}{\mathrm{opt(FA)}}}_{\mathrm{H(1)}}$$

$$\cdot \underbrace{\frac{\sum_{t \in \mathcal{T}} R_t(a_t)}{\sum_{t \in \left( \bigcup_{n=-M}^{M-1} \tilde{\mathcal{T}}^{1(n)} \right) \bigcap \{ t \in \mathcal{T} : \sum_{s=1}^{t} C_s(a_s) \leqslant B - 1 \}} \sum_{a \in \mathcal{K}} r_t(a) \hat{x}_t^{(q_t)*}(a)}}_{\mathrm{H(2)}}.$$

To establish Theorem 4.2, it suffices to show H(1) $\geqslant$ $(1 - o(1))$ $\cdot$ (opt(FA) $-$ $\tilde{O}(\sqrt{L|\mathcal{K}|NT}))/(10\alpha^4 M \cdot \mathrm{opt(FA)})$ (see the forthcoming Section C.4.1) and H(2) $\geqslant 1 - o(1)$ (see the forthcoming Section C.4.2).

### C.4.1 Bounding H(1)

Recall that $\sum_{a \in \mathcal{K}} \hat{r}_t(a) \hat{x}_t^{(q_t)*}(a) / \sum_{a \in \mathcal{K}} \hat{c}_t(a) \hat{x}_t^{(q_t)*}(a) \in [\alpha^{\hat{m}_t}, \alpha^{\hat{m}_t+1}]$. We further define $\hat{m}_t^{(l)*}$ such that $\sum_{a \in \mathcal{K}} \hat{r}_t(a) \hat{x}_t^{(q_l^*)*}(a) / \sum_{a \in \mathcal{K}} \hat{c}_t(a) \hat{x}_t^{(q_l^*)*}(a) \in [\alpha^{\hat{m}_t^{(l)*}}, \alpha^{\hat{m}_t^{(l)*}+1}]$. Likewise, we define $m_t^{(l)*}$ such that $\sum_{a \in \mathcal{K}} r_t(a) \hat{x}_t^{(q_l^*)*}(a) / \sum_{a \in \mathcal{K}} c_t(a) \hat{x}_t^{(q_l^*)*}(a) \in [\alpha^{m_t^{(l)*}}, \alpha^{m_t^{(l)*}+1}]$. We define

$$\tilde{\mathcal{J}}_l = \mathcal{J}_l \bigcap \left\{ t \in \mathcal{T} : \sum_{s=1}^{t} C_s(a_s) \leqslant B - 1 \right\}.$$

Then H(1) can be further decomposed as:

$$\mathrm{H(1)} = \frac{\sum_{t \in \left( \bigcup_{n=-M}^{M-1} \tilde{\mathcal{T}}^{1(n)} \right) \bigcap \{ t \in \mathcal{T} : \sum_{s=1}^{t} C_s(a_s) \leqslant B - 1 \}} \sum_{a \in \mathcal{K}} r_t(a) \hat{x}_t^{(q_t)*}(a)}{\mathrm{opt(FA)}}$$

$$= \frac{\sum_{l \in \mathcal{L}} \sum_{t \in \left( \bigcup_{n=-M}^{M-1} \tilde{\mathcal{T}}^{1(n)} \right) \bigcap \tilde{\mathcal{J}}_l} \sum_{a \in \mathcal{K}} r_t(a) \hat{x}_t^{(q_t)*}(a)}{\sum_{l \in \mathcal{L}} \sum_{t \in \mathcal{J}_l} \sum_{a \in \mathcal{K}} r^{(l)}(a) x_l^*(a)}$$

$$= \frac{\sum_{m=-M}^{M-1} \sum_{l \in \mathcal{L}} \sum_{t \in \left( \bigcup_{n=-M}^{M-1} \tilde{\mathcal{T}}^{1(n)} \right) \bigcap \tilde{\mathcal{J}}_l} \mathbf{1}(m_t^{(l)*} = m) \cdot \sum_{a \in \mathcal{K}} r_t(a) \hat{x}_t^{(q_t)*}(a)}{\sum_{m=-M}^{M-1} \sum_{l \in \mathcal{L}} \sum_{t \in \mathcal{J}_l} \mathbf{1}(m_t^{(l)*} = m) \cdot \sum_{a \in \mathcal{K}} r^{(l)}(a) x_l^*(a)} \qquad (46)$$

$$\geqslant \frac{1}{5} \cdot \frac{\sum_{m=-M}^{M-1} \sum_{l \in \mathcal{L}} \sum_{t \in \left( \bigcup_{n=-M}^{M-1} \tilde{\mathcal{T}}^{1(n)} \right) \bigcap \tilde{\mathcal{J}}_l} \sum_{w=\max\{m-2,-M\}}^{\min\{m+2,M-1\}} \mathbf{1}(m_t^{(l)*} = w) \cdot \sum_{a \in \mathcal{K}} r_t(a) \hat{x}_t^{(q_t)*}(a)}{\sum_{m=-M}^{M-1} \sum_{l \in \mathcal{L}} \sum_{t \in \mathcal{J}_l} \mathbf{1}(m_t^{(l)*} = m) \cdot \sum_{a \in \mathcal{K}} r^{(l)}(a) x_l^*(a)}.$$

$$(47)$$

Inequality (47) holds since by summing over $w \in \{\max\{m-2, -M\}, \ldots, \min\{m+2, M-1\}\}$, we repeat the numerator of (46) for at most 5 times.

We partition set $\{-M, \ldots, M-1\}$ into two disjoint sets $\mathcal{M}_1, \mathcal{M}_2$. An interval $m \in \mathcal{M}_1$ if for all $n \in \{\max\{m-1, -M\}, m, \min\{m+1, M-1\}\}$, we have $\tilde{\mathcal{T}}^{1(n)} = \check{\mathcal{T}}^{1(n)}$, i.e. $\sum_{s \in \check{\mathcal{T}}^{1(n)}} C_s(a_s) \leqslant B/(2M) - 1$. An interval $m \in \mathcal{M}_2$ if for some $n \in \{\max\{m-1, -M\}, m, \min\{m+1, M-1\}\}$, we have $\tilde{\mathcal{T}}^{1(n)} \subsetneq \check{\mathcal{T}}^{1(n)}$.

**Regarding** $m \in \mathcal{M}_1$. In the following analysis, we focus on the good event that (16), (17), (18) in Lemma B.3 hold for all $t \in \mathcal{T}$. We know that with probability at least $1 - 2|\mathcal{K}|T\delta$, the good event holds. On each stationary piece $l \in \mathcal{L}$ and all $t \in \mathcal{J}_l$, for all rounds $t \in \left(\bigcup_{n=-M}^{M-1} \tilde{\mathcal{T}}^{1(n)}\right) \bigcap \mathcal{J}_l$ such that $m_t^{(l)*} = m$, we know that $\hat{m}_t^{(l)*} \in \{\max\{m-1,-M\}, m, \min\{m+1,M-1\}\}$ (see (16) in Lemma B.3). Due to the round-robin technique, at least $1/(M+1)$ fraction of all rounds $t \in \left(\bigcup_{n=-M}^{M-1} \tilde{\mathcal{T}}^{1(n)}\right) \bigcap \mathcal{J}_l, l \in \mathcal{L}$ such that $m_t^{(l)*} = m$ are allocated by resources reserved for intervals $n \in \{\max\{m-1,-M\}, m, \min\{m+1,M-1\}\}$. Therefore, we have

$$\sum_{l\in\mathcal{L}} \sum_{t\in\left(\bigcup_{n=-M}^{M-1}\tilde{\mathcal{T}}^{1(n)}\right)\bigcap\mathcal{J}_l} \mathbf{1}(m_t^{(l)*}=m)\cdot\mathbf{1}(q_t=q_l^*) \geqslant \sum_{l\in\mathcal{L}} \sum_{t\in\left(\bigcup_{n=\max\{m-1,-M\}}^{\min\{m+1,M-1\}}\tilde{\mathcal{T}}^{1(n)}\right)\bigcap\mathcal{J}_l} \mathbf{1}(m_t^{(l)*}=m)\cdot\mathbf{1}(q_t=q_l^*)$$

$$\geqslant \frac{\sum_{l\in\mathcal{L}}\sum_{t\in\left(\bigcup_{n=\max\{m-1,-M\}}^{\min\{m+1,M-1\}}\tilde{\mathcal{T}}^{1(n)}\right)\bigcap\mathcal{J}_l}\mathbf{1}(m_t^{(l)*}=m)}{M+1}. \tag{48}$$

We ignore the stationary pieces $l$ where $|\mathcal{J}_l| \leqslant 2M$, since this cause a loss of at most $O(M)$.

For $m \in \mathcal{M}_1$, although we have $\bigcup_{n=\max\{m-1,-M\}}^{\min\{m+1,M-1\}} \tilde{\mathcal{T}}^{1(n)} = \bigcup_{n=\max\{m-1,-M\}}^{\min\{m+1,M-1\}} \check{\mathcal{T}}^{1(n)}$ (i.e., $\sum_{s\in\check{\mathcal{T}}^{1(n)}} C_s(a_s) \leqslant B/(2M) - 1$ for all $n \in \{\max\{m-1,-M\}, m, \min\{m+1,M-1\}\}$), it is not necessary that all requests assigned to intervals $\{\max\{m-1,-M\}, m, \min\{m+1,M-1\}\}$ are satisfied. The resource units reserved for these intervals can run out due to exploration, i.e., when

$$\bigcup_{n=\max\{m-1,-M\}}^{\min\{m+1,M-1\}} \tilde{\mathcal{T}}^{1(n)} \bigcap \left\{t \in \mathcal{T} : \sum_{s=1}^{t} C_s(a_s) \leqslant B-1\right\} \subsetneq \bigcup_{n=\max\{m-1,-M\}}^{\min\{m+1,M-1\}} \tilde{\mathcal{T}}^{1(n)} \bigcap \mathcal{T}.$$

We define

$$(\dagger)^{(m)} = \sum_{l\in\mathcal{L}} \sum_{t\in\left(\bigcup_{n=-M}^{M-1}\tilde{\mathcal{T}}^{1(n)}\right)\bigcap\mathcal{J}_l\backslash\tilde{\mathcal{J}}_l} \mathbf{1}(m_t^{(l)*}=m)\cdot\sum_{a\in\mathcal{K}}r_t(a)\hat{x}_t^{(q_t)*}(a).$$

Then we have

$$\sum_{l\in\mathcal{L}} \sum_{t\in\left(\bigcup_{n=-M}^{M-1}\tilde{\mathcal{T}}^{1(n)}\right)\bigcap\tilde{\mathcal{J}}_l} \sum_{w=\max\{m-2,-M\}}^{\min\{m+2,M-1\}} \mathbf{1}(m_t^{(l)*}=w)\cdot\sum_{a\in\mathcal{K}}r_t(a)\hat{x}_t^{(q_t)*}(a)$$

$$\geqslant \sum_{l\in\mathcal{L}} \sum_{t\in\left(\bigcup_{n=-M}^{M-1}\tilde{\mathcal{T}}^{1(n)}\right)\bigcap\tilde{\mathcal{J}}_l} \mathbf{1}(m_t^{(l)*}=m)\cdot\sum_{a\in\mathcal{K}}r_t(a)\hat{x}_t^{(q_t)*}(a)$$

$$= \sum_{l\in\mathcal{L}} \sum_{t\in\left(\bigcup_{n=-M}^{M-1}\tilde{\mathcal{T}}^{1(n)}\right)\bigcap\mathcal{J}_l} \mathbf{1}(m_t^{(l)*}=m)\cdot\sum_{a\in\mathcal{K}}r_t(a)\hat{x}_t^{(q_t)*}(a) - (\dagger)^{(m)}$$

$$\geqslant \sum_{l\in\mathcal{L}} \sum_{t\in\left(\bigcup_{n=-M}^{M-1}\tilde{\mathcal{T}}^{1(n)}\right)\bigcap\mathcal{J}_l} \mathbf{1}(m_t^{(l)*}=m)\cdot\sum_{a\in\mathcal{K}}r_t(a)\hat{x}_t^{(q_t)*}(a) - (\dagger)^{(m)}$$

$$\geqslant \sum_{l\in\mathcal{L}} \sum_{t\in\left(\bigcup_{n=\max\{m-1,-M\}}^{\min\{m+1,M-1\}}\tilde{\mathcal{T}}^{1(n)}\right)\bigcap\mathcal{J}_l} \mathbf{1}(m_t^{(l)*}=m)\cdot\mathbf{1}(q_t=q_l^*)\cdot\sum_{a\in\mathcal{K}}r_t(a)\hat{x}_t^{(q_l^*)*}(a) - (\dagger)^{(m)}$$

$$\geqslant \frac{1}{\alpha}\cdot\sum_{l\in\mathcal{L}} \sum_{t\in\left(\bigcup_{n=\max\{m-1,-M\}}^{\min\{m+1,M-1\}}\tilde{\mathcal{T}}^{1(n)}\right)\bigcap\mathcal{J}_l} \mathbf{1}(m_t^{(l)*}=m)\cdot\mathbf{1}(q_t=q_l^*)\cdot\sum_{a\in\mathcal{K}}r^{(l)}(a)x_l^*(a) - (\dagger)^{(m)}$$

$$\tag{49}$$

$$\geqslant \frac{1}{\alpha(M+1)}\cdot\sum_{l\in\mathcal{L}} \sum_{t\in\left(\bigcup_{n=\max\{m-1,-M\}}^{\min\{m+1,M-1\}}\tilde{\mathcal{T}}^{1(n)}\right)\bigcap\mathcal{J}_l} \mathbf{1}(m_t^{(l)*}=m)\cdot\sum_{a\in\mathcal{K}}r^{(l)}(a)x_l^*(a) - (\dagger)^{(m)}. \tag{50}$$

Inequality (49) follows from (17) in Lemma B.3, and inequality (50) follows from inequality (48). Hence, we have

$$\frac{\sum_{m\in\mathcal{M}_1}\sum_{l\in\mathcal{L}}\sum_{t\in\left(\bigcup_{n=-M}^{M-1}\tilde{\mathcal{T}}^{1(n)}\right)\cap\tilde{\mathcal{J}}_l}\mathbf{1}(m_t^{(l)*}=m)\cdot\sum_{a\in\mathcal{K}}r_t(a)\hat{x}_t^{(q_t)*}(a)}{\sum_{m\in\mathcal{M}_1}\sum_{l\in\mathcal{L}}\sum_{t\in\mathcal{J}_l}\mathbf{1}(m_t^{(l)*}=m)\cdot\sum_{a\in\mathcal{K}}r^{(l)}(a)x_l^*(a)}$$

$$\geqslant\max\left\{\frac{\frac{1}{\alpha(M+1)}\cdot\sum_{m\in\mathcal{M}_1}\sum_{l=1}^{L}\sum_{t\in\left(\bigcup_{n=\max\{m-1,-M\}}^{\min\{m+1,M-1\}}\tilde{\mathcal{T}}^{1(n)}\right)\cap\mathcal{J}_l}\mathbf{1}(m_t^{(l)*}=m)\cdot\sum_{a\in\mathcal{K}}r^{(l)}(a)x_l^*(a)-\sum_{m\in\mathcal{M}_1}(\dagger)^{(m)}}{\sum_{m\in\mathcal{M}_1}\sum_{l\in\mathcal{L}}\sum_{t\in\mathcal{J}_l}\mathbf{1}(m_t^{(l)*}=m)\cdot\sum_{a\in\mathcal{K}}r^{(l)}(a)x_l^*(a)},0\right\}$$

$$\geqslant\max\left\{\frac{\frac{1}{\alpha(M+1)}\cdot\sum_{m\in\mathcal{M}_1}\sum_{l=1}^{L}\sum_{t\in\left(\bigcup_{n=\max\{m-1,-M\}}^{\min\{m+1,M-1\}}\tilde{\mathcal{T}}^{1(n)}\right)\cap\mathcal{J}_l}\mathbf{1}(m_t^{(l)*}=m)\cdot\sum_{a\in\mathcal{K}}r^{(l)}(a)x_l^*(a)-|\mathcal{T}_T^{\mathrm{R}}|}{\sum_{m\in\mathcal{M}_1}\sum_{l\in\mathcal{L}}\sum_{t\in\mathcal{J}_l}\mathbf{1}(m_t^{(l)*}=m)\cdot\sum_{a\in\mathcal{K}}r^{(l)}(a)x_l^*(a)},0\right\}$$

(51)

$$\geqslant\max\left\{\frac{\frac{1}{\alpha(M+1)}\cdot\sum_{m\in\mathcal{M}_1}\sum_{l\in\mathcal{L}}\sum_{t\in\mathcal{J}_l}\mathbf{1}(m_t^{(l)*}=m)\cdot\sum_{a\in\mathcal{K}}r^{(l)}(a)x_l^*(a)-\left|\mathcal{T}_T^{\mathrm{R}}\bigcup\left(\bigcup_{m=-M}^{M-1}\hat{\mathcal{T}}^{1(m)}\right)\right|}{\sum_{m\in\mathcal{M}_1}\sum_{l\in\mathcal{L}}\sum_{t\in\mathcal{J}_l}\mathbf{1}(m_t^{(l)*}=m)\cdot\sum_{a\in\mathcal{K}}r^{(l)}(a)x_l^*(a)},0\right\}$$

(52)

$$\geqslant\max\left\{\frac{1}{\alpha(M+1)}-\frac{\tilde{O}(\sqrt{L|\mathcal{K}|NT})}{\sum_{m\in\mathcal{M}_1}\sum_{l\in\mathcal{L}}\sum_{t\in\mathcal{J}_l}\mathbf{1}(m_t^{(l)*}=m)\cdot\sum_{a\in\mathcal{K}}r^{(l)}(a)x_l^*(a)},0\right\}\quad\text{w.p. }1-2M|\mathcal{K}|L\delta.$$

(53)

Inequality (51) holds since $\sum_{m=-M}^{M-1}(\dagger)^{(m)}\leqslant|\mathcal{T}_T^{\mathrm{R}}|$. Inequality (52) is valid since

$$\left(\bigcup_{n=\max\{m-1,-M\}}^{\min\{m+1,M-1\}}\tilde{\mathcal{T}}^{1(n)}\right)\bigcap\mathcal{J}_l=\left(\bigcup_{n=\max\{m-1,-M\}}^{\min\{m+1,M-1\}}\check{\mathcal{T}}^{1(n)}\right)\bigcap\mathcal{J}_l=\mathcal{J}_l\backslash\left(\bigcup_{n=\max\{m-1,-M\}}^{\min\{m+1,M-1\}}\hat{\mathcal{T}}^{1(n)}\right).$$

**Regarding** $m\in\mathcal{M}_2$**.** Suppose in some interval $\hat{n}\in\{\max\{m-1,-M\},\dots,\min\{m+1,M-1\}\}$, we have $\sum_{t\in\tilde{\mathcal{T}}^{1(\hat{n})}}C_s(a_s)>B/(2M)-1$. We aim validate the following two inequalities respectively:

$$\sum_{l\in\mathcal{L}}\sum_{t\in\left(\bigcup_{n=-M}^{M-1}\tilde{\mathcal{T}}^{1(n)}\right)\cap\tilde{\mathcal{J}}_l}\sum_{w=\max\{m-2,-M\}}^{\min\{m+2,M-1\}}\mathbf{1}(m_t^{(l)*}=w)\cdot\sum_{a\in\mathcal{K}}r_t(a)\hat{x}_t^{(q_t)*}(a)$$

$$\geqslant\alpha^{\hat{n}-1/2}\cdot\left(\frac{B}{2M}-\sqrt{\frac{B}{M}\cdot\log(1/\delta)}-4\log(1/\delta)-1\right),$$

(54)

$$\sum_{l\in\mathcal{L}}\sum_{t\in\mathcal{J}_l}\mathbf{1}(m_t^{(l)*}=m)\cdot\sum_{a\in\mathcal{K}}r^{(l)}(a)x_l^*(a)\leqslant\alpha^{m+5/2}\cdot B,$$

(55)

Since given (54) and (55),for any interval $m\in\mathcal{M}_2$,

$$\frac{\sum_{m\in\mathcal{M}_2}\sum_{l\in\mathcal{L}}\sum_{t\in\left(\bigcup_{n=-M}^{M-1}\tilde{\mathcal{T}}^{1(n)}\right)\cap\tilde{\mathcal{J}}_l}\sum_{w=\max\{m-2,-M\}}^{\min\{m+2,M-1\}}\mathbf{1}(m_t^{(l)*}=w)\cdot\sum_{a\in\mathcal{K}}r_t(a)\hat{x}_t^{(q_t)*}(a)}{\sum_{m\in\mathcal{M}_2}\sum_{l\in\mathcal{L}}\sum_{t\in\mathcal{J}_l}\mathbf{1}(m_t^{(l)*}=m)\cdot\sum_{a\in\mathcal{K}}r^{(l)}(a)x_l^*(a)}$$

$$\geqslant\frac{\alpha^{\hat{n}-1/2}}{\alpha^{m+5/2}}\cdot\left(\frac{1}{2M}-\sqrt{\frac{1}{BM}\cdot\log(1/\delta)}-\frac{4\log(1/\delta)+1}{B}\right)$$

$$\geqslant\frac{\alpha^{m-3/2}}{\alpha^{m+5/2}}\cdot\left(\frac{1}{2M}-\sqrt{\frac{1}{BM}\cdot\log(1/\delta)}-\frac{4\log(1/\delta)+1}{B}\right)$$

$$\geqslant\frac{1-o(1)}{2\alpha^4M}.$$

(56)

**Validating (54).** In our algorithm, for any $t \in \bigcup_{m=-M}^{M-1} \tilde{\mathcal{T}}^{1(m)}$, we have $a_t \sim \hat{x}_t^{(q_t)*}$, and hence, $\mathbb{E}[R_t(a_t)|\mathcal{F}_{t-1}] = \sum_{a \in \mathcal{K}} r_t(a)\hat{x}_t^{(q_t)*}(a)$. Therefore, by Lemma B.2, for any $\delta \in (0,1)$, with probability at least $1 - 3\delta$, we have

$$\sum_{t \in \tilde{\mathcal{T}}^{1(\hat{n})}} \sum_{a \in \mathcal{K}} c_t(a)\hat{x}_t^{(q_t)*}(a) \geqslant \sum_{t \in \tilde{\mathcal{T}}^{1(\hat{n})}} C_t(a_t) - \sqrt{2 \sum_{t \in \tilde{\mathcal{T}}^{1(\tilde{m})}} C_t(a_t) \cdot \log(1/\delta)} - 4\log(1/\delta)$$

$$\geqslant \frac{B}{2M} - \sqrt{\frac{B}{M} \cdot \log(1/\delta)} - 4\log(1/\delta) - 1. \tag{57}$$

Inequality (57) holds since $B/(2M) - 1 < \sum_{t \in \tilde{\mathcal{T}}^{1(\hat{n})}} C_s(a_s) \leqslant B/(2M)$. Then we have

$$\sum_{t \in \tilde{\mathcal{T}}^{1(\hat{n})}} \sum_{a \in \mathcal{K}} r_t(a)\hat{x}_t^{(q_t)*}(a) = \sum_{t \in \tilde{\mathcal{T}}^{1(\hat{n})}} \frac{\sum_{a \in \mathcal{K}} r_t(a)\hat{x}_t^{(q_t)*}(a)}{\sum_{a \in \mathcal{K}} c_t(a)\hat{x}_t^{(q_t)*}(a)} \cdot \sum_{a \in \mathcal{K}} c_t(a)\hat{x}_t^{(q_t)*}(a)$$

$$\geqslant \alpha^{\hat{n}-1/2} \cdot \sum_{t \in \tilde{\mathcal{T}}^{1(\hat{n})}} \sum_{a \in \mathcal{K}} c_t(a)\hat{x}_t^{(q_t)*}(a) \tag{58}$$

$$\geqslant \alpha^{\hat{n}-1/2} \cdot \left( \frac{B}{2M} - \sqrt{\frac{B}{M} \cdot \log(1/\delta)} - 4\log(1/\delta) - 1 \right). \tag{59}$$

Inequality (58) holds since for all $t \in \tilde{\mathcal{T}}^{1(\hat{n})}$, we have $\sum_{a \in \mathcal{K}} \hat{r}_t(a)\hat{x}_t^{(q_t)*}(a) / \sum_{a \in \mathcal{K}} \hat{c}_t(a)\hat{x}_t^{(q_t)*}(a) \geqslant \alpha^{\hat{n}}$. Then by inequality (16) in Lemma B.3, we have $\sum_{a \in \mathcal{K}} r_t(a)\hat{x}_t^{(q_t)*}(a) / \sum_{a \in \mathcal{K}} c_t(a)\hat{x}_t^{(q_t)*}(a) \geqslant \alpha^{\hat{n}-1}$. Inequality (59) follows from inequality (57). We further have

$$\sum_{l \in \mathcal{L}} \sum_{t \in \left( \bigcup_{n=-M}^{M-1} \tilde{\mathcal{T}}^{1(n)} \right) \cap \tilde{\mathcal{J}}_l} \sum_{w=\max\{m-2,-M\}}^{\min\{m+2,M-1\}} \mathbf{1}(m_t^{(l)*} = w) \cdot \sum_{a \in \mathcal{K}} r_t(a)\hat{x}_t^{(q_t)*}(a)$$

$$\geqslant \sum_{l \in \mathcal{L}} \sum_{t \in \tilde{\mathcal{T}}^{1(\hat{n})} \cap \tilde{\mathcal{J}}_l} \sum_{w=\max\{m-2,-M\}}^{\min\{m+2,M-1\}} \mathbf{1}(m_t^{(l)*} = w) \cdot \sum_{a \in \mathcal{K}} r_t(a)\hat{x}_t^{(q_t)*}(a)$$

$$= \sum_{l \in \mathcal{L}} \sum_{t \in \tilde{\mathcal{T}}^{1(\hat{n})} \cap \mathcal{J}_l} \sum_{w=\max\{m-2,-M\}}^{\min\{m+2,M-1\}} \mathbf{1}(m_t^{(l)*} = w) \cdot \sum_{a \in \mathcal{K}} r_t(a)\hat{x}_t^{(q_t)*}(a) \tag{60}$$

$$\geqslant \sum_{l \in \mathcal{L}} \sum_{t \in \tilde{\mathcal{T}}^{1(\hat{n})} \cap \mathcal{J}_l} \sum_{a \in \mathcal{K}} r_t(a)\hat{x}_t^{(q_t)*}(a) \tag{61}$$

$$= \sum_{t \in \tilde{\mathcal{T}}^{1(\hat{n})}} \sum_{a \in \mathcal{K}} r_t(a)\hat{x}_t^{(q_t)*}(a)$$

$$\geqslant \alpha^{\hat{n}-1/2} \cdot \left( \frac{B}{2M} - \sqrt{\frac{B}{M} \cdot \log(1/\delta)} - 4\log(1/\delta) - 1 \right).$$

Inequality (60) holds since the total $B$ resource units have not run out before the reserved $B/(2M)$ resource units for interval $\hat{n}$ run out, i.e.,

$$\tilde{\mathcal{T}}^{1(\hat{n})} \bigcap \left\{ t \in \mathcal{T} : \sum_{s=1}^{t} C_s(a_s) \leqslant B - 1 \right\} = \tilde{\mathcal{T}}^{1(\hat{n})} \bigcap \mathcal{T}.$$

Inequality (61) is valid since for $t \in \tilde{\mathcal{T}}^{1(\hat{n})}$, we have $\hat{m}_t^{(l)*} = \hat{n}$. Hence, we have

$$m_t^{(l)*} \in \{\max\{\hat{n}-1,-M\}, \hat{n}, \min\{\hat{n}+1, M-1\}\} \in \{\max\{m-2,-M\}, \ldots, \min\{m+2, M-1\}\}.$$

**Validating (55).** For $l \in \mathcal{L}$ such that $q_l^* = 0$, by (18) in Lemma B.3, we have

$$\sum_{t \in \mathcal{J}_l} \mathbf{1}(m_t^{(l)*} = m) \cdot \sum_{a \in \mathcal{K}} r^{(l)}(a) x_l^*(a)$$

$$= \sum_{t \in \mathcal{J}_l} \mathbf{1} \left( \frac{\sum_{a \in \mathcal{K}} \hat{r}_t(a) \hat{x}_t^{(q_l^*)*}(a)}{\sum_{a \in \mathcal{K}} c_t(a) \hat{x}_t^{(q_l^*)*}(a)} \in [\alpha^m, \alpha^{m+1}] \right) \cdot \sum_{a \in \mathcal{K}} r^{(l)}(a) x_l^*(a)$$

$$\leqslant \sum_{t \in \mathcal{J}_l} \mathbf{1} \left( \frac{\sum_{a \in \mathcal{K}} r^{(l)}(a) x_l^*(a)}{\sum_{a \in \mathcal{K}} c^{(l)}(a) x_l^*(a)} \in [\alpha^{m-3/2}, \alpha^{m+5/2}] \right) \cdot \sum_{a \in \mathcal{K}} r^{(l)}(a) x_l^*(a)$$

$$= \sum_{t \in \mathcal{J}_l} \mathbf{1} \left( \frac{\sum_{a \in \mathcal{K}} r^{(l)}(a) x_l^*(a)}{\sum_{a \in \mathcal{K}} c^{(l)}(a) x_l^*(a)} \in [\alpha^{m-3/2}, \alpha^{m+5/2}] \right) \cdot \frac{\sum_{a \in \mathcal{K}} r^{(l)}(a) x_l^*(a)}{\sum_{a \in \mathcal{K}} c^{(l)}(a) x_l^*(a)} \cdot \sum_{a \in \mathcal{K}} c^{(l)}(a) x_l^*(a)$$

$$\leqslant \alpha^{m+5/2} \cdot \sum_{t \in \mathcal{J}_l} \sum_{a \in \mathcal{K}} c^{(l)}(a) x_l^*(a). \tag{62}$$

For $l \in \mathcal{L}$ such that $q_l^* > 0$, by (15) we have

$$\sum_{a \in \mathcal{K}} \hat{c}_t(a) \hat{x}_t^{(q_l^*)*}(a) \leqslant \eta_{\min} \cdot \alpha^{q_l^*} \leqslant \alpha \sum_{a \in \mathcal{K}} c^{(l)}(a) x_l^*(a).$$

In this case,

$$\sum_{t \in \mathcal{J}_l} \mathbf{1}(m_t^{(l)*} = m) \cdot \sum_{a \in \mathcal{K}} r^{(l)}(a) x_l^*(a)$$

$$\leqslant \alpha \cdot \sum_{t \in \mathcal{J}_l} \mathbf{1}(m_t^{(l)*} = m) \cdot \sum_{a \in \mathcal{K}} \hat{r}_t(a) \hat{x}_t^{(q_l^*)*}(a) \tag{63}$$

$$= \alpha \cdot \mathbf{1}(m_t^{(l)*} = m) \cdot \frac{\sum_{a \in \mathcal{K}} \hat{r}_t(a) \hat{x}_t^{(q_l^*)*}(a)}{\sum_{a \in \mathcal{K}} \hat{c}_t(a) \hat{x}_t^{(q_l^*)*}(a)} \cdot \sum_{a \in \mathcal{K}} \hat{c}_t(a) \hat{x}_t^{(q_l^*)*}(a)$$

$$\leqslant \alpha^{m+1} \cdot \sum_{t \in \mathcal{J}_l} \sum_{a \in \mathcal{K}} \hat{c}_t(a) \hat{x}_t^{(q_l^*)*}(a)$$

$$\leqslant \alpha^{m+2} \cdot \sum_{t \in \mathcal{J}_l} \sum_{a \in \mathcal{K}} c^{(l)}(a) x_l^*(a). \tag{64}$$

Putting together (62) and (64) we have

$$\sum_{l \in \mathcal{L}} \sum_{t \in \mathcal{J}_l} \mathbf{1}(m_t^{(l)*} = m) \cdot \sum_{a \in \mathcal{K}} r^{(l)}(a) x_l^*(a) \leqslant \alpha^{m+5/2} \cdot \sum_{l \in \mathcal{L}} \sum_{t \in \mathcal{J}_l} \sum_{a \in \mathcal{K}} c^{(l)}(a) x_l^*(a) \leqslant \alpha^{m+5/2} \cdot B.$$

Finally, let us combine the two cases where $m \in \mathcal{M}_1$ and $m \in \mathcal{M}_2$. If $\sum_{m \in \mathcal{M}_1} \sum_{l \in \mathcal{L}} \sum_{t \in \mathcal{J}_l} \mathbf{1}(m_t^{(l)*} = m) \cdot \sum_{a \in \mathcal{K}} r^{(l)}(a) x_l^*(a) \leqslant \tilde{O}(\sqrt{L|\mathcal{K}|NT})$, then (53) $= 0$ and we have

$$\mathrm{H}(1) \geqslant \frac{1}{5} \cdot \frac{\sum_{m \in \mathcal{M}_2} \sum_{l \in \mathcal{L}} \sum_{t \in (\bigcup_{n=-M}^{M-1} \tilde{\mathcal{T}}^{1(n)}) \cap \mathcal{J}_l} \mathbf{1}(m_t^{(l)*} = m) \cdot \sum_{a \in \mathcal{K}} r_t(a) \hat{x}_t^{(q_t)*}(a)}{\sum_{m=-M}^{M-1} \sum_{l \in \mathcal{L}} \sum_{t \in \mathcal{J}_l} \mathbf{1}(m_t^{(l)*} = m) \cdot \sum_{a \in \mathcal{K}} r^{(l)}(a) x_l^*(a)}$$

$$\geqslant \frac{1}{5} \frac{\sum_{m \in \mathcal{M}_2} \sum_{l \in \mathcal{L}} \sum_{t \in (\bigcup_{n=-M}^{M-1} \tilde{\mathcal{T}}^{1(n)}) \cap \mathcal{J}_l} \mathbf{1}(m_t^{(l)*} = m) \cdot \sum_{a \in \mathcal{K}} r_t(a) \hat{x}_t^{(q_t)*}(a)}{\sum_{m \in \mathcal{M}_2} \sum_{l \in \mathcal{L}} \sum_{t \in \mathcal{J}_l} \mathbf{1}(m_t^{(l)*} = m) \cdot \sum_{a \in \mathcal{K}} r^{(l)}(a) x_l^*(a)} \cdot \frac{\mathrm{opt(FA)} - \tilde{O}(\sqrt{L|\mathcal{K}|NT})}{\mathrm{opt(FA)}}$$

$$\geqslant \frac{1 - o(1)}{10\alpha^4 M} \cdot \frac{\mathrm{opt(FA)} - \tilde{O}(\sqrt{L|\mathcal{K}|NT})}{\mathrm{opt(FA)}}. \tag{65}$$

If $\sum_{m \in \mathcal{M}_1} \sum_{l \in \mathcal{L}} \sum_{t \in \mathcal{J}_l} \mathbf{1}(m_t^{(l)*} = m) \cdot \sum_{a \in \mathcal{K}} r^{(l)}(a) x_l^*(a) \geqslant \tilde{\Omega}(\sqrt{L|\mathcal{K}|NT})$, then (53) $= (1 - o(1))/(\alpha(M+1))$. Then

$$\mathrm{H}(1) \geqslant \min \left\{ \frac{1 - o(1)}{5\alpha(M+1)}, \frac{1 - o(1)}{10\alpha^4 M} \right\} \geqslant \frac{1 - o(1)}{10\alpha^4 M}. \tag{66}$$

Therefore, we conclude that

$$H(1) \geqslant \frac{1 - o(1)}{10\alpha^4 M} \cdot \frac{\text{opt(FA)} - \tilde{O}(\sqrt{L|\mathcal{K}|NT})}{\text{opt(FA)}}$$

with probability at least $1 - 2|\mathcal{K}|(ML + T)\delta$.

### C.4.2 Bounding H(2)

H(2) is to bound the stochastic reward achieved by randomized decision and the expected reward. In IRES-CM, for $t \in \bigcup_{m=-M}^{M-1} \tilde{\mathcal{T}}^{1(m)}$, we have $a_t \sim \hat{x}_t^{(q_t)*}$, and hence $\mathbb{E}[R_t(a_t)|\mathcal{F}_{t-1}] = \sum_{a \in \mathcal{K}} r_t(a)\hat{x}_t^{(q_t)*}(a)$. Then by Lemma B.1, for any $\delta \in (0, 1)$, with probability at least $1 - \delta$, we have

$$\sum_{t \in \bigcup_{m=-M}^{M-1} \tilde{\mathcal{T}}^{1(m)}} R_t(a_t) \in$$

$$\left[ \left( 1 - \sqrt{\frac{2}{\sum_{l \in \mathcal{L}} \sum_{t \in (\bigcup_{n=-M}^{M-1} \tilde{\mathcal{T}}^{1(n)}) \cap \tilde{\mathcal{J}}_l} \sum_{a \in \mathcal{K}} r_t(a)\hat{x}_t^{(q_t)*}(a)} \log\left(\frac{2}{\delta}\right)} \right) \cdot \sum_{l \in \mathcal{L}} \sum_{t \in (\bigcup_{n=-M}^{M-1} \tilde{\mathcal{T}}^{1(n)}) \cap \tilde{\mathcal{J}}_l} \sum_{a \in \mathcal{K}} r_t(a)\hat{x}_t^{(q_t)*}(a), \right.$$

$$\left. \left( 1 + \sqrt{\frac{3}{\sum_{l \in \mathcal{L}} \sum_{t \in (\bigcup_{n=-M}^{M-1} \tilde{\mathcal{T}}^{1(n)}) \cap \tilde{\mathcal{J}}_l} \sum_{a \in \mathcal{K}} r_t(a)\hat{x}_t^{(q_t)*}(a)} \log\left(\frac{2}{\delta}\right)} \right) \cdot \sum_{l \in \mathcal{L}} \sum_{t \in (\bigcup_{n=-M}^{M-1} \tilde{\mathcal{T}}^{1(n)}) \cap \tilde{\mathcal{J}}_l} \sum_{a \in \mathcal{K}} r_t(a)\hat{x}_t^{(q_t)*}(a) \right]$$

$$= \left[ \left( 1 - \sqrt{\frac{2}{H(1) \cdot \text{opt(FA)}} \log\left(\frac{2}{\delta}\right)} \right) \cdot H(1) \cdot \text{opt(FA)}, \left( 1 + \sqrt{\frac{3}{H(1) \cdot \text{opt(FA)}} \log\left(\frac{2}{\delta}\right)} \right) \cdot H(1) \cdot \text{opt(FA)} \right].$$

Since $H(1) \cdot \text{opt(FA)} \geqslant (1 - o(1)) \cdot (\text{opt(FA)} - \tilde{O}(\sqrt{L|\mathcal{K}|NT}))/(10\alpha^4 M) \geqslant \Omega(\text{opt(FA)}) \geqslant \tilde{\Omega}(\sqrt{L|\mathcal{K}|NT})$. Then it is evident that $H(2) \geqslant 1 - o(1)$ with probability at least $1 - \delta$.

### C.5 Proof of Lemma 2.3

#### C.5.1 Proof of part (a)

The horizon $\mathcal{T}$ is partitioned into $L = T/B$ pieces with equal length $B$. We consider $L$ instances with two arms $\mathcal{K} = \{1\}$ and $a_{\text{null}}$, and instance $n$ happen with probability $p_n$. All instances have deterministic outcomes, and they share the same consumption model $C_t(1) = 1$ for all $t \in \mathcal{T}$. Their reward functions are:

$$\text{Instance 1: } R^{(1)}(1) = \left( \underbrace{\alpha^{-L}, \ldots, \alpha^{-L}}_{\text{Piece 1}}, \underbrace{\alpha^{-L+1}, \ldots, \alpha^{-L+1}}_{\text{Piece 2}}, \ldots, \underbrace{\alpha^{-1}, \ldots, \alpha^{-1}}_{\text{Piece } L} \right),$$

$$\text{Instance 2: } R^{(2)}(1) = \left( \underbrace{\alpha^{-L}, \ldots, \alpha^{-L}}_{\text{Piece 1}}, \underbrace{\alpha^{-L+1}, \ldots, \alpha^{-L+1}}_{\text{Piece 2}}, \ldots, \underbrace{0, \ldots, 0}_{\text{Piece } L} \right),$$

$$\cdots$$

$$\text{Instance } L: R^{(L)}(1) = \left( \underbrace{\alpha^{-L}, \ldots, \alpha^{-L}}_{\text{Piece 1}}, \underbrace{0, \ldots, 0}_{\text{Piece 2}}, \ldots, \underbrace{0, \ldots, 0}_{\text{Piece } L} \right).$$

Denote $\text{FA}^{(n)}$ as the FA for instance $n \in \{1, \ldots, L\}$. It is clear that $\text{opt(FA}^{(n)}) = B\alpha^{-n}$. Recall $X_t(1) = \mathbf{1}(\text{Pull 1 in round } t)$. By the Yao's principle Yao (1977), the competitive ratio of any online algorithm is at most

$$\sum_{n=1}^{L} p_n \cdot \frac{\mathbb{E}^{(n)}[\sum_{t=1}^{T} R_t^{(n)}(1)X_t(1)]}{\text{opt(FA}^{(n)})}, \tag{67}$$

for any $p_n \geqslant 0$ with $\sum_{n=1}^{L} p_n = 1$. The expectation $\mathbb{E}^{(n)}$ is over the randomness in $X_t$ in instance $n$. The instances are crafted such that during piece $j \in \{1, \ldots, L\}$, it is impossible to distinguish among instances $j, \ldots, L$, meaning that the quantity $B_j^{(n)} = \mathbb{E}^{(n)}[\sum_{t \in \text{piece } j} X_t(1)]$ for $n \in \{j, \ldots, L\}$ are all identical, and equal to a common value $B_j$. Thus, for instance $n \in \{1, \ldots, L\}$ we have $\mathbb{E}^{(n)}[\sum_{t=1}^{T} R_t^{(n)}(1) X_t(1)] \leqslant \sum_{j=n}^{L} B_j \alpha^{-j}$. Consequently,

$$(69) \leqslant \sum_{n=1}^{L} p_n \frac{\sum_{j=n}^{L} B_j \cdot \alpha^{-j}}{B \cdot \alpha^{-n}} \leqslant \sum_{j=1}^{L} \frac{B_j}{B} \sum_{n=1}^{j} p_n \cdot \alpha^{n-j}.$$

By defining $p_1 = \frac{1}{L(1-1/\alpha)+1/\alpha} = \left(1 - \frac{1}{\alpha}\right) p_n$ for $n = 2, \ldots, L$, we have for every $j = 1, \ldots, L$,

$$\sum_{n=1}^{j} p_n \cdot \alpha^{n-j} \leqslant \frac{1}{L(1 - 1/\alpha) + 1/\alpha}$$

leading to

$$\sum_{j=1}^{L} \frac{B_j}{B} \sum_{n=1}^{j} p_n \cdot \alpha^{n-j} \leqslant \frac{1}{L(1 - 1/\alpha) + 1/\alpha}$$

by the inventory constraint $\sum_{j=1}^{L} B_j \leqslant B$ on instance 1. Since $L$ can generally be larger than $\log_\alpha(\eta_{\max}/\eta_{\min})$, we have shown that the CR can be significantly larger than $\log_\alpha(\eta_{\max}/\eta_{\min})$ when $\eta_{\min} = 0$.

### C.5.2 Proof of part (b)

While it is possible to derive a worse bound without $\eta_{\max}$ by setting it to its upper bound of 1, knowing the lower bound is essential for our algorithm's functionality. To show that it is necessary to know $\eta_{\min}$, we suppose the DM be provided with a looser lower range parameter $\tilde{\eta}_{\min} < \eta_{\min} \leqslant r_t(a), c_t(a) \; \forall a, t$, and show that it leads to sub-optimal CR.

**A general case construction.** We firstly construct a case with $N + 1$ instances when $\eta_{\min} = \beta^{-N}$ for some absolute constant $\beta > 1$. We consider $N + 1$ instances with two arms $\mathcal{K} = \{1\}$ and $a_{\text{null}}$, and instance $n$ happen with probability $p_n$. All instances have deterministic outcomes, and they share the same reward model $R_t(1) = 1$ for all $t$. Their consumption functions are:

$$\text{Instance 0: } C^{(0)}(1) = \left( \underbrace{1, \ldots, 1}_{\text{Piece 0: } B \text{ rounds}} \right),$$

$$\text{Instance 1: } C^{(1)}(1) = \left( \underbrace{1, \ldots, 1}_{\text{Piece 0: } B \text{ rounds}}, \underbrace{1/\beta, \ldots, 1/\beta}_{\text{Piece 1: } B \cdot \beta \text{ rounds}} \right),$$

$$\text{Instance 2: } C^{(2)}(1) = \left( \underbrace{1, \ldots, 1}_{\text{Piece 0: } B \text{ rounds}}, \underbrace{1/\beta, \ldots, 1/\beta}_{\text{Piece 1: } B \cdot \beta \text{ rounds}}, \underbrace{1/\beta^2, \ldots, 1/\beta^2}_{\text{Piece 2: } B \cdot \beta^2 \text{ rounds}} \right),$$

$$\ldots$$

$$\text{Instance } N : C^{(N)}(1) = \left( \underbrace{1, \ldots, 1}_{\text{Piece 0: } B \text{ rounds}}, \underbrace{1/\beta, \ldots, 1/\beta}_{\text{Piece 1: } B \cdot \beta \text{ rounds}}, \ldots, \underbrace{1/\beta^N, \ldots, 1/\beta^N}_{\text{Piece } N: B \cdot \beta^N \text{ rounds}} \right).$$

Denote $\text{FA}^{(n)}$ as the FA for instance $n \in \{0, \ldots, N\}$. It is clear that $\text{opt}(\text{FA}^{(n)}) = B\beta^n$. Recall $X_t(1) = \mathbf{1}(\text{Pull 1 in round } t)$. By the Yao's principle Yao (1977), the competitive ratio of an online algorithm is at most

$$\sum_{n=0}^{N} p_n \cdot \frac{\mathbb{E}^{(n)}[\sum_{t=1}^{T} R_t^{(n)}(1) X_t(1)]}{\text{opt}(\text{FA}^{(n)})}, \tag{68}$$

for any $p_n \geqslant 0$ with $\sum_{n=0}^{N} p_n = 1$. The expectation $\mathbb{E}^{(n)}$ is over the randomness in $X_t$ in instance $n$. The instances are crafted such that during piece $j \in \{0, \dots, N\}$, it is impossible to distinguish among instances $j, \dots, N$, meaning that the quantity $B_j^{(n)} = \mathbb{E}^{(n)}[\sum_{t \in \text{piece } j} C_t^{(n)} X_t(1)]$ for $n \in \{j, \dots, N\}$ are all identical, and equal to a common value $B_j$. Thus, for instance $n \in \{0, \dots, N\}$ we have $\mathbb{E}^{(n)}[\sum_{t=1}^{T} R_t^{(n)}(1) X_t(1)] \leqslant \sum_{j=n}^{N} B_j \beta^j$. Consequently,

$$(68) \leqslant \sum_{n=0}^{N} p_n \frac{\sum_{j=n}^{N} B_j \cdot \beta^j}{B \cdot \alpha^n} \leqslant \sum_{j=0}^{N} \frac{B_j}{B} \sum_{n=0}^{j} p_n \cdot \beta^{-n+j}.$$

By defining $p_0 = \frac{1}{2(N+1)(1-1/\beta)+1/\beta} = (1 - 1/\beta) p_n$ for $n = 1, \dots, N$, we have for every $j = 0, \dots, N$,

$$\sum_{n=0}^{j} p_n \cdot \beta^{-n+j} \leqslant \frac{1}{(N+1)(1-1/\beta) + 1/\beta}$$

leading to

$$\sum_{j=0}^{N} \frac{B_j}{B} \sum_{n=0}^{j} p_n \cdot \beta^{-n+j} \leqslant \frac{1}{(N+1)(1-1/\beta) + 1/\beta}$$

by the inventory constraint $\sum_{j=0}^{N} B_j \leqslant B$ on instance $N$. Therefore, when the DM is provided with information $\eta_{\min} = \beta^{-N}$ for any $N \in \mathbb{N}$, a CR$= \Theta(N)$ lower bound is derived based on $N + 1$ instances constructed above.

**Not knowing $\eta_{\min} = \beta^{-\Lambda}$ but knowing $\tilde{\eta}_{\min} = \beta^{-\kappa \cdot \Lambda}$.** We suppose the real underlying $\eta_{\min} = \beta^{-\Lambda}$ for some constant $\Lambda$, $\eta_{\max} = 1$, but the DM only has weaker prior information that $\tilde{\eta}_{\min} = \beta^{-\kappa \cdot \Lambda}$ ($\kappa > 1$ can be set arbitrarily large) and $\eta_{\max} = 1$. The pattern of $(R_t, C_t)$ follows the above case, and therefore, different $\eta_{\min}$ leads to different number of instances $N$. The DM only knows the number of instances is no larger than $\kappa \cdot \Lambda$. Then from the DM's point of view, the optimal CR she/he could derive is CR$= \Theta(\kappa \cdot \Lambda)$; while from the perspective of a clairvoyant who knows the real $\eta_{\min} = \beta^{-\Lambda}$, the optimal CR should be $\Theta(\Lambda)$.

We first show that given $\tilde{\eta}_{\min} = \beta^{-\kappa \cdot \Lambda}$, the DM will not benefit from tightening the value ranges by blindly guessing a value of $\eta_{\min}$. We suppose the DM blindly tightens the value range to $[\beta^{-(\kappa \cdot \Lambda - d)}, 1]$ for some $d \geqslant 1$, without knowing the real $\eta_{\min}$. Then he/she derives a CR lower bound with $N + 1 = \kappa \cdot \Lambda - d + 1$ instances based on the above construction. Then the DM can expect to achieve a total reward of

$$\sum_{t=1}^{T} R_t(1) X_t^{\text{TIGHT}}(1) = \frac{\sum_{n=0}^{N} p_n \cdot \text{opt}(\text{FA}^{\text{TIGHT}(n)})}{(N+1)(1-1/\beta) + 1/\beta} = \Theta\left(\frac{B \cdot \beta^{\kappa \cdot \Lambda - d}}{\kappa \cdot \Lambda - d}\right).$$

However, since the DM does not know the real $\eta_{\min}$, it is possible that in fact $\eta_{\min} = \beta^{-\kappa \cdot \Lambda}$. If this is indeed the case, the optimal reward can be as large as

$$\sum_{n=0}^{N} p_n \cdot \text{opt}(\text{FA}^{(n)}) = \Omega(B \cdot \beta^{\kappa \cdot \Lambda})$$

based on the above constructed $N = \kappa \cdot \Lambda$ instances. Hence, from the DM's perspective, she/he could achieve a sub-optimal CR of

$$\frac{\sum_{n=0}^{N} p_n \cdot \text{opt}(\text{FA}^{(n)})}{\sum_{t=1}^{T} R_t(1) X_t^{\text{TIGHT}}(1)} = \Omega\left(B \cdot \beta^{\kappa \cdot \Lambda} \cdot \frac{\kappa \cdot \Lambda - d}{B \cdot \beta^{\kappa \cdot \Lambda - d}}\right) = \Omega(\beta^d \cdot (\kappa \cdot \Lambda - d)),$$

if she/he blindly assume $\eta_{\min} = \beta^{-(\kappa \cdot \Lambda - d)}$. This is significantly worse than the optimal CR$= \Theta(\kappa \cdot \Lambda)$ (if in fact $\eta_{\min} = \beta^{-\kappa \cdot \Lambda}$). Thus, the DM has no motivation to assume a lower bound larger than the provided $\tilde{\eta}_{\min}$.

Therefore, the DM must derive a CR on the full range $[\beta^{\kappa \cdot \Lambda}, 1]$, which involves $N + 1 = \kappa \cdot \Lambda + 1$ instances as constructed above. Therefore the DM expects a reward of $\Theta(B \cdot \beta^{\kappa \cdot \Lambda} / (\kappa \cdot \Lambda))$. However, since in fact there are only $\Lambda + 1$ instances, the DM wastes all her/his resources reserved for instance $\Lambda + 2, \dots, \kappa \cdot \Lambda + 1$ and she/he can only achieve a reward of $O(B \cdot \beta^{\Lambda} / (\kappa \cdot \Lambda))$. Compared with the actual optimal reward $\Omega(B \cdot \beta^{\Lambda})$ with $\Lambda + 1$ instances, the DM achieves a sub-optimal CR of $\Omega(\kappa \cdot \Lambda)$. Since $\kappa$ can be arbitrarily large, the CR derived without correct knowledge of $\eta_{\min}$ is significantly worse than the optimal CR$= \Theta(\Lambda)$.

## C.6 Proof of Theorem 4.5

We prove Theorem 4.5 by considering $2\nu + 1$ instances, which share the same $\mathcal{K} = \{1\}$, $B \in \mathbb{Z}_{>0}$ and $T = B(2\nu + 1)$ (All instances have the null arm, as stipulated by our model definition). All instances have deterministic outcomes, and they share the same consumption model $C_t(1) = 1$ for all $t \in \mathcal{T}$. By contrast, they differ in the reward model. The horizon $\mathcal{T}$ is partitioned into $2\nu + 1$ pieces with equal length $B$. Their reward functions are:

$$\text{Instance } -\nu:\, R^{(-\nu)}(1) = \left( \underbrace{\alpha^{-2\nu}, \ldots, \alpha^{-2\nu}}_{\text{Piece } -\nu}, \underbrace{\alpha^{-2\nu+1}, \ldots, \alpha^{-2\nu+1}}_{\text{Piece } -\nu+1}, \ldots, \underbrace{\alpha^0, \ldots, \alpha^0}_{\text{Piece } \nu} \right),$$

$$\text{Instance } -\nu+1:\, R^{(-\nu+1)}(1) = \left( \underbrace{\alpha^{-2\nu}, \ldots, \alpha^{-2\nu}}_{\text{Piece } -\nu}, \underbrace{\alpha^{-2\nu+1}, \ldots, \alpha^{-2\nu+1}}_{\text{Piece } -\nu+1}, \ldots, \underbrace{\epsilon, \ldots, \epsilon}_{\text{Piece } \nu} \right),$$

$$\cdots$$

$$\text{Instance } \nu:\, R^{(\nu)}(1) = \left( \underbrace{\alpha^{-2\nu}, \ldots, \alpha^{-2\nu}}_{\text{Piece } -\nu}, \underbrace{\epsilon, \ldots, \epsilon}_{\text{Piece } -\nu+1}, \ldots, \underbrace{\epsilon, \ldots, \epsilon}_{\text{Piece } \nu} \right),$$

where $\epsilon = \alpha^{-3\nu}$. Denote $\text{FA}^{(n)}$ as the FA for instance $n \in \{-\nu, \ldots, \nu\}$. It is clear that $\text{opt}(\text{FA}^{(n)}) = B\alpha^{-\nu-n}$. Recall $X_t(1) = \mathbf{1}(\text{Pull 1 in round } t)$. By the Yao's principle Yao (1977), the competitive ratio of an online algorithm is at most

$$\sum_{n=-\nu}^{\nu} p_n \cdot \frac{\mathbb{E}^{(n)}[\sum_{t=1}^{T} R_t^{(n)}(1) X_t(1)]}{\text{opt}(\text{FA}^{(n)})}, \tag{69}$$

for any $p_n \geq 0$ with $\sum_{n=-\nu}^{\nu} p_n = 1$. The expectation $\mathbb{E}^{(n)}$ is over the randomness in $X_t$ in instance $n$. The instances are crafted such that during piece $j \in \{-\nu, \ldots, \nu\}$, it is impossible to distinguish among instances $-\nu, \ldots, -j$, meaning that the quantity $B_j^{(n)} = \mathbb{E}^{(n)}[\sum_{t \in \text{piece } j} X_t(1)]$ for $n \in \{-\nu, \ldots, -j\}$ are all identical, and equal to a common value $B_j$. Thus, for instance $n \in \{-\nu, \ldots, \nu\}$ we have $\mathbb{E}^{(n)}[\sum_{t=1}^{T} R_t^{(n)}(1) X_t(1)] \leq B\epsilon + \sum_{j=-\nu}^{-n} B_j \alpha^{j-\nu} \leq B \cdot \alpha^{-3\nu} + \sum_{j=-\nu}^{-n} B_j \alpha^{j-\nu}$. Consequently,

$$(69) \leq \sum_{n=-\nu}^{\nu} p_n \frac{B \cdot \alpha^{-3\nu} + \sum_{j=-\nu}^{-n} B_j \cdot \alpha^{j-\nu}}{B \cdot \alpha^{-\nu-n}} \leq \frac{1}{\alpha^{\nu} \cdot (1 - 1/\alpha)} + \sum_{j=-\nu}^{\nu} \frac{B_j}{B} \sum_{n=-\nu}^{-j} p_n \cdot \alpha^{j+n}.$$

By defining $p_{-\nu} = \frac{1}{2\nu(1-1/\alpha)+1/\alpha} = \left(1 - \frac{1}{\alpha}\right) p_n$ for $n = -\nu+1, \ldots, \nu$, we have for every $j = -\nu, \ldots, \nu$,

$$\sum_{n=-\nu}^{-j} p_n \cdot \alpha^{j+n} \leq \frac{1}{2\nu(1-1/\alpha) + 1/\alpha}$$

leading to

$$\sum_{j=-\nu}^{\nu} \frac{B_j}{B} \sum_{n=-\nu}^{-j} p_n \cdot \alpha^{j+n} \leq \frac{1}{2\nu(1-1/\alpha) + 1/\alpha}, \tag{70}$$

by the inventory constraint $\sum_{j=-\nu}^{\nu} B_j \leq B$. Since $\nu = \log_\alpha(\eta_{\max}/\eta_{\min})/3$, the Theorem is proved.

