# OpenReview forum: "Piecewise-Stationary Bandits with Knapsacks"
_NeurIPS.cc/2024/Conference — NeurIPS 2024 poster_

### Official Review · Reviewer_engA · 2024-07-12

**Soundness:** 3
**Presentation:** 3
**Contribution:** 2
**Rating:** 5
**Confidence:** 4

**Summary:**

The paper studies Bandits with Knapsack (BwK) under a piecewise stationary environment. For the online matching problem where the true reward is fully known in each time period, the paper obtains a $\Omega(1/\ln(\eta_{\max}/\eta_{\min}))$ where $\eta_{\min}$ and $\eta_{\max}$ are such that all rewards and resources are in $[\eta_{\min}, \eta_{\min}]$. The paper also studies the online learning problem and gives theoretical results.

**Strengths:**

1. The writing is good, and mostly easy-to-follow.
2. The work gives theoretical guarantees to both the online matching and the online learning problem.

**Weaknesses:**

I appreciate all the technical proofs in the paper. However, it seems the paper lacks enough technical novelty as well as strong theoretical results. For details, please see Questions.

**Questions:**

1. The ratio of $\ln(\eta_{\max}/\eta_{\min})$ seems to be weak. Imagine if $r_t=r$ and $c_t=c$ for all $t$, but $r$ and $c$ are dramatically different. It is reasonable to claim that the ratio approaches $1$ when the budget is not too low. Is it possible to refine the definition of $\eta$ so that it only concerns $r/c$ (similar to that in Zhou et al. (2008))?
2. In your contribution, you claim your performance guarantee is w.r.t. a dynamic benchmark, while existing adversarial BwK literature focus on the stationary benchmark. I think in many online matching papers the benchmark is also a dynamic one (see. e.g., Zhou et al. (2008)). Can you elaborate on this?
3. In the online matching part, the algorithm design is not entirely novel. From my perspective, the inventory reserving idea is a deterministic version of the initial randomized policy in Immorlica et al. (2019). It is also similar to the booking limit or nested booking limit in the online matching/booking problem (see, e.g, [1]). Another issue with regard to the inventory reserving design is that it is not adaptive enough and may waste some resources because the reservation policy is fixed at the beginning. Can you elaborate on the technical novelty in your policy design?
4. In the online learning problem, it seems you are essentially decreasing the competitive ratio in exchange for $\sqrt{T}$ regret. Is this necessary? And also how do you set $\alpha$?
5. There are very limited numerical experiments for the planning problem and there is no experiment for the learning problem. I think only comparing to Immorlica et al. (2019) is not convincing. The authors may also consider comparing their algorithms with that in Zhou et al. (2008),

[1] Ball, Michael O., and Maurice Queyranne. "Toward robust revenue management: Competitive analysis of online booking." Operations Research 57.4 (2009): 950-963.

---

> ### Author Rebuttal · Authors · 2024-08-06
>
> We are grateful for the reviewer's time and effort in evaluating our manuscript. We highly value your feedback and would like to address your concerns and questions point by point.
>
> 1. “The ratio of $\ln (\eta_{\max}/\eta_{\min})$ seems to be weak. Imagine if $r_t=r$ and $c_t=c$ for all $t$, but $r$ and $c$ are dramatically different. It is reasonable to claim that the ratio approaches $1$ when the budget is not too low. Is it possible to refine the definition of $\eta$ so that it only concerns $r/c$ (similar to that in Zhou et al. (2008))?
>
> Response: We are grateful for your keen observation. Indeed, we can refine the definition of $\eta$ so that it only concerns $r/c$. We will explain the change based on IRES in Section 3, and the same follows for IRES-CM in Section 4.
>
> By our orignal definition, in line 5 of Algorithm 1, we solve $\text{LP}(r^{(l)}, c^{(l)},\eta_{\min} \cdot \alpha^q)$ $\forall q \in \{0,1, \ldots, M\}$. In this step, each $\eta_{\min} \cdot \alpha^q$ is a guess of $B_l^{\star}/(t_l - t_{l-1})$. This is the only step we require $c^{(l)} \in [\eta_{\min}, \eta_{\max}]$, in order to show that a “correct” guess of $B_l^{\star}/(t_l - t_{l-1})$ leads to a “correct” guess of $ m^{\star}_l$ where $\text{Ratio}^{(l)\star} \in (\alpha^{m_l^{\star}}, \alpha^{m_l^{\star}+1}]$. All other steps only depend on the reward-consumption ratio. However, essentially line 5 only requires to output a guess of  an arbitrary decision $x_l$ such that
>
> \begin{align}
> \frac{\sum_{a \in \mathcal{K}} r^{(l)}(a) x_l(a)}{\sum_{a \in \mathcal{K}} c^{(l)}(a) x_l(a)} \in (\alpha^{m^{\star}_l}, \alpha^{m^{\star}_l+1}]
> \end{align}
>
> and $\sum_{a \in \mathcal{K}} c^{(l)}(a) x_l(a) \geq B_l^{\star}/(t_l - t_{l-1})$.
>
> In our revision, we re-define $\eta_{\min} = \min_{t,a} \frac{r_t(a)}{c_t(a)}, \eta_{\max} = \max_{t,a} \frac{r_t(a)}{c_t(a)}$, similar with $L$ and $U$ in Zhou et al. (2008) but considering multiple arms. Then $\text{Ratio}^{(l)\star} \in [\eta_{\min},\eta_{\max}]$. We define $M=\lceil\log_{\alpha}(\eta_{\max}/\eta_{\min})\rceil$ and partition $[\eta_{\min}, \eta_{\max}]$ into $M$ intervals $(\eta_{\min}\cdot \alpha^{m-1}, \eta_{\min} \cdot \alpha^m]$ where $m=1,\ldots,M$. For each stationary piece $l \in \mathcal{L}$, we denote $m_l^{\star} \in ${$0,\ldots, M-1$} as the interval such that $\text{Ratio}^{(l)\star} \in (\eta_{\min} \cdot \alpha^{m_l^{\star}}, \eta_{\min} \cdot \alpha^{m_l^{\star}+1}]$. Most of our paper remains the same, with only $M$ intervals and $m_l^{\star}$ replaced by their new definitions.
>
> The only major change happens in line 5 of Algorithm 1. Specifically, we change it into solving the following LPs for all $q \in ${$0, \ldots, M-1$} for the optimal solution $x_l^{(q)\star}$:
> \begin{align}
> \max \sum_{a \in \mathcal{K}} c^{(l)}(a) x_l^{(q)}(a)
> \end{align}
> \begin{align}
> \text{s.t.} \sum_{a \in \mathcal{K}} r^{(l)}(a) x_l^{(q)}(a) \geq \alpha^{q} \cdot \sum_{a \in \mathcal{K}} c^{(l)}(a) x_l^{(q)}(a)
> \end{align}
> \begin{align}
> \sum_{a \in \mathcal{K}} r^{(l)}(a) x_l^{(q)}(a) \leq \alpha^{q+1} \cdot \sum_{a \in \mathcal{K}} c^{(l)}(a) x_l^{(q)}(a)
> \end{align}
> \begin{align}
> \sum_{a \in \mathcal{K}} x_l^{(q)}(a)\leq 1
> \end{align}
> \begin{align}
> x_l^{(q)}(a) \geq 0       \qquad \forall a \in \mathcal{K}.
> \end{align}
>
> It can be seen that the modified line 5 of Algorithm 1 realizes its essential function. Then rest of the proofs remain valid (with updated definitions). We maintain a CR of $O(\log(\eta_{\max}/\eta_{\min}))$, with $\eta$ only depending on $r/c$ . We have modified the manuscript accordingly.

---

> ### Author Response · Authors · 2024-08-06
>
> 2. “In your contribution, you claim your performance guarantee is w.r.t. a dynamic benchmark, while existing adversarial BwK literature focus on the stationary benchmark. I think in many online matching papers the benchmark is also a dynamic one (see. e.g., Zhou et al. (2008)). Can you elaborate on this?”
>
> Response: When we mention “while existing adversarial BwK literature focus on the stationary benchmark,” we are focusing solely on research works that consider the **bandit feedback** setting. That is, the DM only observes the outcomes $(R_t(A_t), C_t(A_t))$ **after** pulling arm $A_t$, which is studied in our Section 4. The bandit feedback setting is more stringent than the **full feedback setting** in online matching, which require observing the actual values of $(R_t(a), C_t(a))$ for all $a \in \mathcal{K}$ **before** choosing $A_t$.
>
> Our intuitive discussions in Section 3 has the same full feedback setting as online matching papers (which certainly resonates with your comment), for example [Karp et al. (1990), Mehta et al. (2007), Zhou et al. (2008)], which crucially require knowing $(R_t(a), C_t(a))$ in crafting their potential function [Mehta et al. (2007), Zhou et al. (2008)], or algorithm gadget [Karp et al. (1990)], which cannot be readily generalized to the bandit setting. By contrast, we take a different approach in algorithm design in Section 3, which circumvent the difficulty is transiting from full feedback to bandit feedback setting. The natural generalization from full to bandit feedback shown in Section 4 is one of our core contributions. To this end, we emphasize that bandit feedback does hinder optimization, as illustrated by how the regret term in the bandit setting (Theorem 4.2) degrades with an increasing $L$, the number of change points.
>
> Regarding the benchmark considered by Zhou et al. (2008), it is indeed dynamic. However, it is a **best single arm** benchmark allowing pulling a single arm in each round; while our opt(FA) is a **best distribution over arms** benchmark. In fact, in the bandit-feedback stationary outcome setting, Badanidiyuru et al. (2018) consider a best distribution over arms benchmark, which is similar with our opt(FA). They show, in their Appendix A, that it requires optimizing over the set of probability distributions on the $K$ arms to get the optimal regret, while pulling a best single arm yields significantly worse regret (which could noticably affect the achievable CR). It is not hard to show that in the picewise-stationary setting, this is also the case. Thus, our benchmark is **strictly stronger** than Zhou et al. (2008).
>
> To conclude, our benchmark is strictly stronger than Zhou et al. (2008) and we consider a more difficult setting than online matching in Section 4. We appreciate the reviewer’s comment and will cite and compare with the online matching literature in our revised manuscript.
>
> Reference
>
> [1] Karp, Vazirani, Vazirani, An Optimal Algorithm for On-line Bipartite Matching, STOC 1990
>
> [2] Mehta, Saberi, Vazirani, Vazirani, AdWords and Generalized On-line Matching, JACM 2007
>
> [3] Badanidiyuru, A., Kleinberg, R., & Slivkins, A. (2018). Bandits with knapsacks. Journal of the ACM (JACM), 65(3), 1-55.

---

> ### Author Response · Authors · 2024-08-07
>
> 3. “In the online matching part, the algorithm design is not entirely novel.... Another issue with regard to the inventory reserving design is that it is not adaptive enough and may waste some resources because the reservation policy is fixed at the beginning. Can you elaborate on the technical novelty in your policy design?”
>
> Response: We appreciate the reviewer’s comment, but we argue that it is not accurate to say our inventory reserving idea is a deterministic version of other works such as Immorlica et al. (2019). While some papers consider different inventory reservation strategies, ours offers a novel perspective on the problem, and its simplicity and intuitiveness enhance our contribution. Our approach splits the reward into two terms: the reward-consumption ratio multiplied by the resource consumption on each stationary piece $l$ (see Section 2.3). By reserving a certain amount of inventory for each ratio interval, our task simplifies to guessing the optimal reward-consumption ratio interval in each round. Our idea and corresponding strategy are crucially different from existing literature. Moreover, the technical details of our proofs are not at all similar with any existing paper.
>
> Immorlica et al. (2019)’s inventory reservation strategy depends solely on the cumulated reward (see the definition of their $\hat{g}$ in Algorithm 3), without considering the reward-consumption ratio, using traditional gradient descent-based algorithms. Our experiments demonstrate that their inventory reservation strategy is significantly more conservative than ours in the piecewise-stationary setting.  Ball and Queyranne (2009) study a rather restrictive setting with two customer classes, prior knowledge of each class's fare, and fixed resource consumption of $1$. This is a very simple setting, which allows them to consider a strategy depending only on the ratio between each fare class. If our strategy is perceived as a similar version of existing methods, then by that logic, all inventory reservation strategies could be viewed as fundamentally similar.
>
> We do agree that our algorithm is not as adaptive as some best-of-both-world algorithms, but it does provide a near-optimal competitive ratio in our non-stationarity setting. Although the process of guessing ratio intervals via LPs in a round-robin manner may seem primitive, it leaves room for further refinement which may lead to less resource waste. Our experiments show that although our algorithm wastes some resources, it is less conservative in inventory reservation compared to Immorlica et al. (2019)’s adversarial algorithm, leading to better performance in our piecewise-stationary setting. We also believe that our piecewise-stationary setting is more realistic than stationary/adversarial/non-stationary with bounded global variation settings in practice, and thus worth studying.
>
> To sum up, given that we introduce a novel and intuitive perspective of the piecewise-stationary Bwk, develops new inventory reservation algorithms and proof techniques, and achieves a near-optimal performance guarantee (compared to the best distribution over arms dynamic benchmark), we believe our contribution is solid.

---

> ### Author Response · Authors · 2024-08-07
>
> 4. "In the online learning problem, it seems you are essentially decreasing the competitive ratio in exchange for $\sqrt{T}$ regret. Is this necessary? And also how do you set $\alpha$?"
>
> Response: Thank you for the insightful concern. We do not yet know what is the minimal decrease in competitive ratio in exchange for a sublinear-in-$T$ regret. We choose a simpler way to explore so that we do not need to detect the exact change points. There could potentially be more refined exploration approaches that lead to better coefficients in the competitive ratio, while maintaining a worse regret. We will keep exploring in our future research. Regarding $\alpha$, it can be set as any constant $> 1$, such as $2$ or $e$.
>
> 5. "There are very limited numerical experiments for the planning problem and there is no experiment for the learning problem. I think only comparing to Immorlica et al. (2019) is not convincing. The authors may also consider comparing their algorithms with that in Zhou et al. (2008)."
>
> Response: Thanks for the reviewer’s comment! Given that our paper is primarily theoretical, we conduct only demonstrative experiments on smaller datasets. We clarify that our experiments focus on a bandit feedback setting, which indeed involves a learning problem. The underlying rewards and resource consumption are set to be deterministic for convenience, but the algorithm does not know they are deterministic. We apologize for the misleading typo in line 632 where "known" should have been "unknown", and we have correct it. Hence we believe comparing our proposed IRES-CM and the existing benchmark by Immorlica et al. (2019) is appropriate. We will provide a study on the full feedback setting, and compare with Zhou et al. (2008) which only works with full feedback. This supplements our main study with bandit feedback.
>
> Finally, we sincerely thank the reviewer for the careful inspection and insightful concerns! Your suggestions have significantly helped us improve the quality of our paper. We genuinely hope you could re-evaluate our contributions given our clarifications. We are happy and open to any further discussion.

---

> > ### Comment · Reviewer_engA · 2024-08-12
> >
> > I would like to thank the authors for the very detailed response. My concerns have been moderately addressed and thus I have raised my score. While I very much appreciate authors' efforts and clarifications, I think the paper may require a significant re-structure by adding more results (theoretical & numerical) and discussions in the next iteration.

---

> > > ### Author Response · Authors · 2024-08-12
> > >
> > > Thank you for the feedback! We will surely add more theoretical results to support our revision, and we will also add more discussions distinguishing our work from existing literature that you kindly mentioned. If granted an additional page in the main text, we plan to incorporate the numerical results currently in the appendix, and add further numerical experiments and discussions in the next iteration. At present, we have maximized our page limit to effectively present our results and novel ideas, so we put the numerical results in the appendix.

---

> ### Author Response · Authors · 2024-08-14
> **Numerical experiments regarding Zhou et al.'s algorithm and the bandit learning setting**
>
> Dear reviewer, pardon our late supplementation. We would like to provide some numerical results regarding the Zhou et al. and the bandit setting that you mentioned.
>
> We present a two-piece stationary illustrative case here. We let the rewards and resource consumption in all rounds be uniformly distributed within a $[-0.2,+0.2]$ range regarding their mean values. We let $B=9360$ and $T=20000$ as in our original manuscript. We remark that Zhou et al. require observing $(R_t, C_t)$ in each round before making decisions, which is a way too strong requirement. Hence, we assume that Zhou et al. observe $(r_t, c_t)$ instead of $(R_t, C_t)$ before making decisions in each round, which is still a stronger assumption than the bandit setting considered by us and Immorlica et al.
>
> In the following tables, we take the average rewards over 10 experiments for each algorithm and present the cumulative rewards in a multiple of every 1000 rounds.
>
> We first consider the $(r_t, c_t)$ values in our original manuscript, where we let $r^{(1)}(1)=r^{(1)}(2)=0.5, c^{(1)}(1)=c^{(1)}(2)=1$, $r^{(2)}(1)=1, r^{(2)}(2)=0.5, c^{(2)}(1)=0.5, c^{(2)}(2)=1$. In this case, the optimal solution of the benchmark FA chooses a single arm on each stationary piece, and Zhou el al. outperform both Immorlica et al. and our IRES-CM.
>
> | t                | 1000   | 2000   | 3000    | 4000    | 5000    | 6000    | 7000    | 8000    | 9000    | 10000   | 11000   | 12000   | 13000   | 14000   | 15000   | 16000   | 17000   | 18000   | 19000   | 20000   |
> |------------------|--------|--------|---------|---------|---------|---------|---------|---------|---------|---------|---------|---------|---------|---------|---------|---------|---------|---------|---------|---------|
> | Immorlica et al. | 409.42 | 668.89 | 868.46  | 959.76  | 1004.28 | 1047.26 | 1081.50 | 1338.21 | 1379.11 | 1415.21 | 1486.26 | 1556.22 | 1611.23 | 1680.19 | 1743.14 | 1810.03 | 1873.71 | 1938.27 | 2001.66 | 2066.87 |
> | Zhou et al.      | 498.39 | 998.42 | 1499.79 | 1998.63 | 2100.68 | 2100.68 | 2100.68 | 2100.68 | 2100.68 | 2100.68 | 3100.24 | 4099.27 | 5098.96 | 6096.99 | 7099.31 | 8099.78 | 9025.09 | 9025.09 | 9025.09 | 9025.09 |
> | IRES-CM          | 468.43 | 866.61 | 1294.79 | 1680.58 | 2060.31 | 2471.52 | 2863.14 | 3252.99 | 3544.64 | 3647.43 | 4580.54 | 5579.57 | 6433.62 | 7148.54 | 7148.54 | 7148.54 | 7148.54 | 7148.54 | 7148.54 | 7148.54 |
>
> We next consider a case with a slight change on $(r_t, c_t)$ on the second stationary piece. Specifically, we let $r^{(1)}(1)=r^{(1)}(2)=0.5, c^{(1)}(1)=c^{(1)}(2)=1$, $r^{(2)}(1)=0.5, r^{(2)}(2)=1, c^{(2)}(1)=0.5, c^{(2)}(2)=1$. In this case, the optimal solution of the benchmark FA chooses a distribution over arms on the second stationary piece, where $x^*_{2}(1)=0.128, x^*_{2}(2)=0.872$. As shown, in this case our IRES-CM outperforms both Immorlica et al. and Zhou et al. This is consistent with the theoretical results that Zhou et al. achieve sub-optimal rewards compared with a **best distribution over arms** benchmark.
>
> | t                | 1000   | 2000   | 3000    | 4000    | 5000    | 6000    | 7000    | 8000    | 9000    | 10000   | 11000   | 12000   | 13000   | 14000   | 15000   | 16000   | 17000   | 18000   | 19000   | 20000   |
> |------------------|--------|--------|---------|---------|---------|---------|---------|---------|---------|---------|---------|---------|---------|---------|---------|---------|---------|---------|---------|---------|
> | Immorlica et al. | 434.69 | 656.97 | 909.24  | 943.53  | 988.01  | 1030.39 | 1064.42 | 1101.87 | 1144.65 | 1181.05 | 1248.60 | 1314.08 | 1372.59 | 1437.17 | 1508.36 | 1572.78 | 1631.64 | 1691.74 | 1751.38 | 1826.37 |
> | Zhou et al.      | 494.79 | 988.91 | 1488.38 | 1990.28 | 2444.41 | 2444.41 | 2444.41 | 2444.41 | 2444.41 | 2444.41 | 3442.66 | 4442.90 | 4465.25 | 4465.25 | 4465.25 | 4465.25 | 4465.25 | 4465.25 | 4465.25 | 4465.25 |
> | IRES-CM          | 442.14 | 850.82 | 1273.24 | 1721.91 | 2186.27 | 2636.42 | 3049.17 | 3467.15 | 3884.67 | 4339.88 | 4772.74 | 5021.60 | 5021.60 | 5021.60 | 5021.60 | 5021.60 | 5021.60 | 5021.60 | 5021.60 | 5021.60 |
>
> We believe the above results further validates the strength of our algorithm. We will supplement the figure format results of the above case and more large-scale cases in our revision. We hope this supplementation further addresses your concern, and thank you for your valuable comments!

---

### Official Review · Reviewer_betX · 2024-07-12

**Soundness:** 3
**Presentation:** 4
**Contribution:** 3
**Rating:** 6
**Confidence:** 3

**Summary:**

The paper studies the bandits with knapsacks problem in a piecewise-stationary environment. The paper proposes an algorithm guaranteering a near-optimal competitive ratio for the problem. The guarantees hold wrt a dynamic benchmark which is stronger than the standard stationary benchmark employed in adversarial BwK settings.

**Strengths:**

The setup is interesting and well motivated, and it fits nicely within the line of works trying to bridge between fully stochastic and fully adversarial environments. Moreover, the proposed algorithm provides some interesting insights on the problem and a new perspective with respect to the standard LagrangeBwK approach of Immorlica et al.

**Weaknesses:**

See questions.

**Questions:**

Is the overhead of solving the LP at each iteration significant in your experiments? For example, how does it compare to "lighter" per-iteration updates, such as the primal-dual approach by Immorlica et al.?

**Limitations:**

n.a.

---

> ### Author Rebuttal · Authors · 2024-08-06
>
> We are genuinely grateful to the reviewer for dedicating time and effort to evaluating our paper. We are glad to address your question.
>
> In fact, the computational load of our algorithm is noticeably lighter than Immorlica et al. (2019). We solve $\text{LP}(r^{(l)}, c^{(l)},\eta_{\min} \cdot \alpha^q)$ for each $q \in ${$0,1, \ldots, M$} in each iteration, which leads to no more than $(M+1)T = \lceil \log(\eta_{\max}/\eta_{\min}) \rceil T$ LPs solved over the planning horizon. Immorlica et al. (2019) needs to solve $\hat{g}(t) = \max_{\tau \in [t]} \tau \cdot \text{opt}(\text{LP}(\bar{M}^{\text{ips}}_{\tau}, B, \tau))$ in each iteration (see their Algorithm 3 on page 20), which involves solving $t$ LPs in each round $t$. Thus, their primal-dual algorithm requires solving $T(T+1)/2$ LPs over the planning horizon. Since our $M$ is a logarithmic factor regarding the value range of $r_t(a), c_t(a) \in [0,1]$ for all $t,a$, we typically have $M<<T$.
>
> Our numerical experiments consistently exhibit computational time aligned with the theoretical complexity, with our algorithm being around 5-6 times faster than that of Immorlica et al. (2019). However, given that our paper is primarily theoretical, we conducted demonstrative experiments on smaller datasets where the overhead of solving LPs is not prominent.
>
> We do agree that solving $(M+1)T$ LPs could still be expensive, and there could potentially be lighter approaches regarding per-iteration updates. Concerning non-stationary Bwk, the following papers do not require solving more than one LP per iteration: Fikioris and Tardos (2023) study an approximate stationary setting; Castiglioni et al. (2022a) study a large budget setting; Liu et al. (2022) focus on a bounded global variation setting. Their algorithms are lighter than ours in terms of computational buren, but their settings are significantly different from ours. Therefore, these works are not directly comparable with ours.
>
> Finally, we thank the reviewer for the valuable point on computational complexity. We will continue to work on reducing the computational load of our algorithm. We are happy and open to discussing any further questions.
>
> References
>
> [1] Fikioris, G., & Tardos, É. (2023, July). Approximately stationary bandits with knapsacks. In The Thirty Sixth Annual Conference on Learning Theory (pp. 3758-3782). PMLR.
>
> [2] Castiglioni, M., Celli, A., & Kroer, C. (2022, June). Online learning with knapsacks: the best of both worlds. In International Conference on Machine Learning (pp. 2767-2783). PMLR.
>
> [3] Liu, S., Jiang, J., & Li, X. (2022). Non-stationary bandits with knapsacks. Advances in Neural Information Processing Systems, 35, 16522-16532.

---

> ### Author Response · Authors · 2024-08-13
> **More on improving the computational overhead**
>
> (We apologize for writing $\hat{r}, \hat{c}$ as $r,c$ in this comment. Complicated math notation cannot show well.)
>
> Pardon our supplement discussion. In fact, the computational load of our algorithm can be easily reduced to solving no more than $1$ LP per iteration. Instead of solving LP$(r_t, c_t, \eta_{\min} \cdot \alpha^{q})$ for all $q \in${$0,1, \ldots, M-1$} in each round (line 6 of Algorithm 2), we can solve a single LP$(r_t, c_t,\eta_{\min} \cdot \alpha^{q_t})$ after deciding the $q_t$ value in line 16. By doing so, we retain the same performance guarantee as the original manuscript.
>
> In our original manuscript (and our rebuttal), we didn't write down Algorithm 2 in the most computationally efficient way. Since we mainly focused on illustrating the learning process naturally, we chose to present Algorithm 2 in a way that shows how it naturally generalizes from Algorithm 1. In the deterministic outcome setting (Algorithm 1), we solve $M+1$ LPs for each stationary piece. Then in the bandit setting (Algorithm 2), we largely keep the structure of Algorithm 1 for the purpose of demonstrating the main ideas. However, after running more experiments on problems with larger scales, we find that the aforementioned modification significantly enhanced our algorithm's efficiency. Therefore, we think it's better to implement our algorithm more efficiently. We will revise the manuscript so that Algorithm 2 solves no more than $1$ LP per iteration. We will also supplement more experiments with larger scales in our revised manuscript.
>
> Thank you so much for your valuable advice on computational overhead!

---

### Official Review · Reviewer_SDDb · 2024-07-12

**Soundness:** 3
**Presentation:** 2
**Contribution:** 3
**Rating:** 6
**Confidence:** 2

**Summary:**

This paper studies the bandits with knapsacks problem in a piecewise-stationary environment and designs an algorithm that achieves a provably near-optimal competitive ratio. Instead of using a static benchmark, the performance guarantee is present based on a dynamic benchmark.

**Strengths:**

- The problem setup, piecewise-stationary BwK, complements the BwK literature.
- This paper is well-structured. The warm-up section 3 on deterministic outcome setting really helps in understanding the algorithm for stochastic outcome setting.
- For each algorithm/theoretical result, there are detailed explanations trying to give the intuition behind it.
- The theoretical results are built on mild assumptions. They are solid and technical.

**Weaknesses:**

- It is hard for readers not having expertise in BwK to get the intuition of why the algorithm is near-optimal. Specifically, why IRES is good by reserving an equal budget for each ratio?

**Questions:**

- In line 5 of algorithm 1, why using $\eta_{\min}\cdot\alpha^q$ with restricting $q$ in ${\0,1,...,M-1\}$ to guess $B^*_l/(t_l-t_{l-1})$?
- In Theorem 4.2, is it possible that $opt(FA)$ is close to the term $\tilde{O}(L\sqrt{|\mathcal{K}NT|})$? In this case, would this bound be close to zero regardless of the preceding product term?

**Limitations:**

The authors have discussed the limitations as they claimed in the paper checklist.

---

> ### Author Rebuttal · Authors · 2024-08-05
>
> We sincerely appreciate the reviewer’s time and effort in evaluating our manuscript. Your feedback is highly valued, and we would like to address your concerns and questions point by point.
>
> Regading "Weaknesses": “It is hard for readers not having expertise in BwK to get the intuition of why the algorithm is near-optimal. Specifically, why IRES is good by reserving an equal budget for each ratio?”
>
> Response: We understand and share the concern about making the intuition behind our algorithm clear. In addition to the warm-up section 3, we have included a high-level overview and intuition in Section 2.3 (pages 3-4) to help clarify the technical discussions and makes our contributions more accessible. Based on the reviewer’s feedback, we recognize the need to introduce more foundational concepts of BwK before presenting our idea and will modify our paper accordingly.
>
> BwK differs from traditional multi-armed bandits (MAB) due to resource constraints. Each action in BwK not only yields a reward but also consumes resources from a limited budget, making it crucial to balance exploration and exploitation while ensuring the resource budget is not exceeded. In piecewise-stationary MAB, the performance of a non-anticipatory algorithm $\pi$ is expressed as $\sum_{t=1}^T R_t(a_t) \geq \sum_{t=1}^T r_t(a_t^{\star}) - Reg$, where $\sum_{t=1}^T R_t(a_t)$ is the cumulative reward achieved by algorithm $\pi$, $\sum_{t=1}^T r_t(a_t^{\star})$ is the cumulative expected reward obtained by the best arm selections (with prior knowledge on $r_t$ over the entire planning horizon), and Reg is a sublinear-in-$T$ regret.
>
> In piecewise-stationary Bwk, without assuming bounded global variation (Appendix A.2 on page 11), achieving a sublinear-in-$T$ regret is impossible. Therefore, the performance guarantee is in the form $\sum^T_{t=1} R_t(a_t) \geq \frac{1}{\text{CR}}\cdot \text{opt(FA)} - \text{Reg}$, where opt(FA) is the cumulative expected reward obtained by the best dynamic policy (with prior knowledge on $(r_t, c_t)$ over the entire planning horizon, see Section 2.1 on page), CR is the competitive ratio and Reg is a deductive regret. In our work, we aim to develop algorithms that minimizes CR and ensures Reg to be sublinear-in-$T$. Specifically, IRES (full-feedback deterministic algorithm) and IRES-CM (bandit-feedback stochastic algorithm) both achieve CR=$O(\log (\eta_{\max}/ \eta_{\min})) = O(M)$ (Theorem 3.2, Theorem 4.2), and we provide a matching CR lower bound (Theorem 4.5).
>
> Recall that we decompose the optimal reward opt(FA) into the reward-consumption ratio product the amount of resources assigned for each stationary piece $l$ (see equation (3) on page 4, sorry that the math formulation in the rebuttal may not show well).
> \begin{equation}
> \text{opt(FA)} = \sum_{m=-M}^{M-1} \sum_{l \in \mathcal{L}} \text{Ratio}^{(l)\star} \cdot \mathbf{1}(m^{\star}_l =m) \cdot B^{\star}_l.
> \end{equation}
>
> IRES aims to achieve a reward guarantee for each interval $m$ regarding the reward-consumption ratio $1(m_l^{\star} =m) \cdot \text{Ratio}^{(l)\star}$ and the resource consumption $\sum_{l \in \mathcal{L}} 1(m_l^{\star} =m) \cdot B_l^{\star}$. Regarding why IRES is good by reserving an equal budget for each ratio, a high-level explanation is that by doing so, we reserve adequate inventory (compared to $\sum_{l \in \mathcal{L}} 1(m_l^{\star} =m) \cdot B_l^{\star}$) for the correct guess of $1(m_l^{\star} =m) \cdot \text{Ratio}^{(l)\star}$ for each interval $m$. Specifically, we achieve the guarantee by performing two tasks:
>
> (a) for each $l \in \mathcal{L}$, we guess the value of $m$ such that $\text{Ratio}^{(l) \star} \in (\alpha^m, \alpha^{m+1}]$. By solving a series of LPs in a round-robin manner, we guarantee that for at least $1/(M+1)$ fraction of requests on each $l$, our guessed ratio interval are close to the correct interval $m^{\star}_l$. By accomplishing task (a), we ensure that for each $l \in \mathcal{L}$, at least $1(m_l^{\star} =m) \cdot B_l^{\star}/(M+1)$ requested resources are served by resources reserved for interval $m$, generating reward at a ratio of at least $\alpha^m$.
>
> (b) for each interval $m$, we "reserve" $B/2M$ resource units. That is, we reserve an inventory of $B/2M$ resource units to satisfy requests with a guessed reward-consumption ratio interval $m$. When the inventory reserved for interval $m$ is depleted, the DM rejects (by choosing $a_{\text{null}}$) all future requests with a guessed interval $m$. By accomplishing task (b), if the reserved inventory for interval $m$ are not depleted by round $T$, our algorithm earns a reward of at least $\alpha^m \cdot 1(m_l^{\star} =m) \cdot B_l^{\star}/(M+1)$ during stationary piece $l$. Else, if the reserved $B/2M$ resource units for interval $m$ are depleted by round $T$, then the DM earns a reward of at least $\alpha^m \cdot B/(2M) \geq \alpha^m \cdot \sum_{l \in \mathcal{L}} 1(m_l^{\star} =m) \cdot B_l^{\star}/(2M)$ from resources reserved for interval $m$, since $\sum_{l \in \mathcal{L}} 1(m_l^{\star} =m) \cdot B_l^{\star} \leq B$.
>
> Given (a) & (b), a judicious analysis on the relationship between stationary pieces and reward-consumption ratio intervals is required to ensure a reward of
> \begin{align}
> \frac{1}{O(M)} \cdot \sum_{l \in \mathcal{L}} \alpha^m \cdot \mathbf{1}(m^{\star}_l =m) \cdot B^{\star}_l
> \end{align}
> is accrued for each interval $m$. Hence, we achieve CR$=O(M)$.

---

> ### Author Response · Authors · 2024-08-05
> **Regarding "Questions"**
>
> (We apologize for the weird section division in this part. Otherwise the math notation do not show correctly.)
>
> 1. "In line 5 of algorithm 1, why using $\eta_{\min}\cdot\alpha^q$ with restricting $q \in${$0,1,\ldots,M−1$} to guess $B_l ^{\star} /(t_l-t_{l-1})$?"
>
> Response: We are sorry that there is a typo in line 5 of Algorithm 1. It should be “solve $\text{LP}(r^{(l)}, c^{(l)},\eta_{\min} \cdot \alpha^q)$ $\forall q \in${$0,1,\ldots,M$}” instead of solving for $q \in${$0,1,\ldots,M−1$}. We will correct this typo. Rest asured that the algorithm description and proofs are all correct regarding the value range of $q$. We next explain why we use $\eta_{\min}⋅\alpha^q$ with restricting $q \in${$0,1,\ldots,M$} to guess $B_l ^{\star} /(t_l-t_{l-1})$.
>
> Recall from Section 2.1 that $r^{(l)}(a), c^{(l)}(a) \in [\eta_{\min}, \eta_{\max}]$ for all $l, a$, and recall from Section 2.3 that we define the set $\mathcal{L} = ${$l \in ${$1, \ldots, L$}: $\sum_{a \in \mathcal{K}} x_l^{\star}(a) > 0$}. Then we have $\sum_{a \in \mathcal{K}} c^{(l)}(a) x_l^{(q)\star}(a) \in [\eta_{\min}, \eta_{\max}]$ since $0<\sum _{a \in \mathcal{K}} x^{(q)\star}_l(a) \leq 1$ for all $l \in \mathcal{L}$.
>
> This further indicates that $B_l ^{\star} /(t_l-t_{l-1}) \in [\eta_{\min}, \eta_{\max}]$. Plugging in $q \in${$0,1,\ldots,M$}, we essentially use $M+1$ intervals $[\eta_{\min}, \eta_{\min} \cdot \alpha],  (\eta_{\min} \cdot \alpha, \eta_{\min} \cdot \alpha^2], \ldots, (\eta_{\min} \cdot \alpha^{M-1}, \eta_{\max}]$ to cover $[\eta_{\min}, \eta_{\max}]$ and guess which interval $B_l ^{\star} /(t_l-t_{l-1})$ falls into.
>
> In Claim 3 (see Appendix B.3 on page 12), we show that the round-robin technique in IRES ensures that, on each stationary piece $l$, $x_l^{(q)\star}$ (optimal solution to $\text{LP}(r^{(l)}, c^{(l)},\eta_{\min} \cdot \alpha^q)$) for at least one $q \in${$0,1,\ldots,M$} is close to the $x^{\star}_l$ (optimal solution to FA) in terms of both the resource consumption and the reward-consumption ratio. Recall that we denote $m^{\star}_l \in ${$-M,\ldots, M-1$} as the interval such that $\text{Ratio}^{(l) \star} \in (\alpha^{m^{\star}_l}, \alpha^{m^{\star}_l+1}]$, and we denote $m_t \in ${$-M, \ldots, M-1$} such that $\text{Ratio}_l^{(q_t)} \in (\alpha^{m_t},\alpha^{m_t+1}]$.
>
> Then Claim 3 leads to the important result that for all $l$, in at least $1/(M+1)$ fraction of rounds $t \in ${$t_{l-1}+1, \ldots, t_l$}, we have $m_t \in ${$m^{\star}_l-1,m^{\star}_l$}.
>
> To conclude, by guessing a $q$ such that $\eta_{\min} \cdot \alpha^q$ is within a factor of $\alpha$ from $B_l^{\star}/(t_l - t_{l-1})$ (which happens in at least $1/(M+1)$ fraction of rounds), we also have $\text{Ratio}_l ^{(q)} $ to be at most a factor of $\alpha$ from $\text{Ratio}^{(l)\star}$. Therefore, for at least $1/(M+1)$ fraction of rounds, we have the correct guess of the reward-consumption ratio interval on each stationary piece.
>
> 2. "In Theorem 4.2, is it possible that opt(FA) is close to the term $\tilde{O}(L\sqrt{|\mathcal{K}|NT})$? In this case, would this bound be close to zero regardless of the preceding product term?"
>
> Response: Yes, the reviewer’s point is correct. If opt(FA) is close to $\tilde{O}(L\sqrt{|\mathcal{K}|NT})$, then the bound in Theorem 4.2 will be close to zero. Unfortunately, this order of loss is unavoidable in bandit problems. In stationary unconstrained MAB, the optimal regret is $\tilde{O}(\sqrt{|\mathcal{K}|T})$; in piecewise-stationary unconstrained MAB, the optimal regret is $\tilde{O}(\sqrt{L|\mathcal{K}|T})$. We emphasize that even the theoretical regret exceeds the optimal reward, these results, as well as our result, are still meaningful from the following aspects: (1) The performance bound serves as a measure of how well our algorithm performs relative to the best possible strategy. Even if the theoretical regret exceeds the optimal reward, the bound indicates that our algorithm's regret grows sub-linearly with time $T$; (2) The bound provides a worst-case scenario guarantee, reassuring users that the algorithm won't perform significantly worse than the optimal strategy, even in less favorable conditions; (3) The regret bound provides a theoretical guarantee that the performance will remain within certain limits, offering confidence in the algorithm's consistency and reliability.
>
> Finally, we thank the reviewer for the thoughtful feedback, which has significantly helped us improve our paper. We are happy and open to discussing any further questions.

---

> > ### Comment · Reviewer_SDDb · 2024-08-12
> >
> > Thanks for these detailed explanations. As I am unfamiliar with this topic, it is hard for me to evaluate the value of this job and I prefer to keep my score.

---

### Official Review · Reviewer_BBCY · 2024-07-12

**Soundness:** 3
**Presentation:** 2
**Contribution:** 3
**Rating:** 6
**Confidence:** 3

**Summary:**

This paper addresses the challenge of piecewise non-stationary stochastic bandits with knapsacks. In bandit with knapsacks, at each round a learner is asked to choose an action and receives both a reward and a budget cost. The goal of the learner is to maximize its cumulative reward while satisfying some cumulative budget constraints. The authors prove a competitive ratio with respect to a non-stationary oracle with $L$ changes of order $O(1/\log(\eta_{\max}/\eta_{\min}))$ under the assumption that budget costs smaller than $\eta_\max$ and rewards larger than $\eta_\min$ and if L is sufficiently small. Earlier results on non-stationary bandit with knapsacks are recent and depend on a global variation measure (glo) that is hardly satisfied in practice.

**Strengths:**

- The considered setting is relevant and of interest for practical applications.
- The authors design an algorithm that achieves an optimal competitive ratio with a matching lower bound.
- The non-stationary assumption is much more realistic than previous work.
- Experiments included in the appendix demonstrate that the performance improvements with respect to the only existing baseline. The latter is however designed for the adversarial setting and is thus too conservative in their setting.

**Weaknesses:**

- The main part of the paper is too technical with many different notations, inline mathematical formulas, and not easy too follow.
- The algorithm needs to know the loss and budget bounds in advances.
- The dependence of the regret in $L$ seems suboptimal since the results only hold for $L \leq o(\sqrt{T\eta_{\min}})$ while we expect $L$ to be possibly as large as $o(T)$.
- I am not convinced by the lower bound of Lemma 2.3 which uses $L = T/B$ and is thus in a setting where the upper-bound does not hold. The result would be stronger for a fixed value of $L$.

**Questions:**

- I suggest the authors to address my limitations.
- The paper only considers piecewise stationarity, would it be possible to generalize it to smooth variations of losses and budget costs?

Typos:
- p3, l117: $B^2/T$
- p5, l160: "The DM does not *know*"

**Limitations:**

Yes

---

> ### Author Rebuttal · Authors · 2024-08-05
>
> We appreciate your time and effort in reviewing our paper. We address your concerns and questions one by one below:
>
>
> 1. “The main part of the paper is too technical with many different notations, inline mathematical formulas, and not easy to follow.”
>
> Response: We apologize for the dense notation in our paper. Given the highly technical and unconventional nature of our work, detailed notation is necessary to convey our ideas accurately. To help readers, we have included a high-level overview and intuition in Section 2.3 (pages 3-4). We hope this section clarifies the technical discussions and makes our novel contributions more accessible.
>
>
> 2. “The algorithm needs to know the loss and budget bounds in advances.”
>
> Response: We believe there may be a misunderstanding. Our algorithm requires knowing the range where the reward and resource consumption fall in, but it does **not** require prior knowledge of budget bounds. In the full-feedback deterministic outcome setting in Section 3, not only our algorithm, but also our performance guarantee does not rely on the budget $L$ (see Theorem 3.2 on page 6). In the bandit-feedback stochastic outcome setting in Section 4, while our performance guarantee (see Theorem 4.2 on page 8) depends on $L$, the algorithm itself does not need to know $L$. That being said, we do provide an improved performance guarantee when $L$ is known (see Remark 4.3 on pages 8-9).
>
>
> 3. “The dependence of the regret in $L$ seems suboptimal since the results only hold for $L = o(\sqrt{T \eta_{\min}})$ while we expect $L$ to be possibly as large as $o(T)$.”
>
> Response: We acknowledge that this is indeed a limitation of our piecewise-stationary setting, as discussed in Section 2.2. In the bandit-feedback stochastic outcome setting, without prior knowledge of $L$, our result is meaningful only when $L = o(\sqrt{T \cdot \eta_{\min}})$.  With prior information of $L$, our result is meaningful when $L=o(T \cdot \eta_{\min})$. In the full-feedback deterministic outcome setting, our result is meaningful even when $L=T$.
>
> This limitation arises from our exploration process (line 9-13, Algorithm 2 on page 8) in the bandit-feedback setting. There could potentially be other change point monitoring algorithms where this restriction can be lifted. Nevertheless, our main contribution lies in our novel design of breaking the problem down into guessing the reward-consumption ratio interval and reserving adequate inventory for each interval. This design allows us to achieve a near-optimal competitive ratio without knowing the exact change points, making our contribution meaningful despite this limitation.
>
>
> 4. "I am not convinced by the lower bound of Lemma 2.3 which uses $L=T/B$ and is thus in a setting where the upper-bound does not hold. The result would be stronger for a fixed value of $L$."
>
> Response: We appreciate the reviewer’s careful inspection and insightful concern. The competitive ratio (CR) lower bound in Lemma 2.3 is derived for a full-feedback deterministic outcome setting, where our algorithm has no restriction on $L$. Our deterministic performance guarantee (for $\eta_{\min}>0$) in Theorem 3.2 holds even when $L=T$.  Therefore, $L=T/B$ is in a setting where the upper-bound holds, in a full-feedback deterministic setting.
>
> We do agree with the reviewer that in the bandit-feedback stochastic setting, we require $L = o(\sqrt{T \cdot \eta_{\min}})$ (or $L=o(T \cdot \eta_{\min})$ with prior knowledge of $L$), which indeed contradict Lemma 2.3 where we set $L=T/B$, when $B$ is small. The rationale behind Lemma 2.3 is that we want to provide a bound that surpasses Immorlica et al. (2019), where the CR depends on $T$.
>
> In fact, our Theorem 4.5 on page 9 already provides a valid CR lower bound, where we set $L$ to be a fixed value $2 \log(\eta_{\max}/\eta_{\min})$. The lower bound also show that $\eta_{\min}>0$ is a necessary condition for obtaining a non-trivial CR.
>
>
> 5. “The paper only considers piecewise stationarity, would it be possible to generalize it to smooth variations of losses and budget costs?”
>
> Response: We think it could be possible to generalize our approach to smooth variations, since Algorithm 1 (full-feedback deterministic outcome setting) applies to any non-stationarity (see Theorem 3.2 on page 6). The performance dependence on $L$ in the bandit-feedback stochastic setting arises from our exploration process in Algorithm 2. There could potentially be more refined change point monitoring algorithms that adapt to other types of non-stationarity.
>
> Our key contribution is the new perspective of decomposing the reward into the reward-consumption ratio and the amount of resources assigned for each stationary piece $l$ (see equation (3) on page 4). Then by reserving a fixed amount of inventory for each ratio interval, our task simplifies to guessing the optimal reward-consumption ratio interval in each round. Since our paper is the first to propose this design, we believe a simpler non-stationarity helps us better illustrate our main contribution and make the paper easier to follow.  We will continue to refine our algorithm to adapt to more complex non-stationarity.
>
>
> Finally, we thank the reviewer for the careful inspection and insightful question, which indeed help us to improve our paper! We have corrected the typos and hope that we have clarified your concerns. We are happy and open to discussing any further questions.

---

> > ### Comment · Reviewer_BBCY · 2024-08-09
> >
> > Thank you for your response. This addresses some of the points I raised.
> >
> > > We believe there may be a misunderstanding. Our algorithm requires knowing the range where the reward and resource consumption fall in, but it does not require prior knowledge of budget bounds.
> >
> > This was actually what I meant by my point. Is is possible to not require the knowledge of the reward and ressource ranges?
> >
> > > The competitive ratio (CR) lower bound in Lemma 2.3 is derived for a full-feedback deterministic outcome setting, where our algorithm has no restriction on $L$. Our deterministic performance guarantee (for $\eta_{\min}>0$) in Theorem 3.2 holds even when $L=T$.
> >
> > Thank you. I understand better but this is still unfortunate to have the result for the full-information feedback only. Wouldn't it be possible to have a lower-bound for any fixed value L?
> > Additionally, after revisiting Theorem 3.2, I realize that the paper is indeed quite challenging to engage and the analysis of Thm. 3.2 hard to follow. It's also unclear to me how the dependence on  $L$ is reflected in the result.

---

> > > ### Author Response · Authors · 2024-08-11
> > > **Demonstrating why knowing $\eta_{\min}$ is necessary**
> > >
> > > To show that it is necessary to know $\eta_{\min}$, we let the DM be provided with a lower range $<\eta_{\min}$ (a looser range), and show that it leads to sub-optimal CR. We first construct a general case with $N+1$ instances when $\eta_{\min} = \beta^{-N}$ for some absolute constant $\beta >1$. We consider $N+1$ instances with two arms $\mathcal{K}= \{1\}$ and $a_{\text{null}}$, and instance $n$ happen with probability $p_n$. All instances have deterministic outcomes, and they share the same reward model $R_t(1) = 1$ for all $t$. Their consumption functions are:
> > >
> > > $\text{Instance $1$: }C^{(1)}(1)= \left(\underbrace{1, \ldots, 1}_{B\text{ rounds}}\right)$
> > >
> > > $\text{Instance $2$: }C^{(2)}(1)= (\underbrace{1, \ldots, 1},\underbrace{1/\beta, \ldots, 1/\beta}_{B \cdot \beta\text{ rounds}})$
> > >
> > > $\text{Instance $3$: }C^{(3)}(1)= \left(\underbrace{1, \ldots, 1}, \underbrace{1/\beta, \ldots, 1/\beta}, \underbrace{1/\beta^2, \ldots, 1/\beta^2}_{B \cdot \beta^2\text{ rounds}}\right)$
> > >
> > > $\ldots$
> > >
> > > $\text{Instance }N: C^{(N)}(1)=  \left(\underbrace{1, \ldots, 1}, \underbrace{1/\beta, \ldots, 1/\beta}, \ldots, \underbrace{1/\beta^{N-1}, \ldots, 1/\beta^{N-1}}_{B \cdot \beta^{N-1}\text{ rounds}}\right)$
> > >
> > > $\text{Instance }N+1: C^{(N+1)}(1)=  \left(\underbrace{1, \ldots, 1}, \underbrace{1/\beta, \ldots, 1/\beta}, \ldots, \underbrace{1/\beta^{N-1}, \ldots, 1/\beta^{N-1}}, \underbrace{1/\beta^{N}, \ldots, 1/\beta^{N}}_{B \cdot \beta^{N}\text{ rounds}}\right).$
> > >
> > > (We are sorry that the subfixes for the underbraces cannot show properly in this bubble. In any instance, the $n$-th stationary piece has a length of $B \cdot \beta^{n-1}$ rounds.)
> > >
> > > Through a similar analysis to the proof of Lemma 2.3, we can derive a CR lower bound of $N(1-1/\beta)+1/\beta$ using the instance family above. In the following CR expressions, we omit coefficient regarding $1/\beta$ as they are constants. For example, we write CR$=N(1-1/\beta)+1/\beta$ simply as CR$=\Theta(N)$. When the DM is provided with information $\eta_{\min} = \beta^{-N}$ for any $N \in \mathbb{N}$, a CR lower bound is derived based on $N+1$ instances as constructed above.
> > >
> > > We suppose the real $\eta_{\min} = \min_{t,a} \frac{r_t(a)}{c_t(a)} = \beta^{-\Lambda}$, $\eta_{\max} = \max_{t,a} \frac{r_t(a)}{c_t(a)} = 1$, but the DM only has weak prior information that
> > >
> > > $\tilde{\eta}_{\min}=\beta^{- \kappa \cdot \Lambda}$
> > >
> > > ($\kappa>1$ can be set to be arbitrarily large) and $\eta_{\max} = 1$. Then from the DM's point of view, the optimal CR she/he could derive is CR$=\Theta(\kappa \cdot \Lambda)$, while from the perspective of who knows the real $\eta_{\min} = \beta^{-\Lambda}$, the optimal CR should be $\Theta(\Lambda)$.
> > >
> > > We first show that given the loose lower range $\beta^{- \kappa \cdot \Lambda}$, the DM will not benefit from tightening the value ranges by blindly guessing a value of $\eta_{\min}$. We suppose the DM tightens the value range to $[\beta^{-(\kappa \cdot \Lambda - d)}, 1]$ for some $d \geq 1$, without knowing the real $\eta_{\min}$. Then he/she derives a CR lower bound with $N +1= \kappa \cdot \Lambda - d+1$ instances based on the above construction. Then the DM can expect to achieve a total reward of $\Theta(B \cdot \beta^{\kappa \cdot \Lambda-d}/(\kappa \cdot \Lambda-d))$ (similar analysis with proof of Theorem 2.3). However, since the DM does not know the real $\eta_{\min}$, it is possible that in fact $\eta_{\min} = \beta^{- \kappa \cdot \Lambda}$. In this case, the optimal reward can be as large as $\Omega(B \cdot \beta^{\kappa \cdot \Lambda})$ if $N = \kappa \cdot \Lambda$. Hence, from the DM's perspective, she/he could achieve a sub-optimal CR of $\Theta(\beta^d \cdot (\kappa \cdot \Lambda-d))$ if she/he blindly assume $\eta_{\min} = \beta^{-(\kappa \cdot \Lambda - d)}$, which is significantly worse than the optimal CR$=\Theta(\kappa \cdot \Lambda)$.
> > >
> > > Therefore, the DM must derive a CR on the full range $[\beta^{\kappa \cdot \Lambda},1]$, which involves $N+1=\kappa \cdot \Lambda+1$ instances as constructed above. Therefore the DM expects a reward of $\Theta(B \cdot \beta^{\kappa \cdot \Lambda}/(\kappa \cdot \Lambda))$ when there are indeed $N+1 =\kappa \cdot \Lambda+1$ instances. However, since in fact there are only $\Lambda+1$ instances, the DM wastes all her/his resources reserved for instance $\Lambda+2, \ldots, \kappa \cdot \Lambda+1$ and she/he can only achieve a reward of $O(B \cdot \beta^{\Lambda}/(\kappa \cdot \Lambda))$. Compared with the actual optimal reward $\Omega(B \cdot \beta^{\Lambda})$ with $\Lambda+1$ instances, the DM achieves a sub-optimal CR of $\Omega(\kappa \cdot \Lambda)$. Since $\kappa$ can be arbitrarily large, the CR derived without correct knowledge of $\eta_{\min}$ is significantly worse than the optimal CR$=\Theta(\Lambda)$.

---

> > > ### Author Response · Authors · 2024-08-11
> > > **General response to the comment by Reviewer BBCY**
> > >
> > > We appreciate the reviewer’s insightful and pointed concerns. Please allow us to address them further.
> > >
> > > Regarding the requirement on the knowledge of the reward and resource ranges, unfortunately, it is necessary. While it is possible to derive a worse bound without $\eta_{\max}$ by setting it to its upper bound of 1, knowing the lower bound is essential for our algorithm's functionality. We have provided a case with some light proof in the comment above this one to show that, to achieve a near-optimal CR, it is indeed a necessary condition to know $\eta_{\min}$. To our knowledge, existing literature that derives near-optimal performance bounds with respect to $\eta_{\max}/\eta_{\min}$ typically requires knowledge of both $\eta_{\min}$ and $\eta_{\max}$ (Zhou et al., 2008; Im et al., 2021; Zeynali et al., 2021). Thus, we believe this assumption is reasonable.
> > >
> > > Regarding the analysis in Section 3.2, our algorithm aims to perform two tasks: (i) ensuring at least one out of every $M+1$ guesses on the reward-consumption ratio is accurate for each stationary time segment; (ii) allocating sufficient resources for each reward-consumption ratio interval (interval for short), to establish a near-optimal CR. Our algorithm IRES guarantees a interval-wise $O(M)$ CR. To attain this, we focus on two scenarios: intervals where the reserved resources are depleted before the end of the horizon (Claim 2, see Line 199) and intervals where the reserved resources are not fully consumed until the end (Claim 1, see Line 197). The interval-wise CR naturally leads to the overall CR result.
> > >
> > > For Claim 1, we need the total number of stationary pieces where $t_{l} - t_{l-1} \leq M$ to be at most $o(T)$. Thus we cannot set $L=T/B$ even in the deterministic setting. We have clarified Theorem 3.2’s dependence on $L$ in the revised manuscript and removed Lemma 2.3. Our original mentality behind providing Lemma 2.3 is that we find a CR lower bound which is better than Immorlica et al. (2019) w.r.t. $T$, but we indeed fail to keep it aligned with our assumptions. Thank you very much for pointing this out! We could potentially make Theorem 3.2 stands when $L=O(T)$ by modifying the algorithm to draw $q$ (see line 5 of Algorithm 1) in a randomized manner (instead of a round-robin manner). However, we agree with you that this is only applicable in the full-feedback setting.
> > >
> > > Despite the limitation behind Lemma 2.3, we highlight that its proof (see Appendix C.5) still provides a valid CR lower bound of $\Omega(L)$, which applies to any legit $L$ values. In the proof of Theorem 4.5 (Appendix C.6), we essentially show that $L$ can be no larger than $\log(\eta_{\max}/\eta_{\min})$ in our constructed instance, leading to a matching CR lower bound to our performance guarantee. We will revise these proofs to clearly present the lower bound for all valid $L$ values. We hope this addresses your concern regarding the CR lower bound.
> > >
> > > We further remark that demonstrating Claim 1 and Claim 2 involves a judicious analysis on the rewards gained on each stationary piece and the rewards gained for each ratio interval, which depend heavily on the resource consumption status for each interval. Hence, the somewhat complex analysis involving the term $\tilde{\mathcal{T}}^{(m)}$ (defined in Section 3.3) is necessary. We understand and share the concern about making the intuitions clear (as our paper is technical and highly unconventional). Thus, we strive to shed light on our analysis by providing a high-level overview and intuition in Section 2.3 (pages 3-4), making our contributions more accessible. We have also revised Theorem 3.2 to clearly show the dependence on $L$.
> > >
> > > To conclude our comments:
> > >
> > > 1. Knowing $\eta_{\min}, \eta_{\max}$ beforehand is a necessary condition for any non-anticipatory algorithm to achieve a CR of $O(\log(\eta_{\max}/\eta_{\min})), but we believe it is a mild assumption since it is assumed in relevant papers. We will clearly point out this limitation in Section 2.2.
> > >
> > > 2. Our Lemma 2.3 of setting $L=T/B$ is not aligned with our assumption, yet we do provide a CR lower bound for all legit $L$.
> > >
> > > 3. While the notation in our analysis are unavoidable, we will ensure that the dependence on $L$ is clearly highlighted in Theorem 3.2.
> > >
> > > We hope this addresses all your concerns. Despite the limitations, we believe our paper is novel and makes significant advances over existing works.
> > >
> > > References
> > >
> > > [1] Im, S., Kumar, R., Montazer Qaem, M., & Purohit, M. (2021). Online knapsack with frequency predictions. Advances in neural information processing systems, 34, 2733-2743.
> > >
> > > [2] Zeynali, A., Sun, B., Hajiesmaili, M., & Wierman, A. (2021, May). Data-driven competitive algorithms for online knapsack and set cover. In Proceedings of the AAAI Conference on Artificial Intelligence (Vol. 35, No. 12, pp. 10833-10841).

---

> > > > ### Comment · Reviewer_BBCY · 2024-08-12
> > > >
> > > > Thank you for your detailed response. I understand know the difficulties better and increase my score to 6. All the best.

---

> > > > > ### Author Response · Authors · 2024-08-12
> > > > >
> > > > > Thank you for your understanding and for increasing the score; we appreciate your thoughtful consideration of the challenges involved. All the best to you!

---

### Official Review · Reviewer_ifdP · 2024-07-14

**Soundness:** 2
**Presentation:** 3
**Contribution:** 2
**Rating:** 5
**Confidence:** 4

**Summary:**

The paper studies Bandits with Knapsacks in a piecewise-stationary environment, where the underlying reward can change over time.
The authors provide provably near-optimal competitive ratio for this setting, which achieves a dynamic benchmark and obtains stronger results compared to existing adversarial Bwk works. Specifically, the algorithm proposed in the paper does not rely on the prior information of the number of stationary pieces and the time indexes when changes happen.

**Strengths:**

1. The contribution of this work is solid. Compared to previous works that use the global variation glo to quantify performance, the algorithm in this paper uses the number of reward changes as a measure. This is a better choice in non-stationary bandits in general.
2. The paper is well written and provides adequate details to understand the flow of the material.

**Weaknesses:**

As I understand, the main idea of this paper is to categorize the rewards into $M$ levels based on the maximum per unit reward (Ratio*). Each level's maximum per unit reward is a multiple of the previous one, allowing to focus solely on the highest level's maximum per unit reward. In this setting, the algorithm allocates $\frac{B}{2M}$ resources to each level. In this regard, the algorithm's reward is at least the maximum per unit reward multiplied by $ \frac{B}{2M} $, while the optimal algorithm's profit is at most the maximum unit profit multiplied by $ B $. This leads to a competitive ratio $O(M) = O(\log(\eta_{max}/\eta_{min} ))$.
In this regard, such a design seems somewhat trivial. I can understand that for this problem, perhaps this method is optimal. However, given such an algorithm, I am not sure if the problem itself is sufficiently meaningful.


Typos:
1. Line 160: The DM does not L...

**Questions:**

For the non-stationary bandit research I known before, algorithms typically need to detect whether the reward has changed every round. When a change is detected, the algorithm usually employs a restart mechanism to forget the past information. However, in Algorithm 2 of this paper, there is no such detection-restart mechanism. In this regard, I am curious about how the non-stationary rewards impact the algorithm in this paper.

**Limitations:**

No limitations.

---

> ### Author Rebuttal · Authors · 2024-08-02
>
> We are grateful for the reviewer's time and effort in evaluating our manuscript. We highly value your feedback and would like to address your concerns and questions in what follows.
>
> Regarding "Weaknesses":
>
> 1.	Clarification on Algorithm Details (Full-Feedback Deterministic Outcome Setting): We would like to point out some missing situations and details in the reviewer's summary, which could potentially resolve some possible misunderstanding. The reviewer’s summary, "the algorithm's reward is at least the maximum per unit reward multiplied by $𝐵/2𝑀$, while the optimal algorithm's profit is at most the maximum unit profit multiplied by $𝐵$," captures only the special case when our reserved $𝐵/2𝑀$ units of inventory for each reward-consumption ratio interval are depleted before the end of the planning horizon. The more general and complex scenario arises when our reserved $𝐵/2𝑀$ units are fully consumed for some ratio intervals $m \in \mathcal{M}_2$ (see Claim 2, Section 3.3, on page 6) and not fully consumed for some ratio intervals $m \in \mathcal{M}_1$ (see Claim 1, Section 3.3, on page 6). In the latter case, our algorithm ensures that on each stationary piece $l$ with the optimal ratio $\text{Ratio}^{(l)\star}$ falling in intervals $m \in \mathcal{M}_1$, at least $B^*_l/M$ units are consumed at the optimal ratio. However, it does not provide any direct guarantee that $1/O(M)$ fraction of the optimal algorithm's profit is obtained for intervals $m \in \mathcal{M}_1$. Establishing our competitive ratio requires judicious analysis to bridge between the piece-wise and interval-wise results, which is not trivial (see Appendices B.4, C.2, C.3).
>
> 2.	Clarification on Algorithm Details (Bandit-Feedback Stochastic Outcome Setting): Our primary focus is on the bandit-feedback stochastic outcome setting, where existing algorithms for non-stationary unconstrained multi-armed bandits (e.g., sliding window) cannot be naïvely generalized. Our novel design allows for monitoring underlying changes in parameters through sampling rounds rather than change point detection algorithms, which further simplifies the problem. More details, please refer to the next bubble where we address your questions.
>
> 3.	Comparison with Prior Work: Previous works on online non-stationary optimization/bandits with knapsacks focus on either adversarial settings (Immorlica et al., 2019; Kesselheim and Singla, 2020) or a global variation budget (Jiang et al., 2020; Balseiro et al., 2022; Liu et al., 2022). We believe our piecewise-stationary setting is meaningful both theoretically and practically:
>
>   $\bullet$	 Theoretically, Immorlica et al. (2019), Kesselheim and Singla (2020) consider a more general non-stationary setting than ours, but compare their performance with a stationary benchmark where a fixed optimal arm (or a fixed optimal distribution over arms) is applied in all $T$ rounds. Jiang et al. (2020), Balseiro et al. (2022), Liu et al. (2022) require the total non-stationarity to be bounded within a global variation budget (see Remark 4.2), which is a very strong assumption. Our setting does not require bounded global variation and our performance guarantee is compared with a dynamic benchmark, which we find to be an interesting and meaningful result.
>
>   $\bullet$	Practically, the adversarial setting may be too conservative, and the global variation budget setting can apply to very limited data structures. Our piecewise assumption on non-stationarity is reasonable and observable in many real-life scenarios, such as sales patterns that remain stationary for certain periods before changing during hot seasons/promotions/new trends.
>
>   We further argue that piecewise-stationary bandits without constraints has drawn significant research attention. We list some of the relevant papers in the following:
>
>   [1] Auer, P., Gajane, P., & Ortner, R. (2019, June). Adaptively tracking the best bandit arm with an unknown number of distribution changes. In Conference on Learning Theory (pp. 138-158). PMLR.
>
>   [2] Cao, Y., Wen, Z., Kveton, B., & Xie, Y. (2019, April). Nearly optimal adaptive procedure with change detection for piecewise-stationary bandit. In The 22nd International Conference on Artificial Intelligence and Statistics (pp. 418-427). PMLR.
>
>   [3] Besson, L., Kaufmann, E., Maillard, O. A., & Seznec, J. (2022). Efficient change-point detection for tackling piecewise-stationary bandits. Journal of Machine Learning Research, 23(77), 1-40.
>
>   [4] Bhatt, S., Fang, G., & Li, P. (2023, April). Piecewise stationary bandits under risk criteria. In International Conference on Artificial Intelligence and Statistics (pp. 4313-4335). PMLR.
>
>   Our work is the first to study piecewise-stationary bandits with knapsacks, which is a new and interesting setting that requires extra care in maintaining the order optimal competitive ratio in Section 4.
>
> 4.	Simplicity and Contribution: While our algorithm is not complex, its simplicity and intuitiveness enhance our contribution. We offer a novel perspective on non-stationary BwK, distinct from traditional gradient descent-based algorithms. Our approach simplifies the problem by focusing on two elements: guessing the reward-consumption ratio interval and reserving adequate inventory for each interval. Although the process of deciding ratio intervals via LPs in a round-robin manner may seem primitive, it leaves room for further refinement of our competitive ratio's coefficients (we already achieve the optimal competitive ratio’s order w.r.t. $\eta_{\max}/\eta_{\min}$). Nevertheless, given that our paper introduces this novel perspective, we believe our contribution is solid.

---

> ### Author Response · Authors · 2024-08-02
> **Response to "Questions"**
>
> Regarding "Questions": We appreciate your question on how our algorithm monitors changes in rewards and resource consumption without a restarting mechanism. The key is that we decompose the reward into the reward-consumption ratio product the amount of resources assigned for each stationary piece $l$ (see equation (3) on page 4, sorry that the math formulation in comment may not show well).
> \begin{equation}
> \text{opt(FA)} = \sum_{m=-M}^{M-1} \sum_{l \in \mathcal{L}} \text{Ratio}^{(l)\star} \cdot \mathbf{1}(m^{\star}_l =m) \cdot B^{\star}_l.
> \end{equation}
> By reserving a fixed amount of inventory for each ratio interval, our task simplifies to guessing the optimal reward-consumption ratio interval in each round.
>
> In the full-feedback deterministic outcome setting (Algorithm 1 on page 5),
> \begin{equation}
> \text{Ratio}^{(q)} := \frac{\sum_{a \in K} r^{(l)}(a) x_l^{(q) \star}(a)}{\sum_{a \in K} c^{(l)}(a) x_l^{(q)\star}(a)},
> \end{equation}
> which is a guess of $\text{Ratio}^{(l)\star}$, is observable for each $q$. Then it suffices to check the ratio interval that $\text{Ratio}_{l}^{(q)}$ belongs to.
>
> In the bandit-feedback stochastic outcome setting, we only need an additional estimate on $(r_t(a), c_t(a))$ to ensure the estimated $Ratio^{(l)\star}$  falls within the correct interval. This is achieved through random sampling rounds, termed "exploration rounds" (see Lines 7-13 in Algorithm 2). Specifically, in each round $t$, we conduct sampling with probability $\gamma_t$, where we uniformly at random choose an arm $a \in \mathcal{K}$ and pull it for $N$ times (see equation (5) on page 7). Then we update $(\hat{r}_t(a), \hat{c}_t(a))$, which is an estimate on $(r_t(a), c_t(a))$, as in equation (5) on page 7. We then guess $\text{Ratio}^{(l)\star}$ based on $(\hat{r}_t(a), \hat{c}_t(a))$.
>
> We highlight that the major performance difference between IRES and IRES-CM is the loss caused by estimating $(r_t,c_t)$, reflected in the following aspects: (i) reward loss caused by exploration rounds; (ii) the most recent exploration rounds contain change points, causing failed estimation of $(r_t, c_t)$; (iii) the most recent exploration rounds do not contain change points, but there is a large discrepancy between $(r_t,c_t)$ and $(\hat{r}_t,\hat{c}_t)$ (due to untimely update).
>
> The performance guarantee of our algorithms is in the form of $\sum^T_{t=1} R_t(a_t)\geq \frac{1}{\text{CR}}\cdot \text{opt}(\text{FA}) - \text{Reg}$. In Appendix B.7 & C.4, we respectively prove that the losses due to (i, ii) are accounted for in Reg, while (iii) is accounted for in the CR, with high probability.
>
> In the full-feedback deterministic outcome setting, our performance guarantee holds as long as $L=o(T)$. Nevertheless, in the bandit-feedback stochastic outcome setting, $L$ shows up in Reg$=\tilde{O}(L \sqrt{|\mathcal{K}|NT})$ (or Reg=$\tilde{O}(\sqrt{L|\mathcal{K}|NT})$ when $L$ is known) (see Theorem 4.2 on page 8). Reg is a sublinear-in-$T$ reward loss, provided $L=o(\sqrt{\eta_{\min}T})$ (or $L=o(\eta_{\min}T)$ when $L$ is known). While our sampling process is effective, we acknowledge that it could potentially be complemented or replaced by other change point monitoring algorithms.
>
> We finally remark that our algorithm design does not need to detect the exact change points, but our sampling (exploration rounds) works primarily due to the piecewise-stationary nature. The intuition is that, given the presence of resource constraints, identifying the change point is not enough, different from the unconstrained setting. Indeed, even in the simple case when we know there is only one change point occurring at $t=T/2$, the DM still needs to estimate how many resource units are consumed in the first piece $\{1, \ldots, T/2\}$ and in the second piece $\{T/2 + 1, \ldots, T\}$ in the optimal solution. This means that, during $\{1, \ldots, T/2\}$, the DM shall need some form of knowledge on the mean outcome in the second piece. Such a need is different from the setting without resource constraint, where the DM can achieves a SOTA regret bound by restarting the standard UCB at $t=T/2$. This explains why we do not have a detect-restart mechanism frequently used in unconstrained cases, but we discover that our sampling strategy is better suited for the resource constrained setting.
>
> We thank the reviewer for the valuable comments, and we have corrected the typos. We hope we have clarified all your concerns, and we hope you will consider re-evaluating our contributions after the clarification. We are happy and open to discussing any further questions.

---

### Author Rebuttal · Authors · 2024-08-07

Dear Review Team, we are grateful for your careful reading and thoughtful comments. They are highly relevant and very insightful. On top of the point-to-point responses to individual reviewers, we would like to summarize and clarify critical concerns.

1. Are the problem and the algorithm sufficiently meaningful?

Response: We believe our piecewise-stationary setting is meaningful. Theoretically, unconstrained piecewise-stationary bandits has drawn significant attention in recent years, with many interesting results derived. However, no prior paper addresses the bandits with knapsacks (Bwk) case. In the presence of resource constraints, merely detecting change points in {$(r_t, c_t)$}$_{t=1}^T$ is insufficient. Even with known change points, not knowing the means ${r^{(l)}, c^{(l)}}$ in each stationary piece $l = 1, \ldots, L$ prevents us from determining the optimal resource consumption in each stationary piece. Thus, besides hedging against arbitrary change points, we must address the uncertainty in ${r^{(l)}, c^{(l)}}$ and its impact on the "correct" amount of resource consumption. This marks a difference from unconstrained non-stationary bandits. Practically, our setting is more realistic than existing stationary/pure adversarial/bounded global variation settings.

We believe our algorithm is highly meaningful. Our Sections 3, 4 concern the **full** and **bandit** feedback settings respectively. In the former, the DM observes $r_t(a), c_t(a)$ before choosing $A_t$. In the latter, the DM only observes $R_t(A_t), C_t(A_t)$ (with means $r_t(A_t), c_t(A_t)$) after choosing $A_t$. In Section 3, we provide a novel perspective on the non-stationary Bwk problem, where we decompose the reward into two terms: the reward-consumption ratio multiplied by the resource consumption on each stationary piece. Correspondingly, we propose an intuitive inventory reservation algorithm by reserving a certain amount of inventory for each ratio interval and guessing the optimal reward-consumption ratio interval in each round. In Section 4, we carefully control the estimation procedure on {$(r_t, c_t)$}$_{t=1}^T$, which maintains the competitive ratio while achieving a regret term similar to that in unconstrained piecewise-stationary bandits. We provide the first provably near-optimal performance guarantee in the piecewise stationary BwK), which (a) **compares against the true optimum**, and (b) **allows bandit feedback**. Our ideas and algorithms are completely novel, and the technical details of our proofs are not at all trivial or similar to any existing paper.

2. Requiring $r_t(a), c_t(a) \in [\eta_{\min}, \eta_{\max}]$ for all $t,a$ weakens the result.

Response: We have redefined $\eta$ so that $r_t(a)/c_t(a) \in [\eta_{\min}, \eta_{\max}]$ for all $t,a$. We have revised our algorithm mildly to achieve the same near-optimal competitive ratio, which is a stronger result. More details are provided in the response to question 1 of Reviewer engA.

3. Novelty of algorithm design compared to Immorlica et al. (2019), Zhou et al. (2008) and other existing works.

Response:

$\bullet$ Comparing with online matching papers (e.g. Zhou et al. (2008)):

&nbsp;&nbsp;&nbsp; (i) Requiring less information: Our Section 3 has the same full feedback setting as online matching papers, which crucially require observing the actual values of $(R_t(a), C_t(a))$ for all $a \in \mathcal{K}$ **before** choosing $A_t$. Online matching algorithms cannot be readily generalized to our bandit setting (Section 4), where the DM only observes the outcomes $(R_t(A_t), C_t(A_t))$ **after** pulling arm $A_t$. The natural generalization from full to bandit feedback is one of our core contributions.

&nbsp;&nbsp;&nbsp; (ii) Stronger benchmark: Zhou et al. (2008) measures the performance of their algorithm by the **best single arm** benchmark which requires pulling a single arm in each round; while our benchmark opt(FA) is a **best distribution over arms** benchmark, which is theoretically much stronger than Zhou et al. (2008)'s benchmark.

&nbsp;&nbsp;&nbsp; (iii) More general results: In the full feedback setting, our performance guarantee encompasses Zhou et al. (2008), and in Section 4 we bypass the difficulty in Zhou et al. (2008) with the bandit setting.

$\bullet$ Comparing with inventory reservation papers: Immorlica et al. (2019)’s inventory reservation strategy depends solely on the cumulated reward without considering the reward-consumption ratio, using traditional gradient descent-based algorithms. Our experiments demonstrate that their inventory reservation strategy is significantly more conservative than ours in the piecewise-stationary setting. Other works such as Ball and Queyranne (2009) all focus on different or more specialized settings. Our inventory reservation strategy is highly distinctive from existing methods.

$\bullet$ Comparing with non-stationary Bwk papers: Adversarial Bwk papers (Immorlica et al. (2019)) consider a more general non-stationary setting than ours, but compare their performance with a stationary benchmark where a fixed optimal arm (or a fixed distribution over arms) is applied in all $T$ rounds. Bwk with bounded variation papers (Liu et al. (2022)) require the total parameter variation to be bounded in terms of a global budget, which is a very strong assumption. Our setting does not require bounded global variation and our performance guarantee is compared with a dynamic benchmark, which makes our result strong both theoretically and practically, and comparatively more realistic.

4. Lower bound of Lemma 2.3 requires $L=T/B$, which does not look convincing.

Response: Theorem 4.5 already provides a valid CR lower bound, where we set $L$ to be a fixed value $2 \log(\eta_{\max}/\eta_{\min})$ and show that $\eta_{\min}>0$ is a necessary condition for obtaining a non-trivial CR.

We thank the review team for the valuable comments, and we are happy and open to discussing any further questions.

---

### Decision · Program_Chairs · 2024-09-25

**Decision:**

Accept (poster)

**Comment:**

The paper studies the bandits with knapsacks (BwK) problem under a piecewise-stationary setting. The setting is different from the existing works on adversarial BwK and nonstationary BwK. It develops a new algorithm with the corresponding competitiveness ratio; in my opinion, the competitiveness ratio provides a different characterization (compared to the existing works) on how the algorithm performance is related to the distribution shift over time.

Please consider including (i) more motivation for the setup, (ii) discussions compared to related works, and (iii) the new experiments in your future version of the paper.